# BOOM: Benchmarking Out-Of-distribution Molecular Property Predictions of Machine Learning Models

**Evan R. Antoniuk**[†][*]     **Shehtab Zaman**[‡][*]     **Tal Ben-Nun**[†]     **Peggy Li**[†]

**James Diffenderfer**[†]     **Busra Demirci**[‡]     **Obadiah Smolenski**[‡]     **Tim Hsu**[†]

**Anna M. Hiszpanski**[†]     **Kenneth Chiu**[‡]     **Bhavya Kailkhura**[†]

**Brian Van Essen**[†]

[†] Lawrence Livermore National Laboratory, Livermore, CA
[‡] Binghamton University, Binghamton, NY

## Abstract

Data-driven molecular discovery leverages artificial intelligence/machine learning (AI/ML) and generative modeling to filter and design novel molecules. Discovering novel molecules requires accurate out-of-distribution (OOD) predictions, but ML models struggle to generalize OOD. Currently, no systematic benchmarks exist for molecular OOD prediction tasks. We present BOOM, **b**enchmarks for **o**ut-**o**f-**d**istribution **m**olecular property predictions: a chemically-informed benchmark for OOD performance on common molecular property prediction tasks. We evaluate over 150 model-task combinations to benchmark deep learning models on OOD performance. Overall, we find that no existing model achieves strong generalization across all tasks: even the top-performing model exhibited an average OOD error $3\times$ higher than in-distribution. Current chemical foundation models do not show strong OOD extrapolation, while models with high inductive bias can perform well on OOD tasks with simple, specific properties. We perform extensive ablation experiments, highlighting how data generation, pre-training, hyperparameter optimization, model architecture, and molecular representation impact OOD performance. Developing models with strong out-of-distribution (OOD) generalization is a new frontier challenge in chemical machine learning (ML). This open-source benchmark is available at `https://github.com/FLASK-LLNL/BOOM`.

## 1 Introduction

Molecular discovery pipelines have increasingly relied upon machine learning (ML) models [Bohacek et al., 1996, Reymond, 2015, Kailkhura et al., 2019]. These models discover new molecules by either screening a list of enumerated molecules or by guiding a generative model towards molecules of interest [Wang et al., 2023a]. Molecular discovery is inherently an out-of-distribution (OOD) prediction problem, since the molecules need to either (i) exhibit properties that extrapolate beyond the training dataset, or (ii) possess a previously unconsidered chemical substructure. In either case, success depends on the learned model's ability to make accurate predictions on samples that are not in the same distribution as the training data.

Despite the importance of OOD performance to real-world molecular discovery, the OOD performance of common ML models for molecular property prediction has yet to be systematically explored. Due to the lack of standardized splits for testing models, especially splits based on the data distribution, we believe that current ML models are optimizing in-distribution performance on

---

[*]Equal Contribution

insufficiently challenging datasets that do not adequately measure real-world performance. Currently, little empirical knowledge exists about how choices regarding the pretraining task, model architecture, and/or dataset diversity impact the generalization performance of chemistry foundation models that are expected to generalize across all chemical systems.

In this work, we develop BOOM, **b**enchmarks for **o**ut-**o**f-distribution **m**olecular property predictions, a standardized benchmark for assessing the OOD generalization performance of molecule property prediction models. Our work consists of the following main contributions:

- We develop a general and robust methodology for evaluating the performance of chemical property prediction models for property values beyond their training distribution. We introduce OOD-specific metrics such as binned $R^2$ to allow comparisons of OOD performance across all models.

- We perform the first large-scale OOD performance benchmarking of state-of-the-art ML chemical property prediction models. Across 10 diverse OOD tasks and 15 models, we do not find any existing models that show strong OOD generalization across all tasks. We therefore put forth BOOM OOD property prediction as a frontier challenge for chemical foundation models.

- Our work highlights insights into how pretraining strategies, model architecture, molecular representation, and data augmentation impact OOD performance. These findings point towards strategies for the chemistry community to achieve chemical foundation models with strong OOD generalization across all chemical systems.

## 2  BOOM

**Defining Out-of-distribution.**     Consider a supervised dataset $\mathcal{D}$ with $N$ molecules $\mathcal{M} \in \{\mathcal{M}_1, \mathcal{M}_2, ..., \mathcal{M}_N\}$ and associated labels or properties $y \in \{y_1, y_2, ..., y_N\}$. The problem of out-of-distribution prediction can be defined as the mismatch in the probability distribution, $P$ of the training and test sets, $\mathcal{D}_{\text{train}}$ and $\mathcal{D}_{\text{test}}$ such that,

$$P(\mathcal{M}, y | \mathcal{D}_{test}) \neq P(\mathcal{M}, y | \mathcal{D}_{train}) \tag{1}$$

The key question is defining the density function $P(\mathcal{M}, y)$ over a set of molecules and their respective properties. The density can be defined over the chemical structure or molecule features, or over the properties. Formally, we define out-of-distribution as low-density regions over the property space, such that:

$$0 < P(y_{\text{test}}) \leq P(y_{\text{train}}) \tag{2}$$

Farquhar and Gal [2022] define this as a complement distribution conditioned on the targets. This is known as concept or label shift as well [Liu et al., 2024]. While we focus on designing splits with a concept shift, it is important to note that depending on the property, this may result in a covariate shit, resulting in a structural or chemical imbalance. The probability density over the labels is determined using kernel density estimation (KDE), allowing us to generalize to multimodal distributions. The split strategy algorithm for each dataset is detailed in Appendix A.1. The lowest probability samples from the KDE estimated distribution are held-out (see Fig. 1) to evaluate the consistency of ML models to discover molecules with state-of-the-art properties that extrapolate beyond the training data.

**Datasets.**     BOOM consists of 10 quantum chemical molecular property datasets derived from QM9 [Ramakrishnan et al., 2014] and the 10k Dataset [Antoniuk et al., 2025], derived from the Cambridge Structural Database. The 10k Dataset was sourced from 10,206 experimentally synthesized, small organic molecules and contains the density functional theory calculated values of their molecular density and solid heat of formation (HoF). We collect 8 molecular property datasets from the QM9 Dataset: isotropic polarizability ($\alpha$), heat capacity ($C_v$), highest occupied molecular orbital (HOMO) energy, lowest unoccupied molecular orbital (LUMO) energy, HOMO-LUMO gap, dipole moment ($\mu$), electronic spatial extent ($\langle R^2 \rangle$), and zero point vibrational energy (ZPVE). We also select a random subset of the dataset to serve as the ID test set, detailed in Appendix A. To further expand the application space of BOOM, we also perform benchmarking on the Lipophilicity dataset[Wu et al., 2018] of 4,200 experimental measurements of the octanol/water distribution coefficient, which is of relevance for drug compounds. The inclusion of the Lipophilicity dataset serves as an exemplary

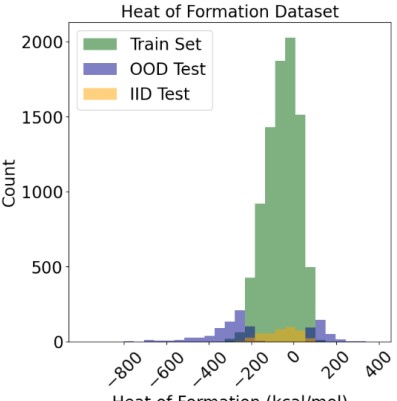 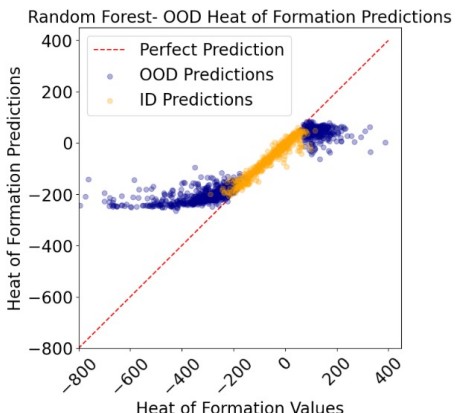

Figure 1: (Left) An example OOD dataset included in the BOOM benchmark. To assess OOD performance, we split each chemical property dataset into an out-of-distribution (OOD) Test Set (blue), an in-distribution (ID) Test Set (orange) and a Train Set (green), as described in Section 2. (Right) Example model predictions on this task exhibiting weak correlation on the OOD samples.

dataset for performing OOD evaluations on experimentally measured properties, rather than only computed physicochemical properties (See Table 9).

**Metrics.** We also propose standardized metrics over the ID and OOD to compare models. We use root mean square error (RMSE) over respective data splits. Models achieving OOD RMSE comparable to ID RMSE show strong generalization. Short of achieving strong OOD generalization, the next-best case is for the model to achieve a strong correlation on the OOD samples. As OOD predictions span disparate ranges in the prediction space by design, the sample mean is far from all the samples and results in a large total sum of squares, artificially increasing the coefficient of determination. Therefore, we evaluate the correlation on the OOD samples by calculating a *binned* $R^2$ value, which is the average value $R^2$ of the OOD samples in the lower and upper tails of the property distribution. For all experiments, we perform 3 training runs and report the average and variance of each performance metric.

**Models.** To evaluate BOOM, we use a number of traditional ML models, GNNs, and hybrid architectures to compare against large-scale transformer models. Traditional ML models utilize molecular fingerprints or other vector representations of molecules as input to statistical methods. We use RDKit Featurizer [Landrum et al., 2013] coupled with a Random Forest regressor and a multilayer perceptron (MLP) as the baseline structure-to-property models. We choose four representative transformer-based models: MoLFormer [Ross et al., 2022], ChemBERTa [Chithrananda et al., 2020], Regression Transformer [Born and Manica, 2023], and ModernBERT [Warner et al., 2024]. We also explore recent 3D molecular models, GotenNet and Geoformer.[Aykent and Xia, 2024, Wang et al., 2023b] The model and training details are presented in Appendices B.3 and C.2, respectively.

## 3 Related Work

OOD predictions present a key challenge for incorporating data-driven models into production pipelines where test time input may significantly shift from training data [Yang et al., 2022a, Liu et al., 2021, Salehi et al., 2021]. OOD detection has been approached through the lens of anomaly detection, uncertainty quantification [Abdar et al., 2021], and open-set detection [Scheirer et al., 2012, Bendale and Boult, 2016, Bulusu et al., 2020]. OOD generalization has also been investigated from the lens of invariant risk minimization [Ahuja et al., 2021], but has not been tailored for molecular discovery and property prediction. Yang et al. [2022b] derive Mole-OOD, a representation learning framework based on invariant learning to learn molecular properties on only varying graph structural environments. Similarly, Liu et al. [2024] and Shen et al. [2024] focus on OOD generalization solely on graph models, while BOOM is applicable for molecules in any representation.

Dunn et al. [2020] present MatBench, a benchmark for inorganic crystalline materials with regression and classification tasks. Omee et al. [2024] proposes a follow-up of MatBench with structure and

property-based OOD for graph neural networks. Li et al. [2025] similarly propose structure and composition-based OOD for materials. Our work differs from MatFold/Matbench in that i) we focus on OOD generalization in the property (y) space, instead of the input (x) space, and ii) evaluate small molecule properties instead of inorganic crystalline materials.

For small molecules, MoleculeNet Wu et al. [2018] is a widely used benchmark for molecular property prediction models consisting of 17 small molecule prediction tasks, along with four splitting protocols: random, scaffold splitting, stratified splitting, and time splitting (test set consists of the newest data). Segal et al. [2025] also explore zero-shot extrapolation of molecular properties on the MoleculeNet dataset prediction beyond the training data. Similar to our work, they also define OOD samples in the property space, but focus on drug-like properties. While they focus on descriptor-based models, BOOM focuses on intensive and extensive quantum chemical properties that are representation agnostic. Ji et al. [2023] curate OOD datasets focused on drug-like molecules, focusing on defining a structure-based definition of molecules such as the molecular size, paired protein and protein family, and binding assay.

## 4  Results

### 4.1  Model Architectures Performance

The OOD performance of our selected models are summarized in Table 7. The leaderboard is presented with a heatmap in Fig. 2. These results were obtained with models used "out-of-the-box". However, we perform hyperparameter optimizations of the training parameters to achieve the highest possible accuracy for each task. Additional visualizations are provided in Appendix C.5.

Overall, we do not find any model that clearly outperforms the others on ID performance across all tasks, but SOTA models like GotenNet and GeoFormer consistently perform strongly across all tasks. The Geoformer achieves the best overall ID performance, achieving the lowest ID RMSE on 3 out of 10 tasks. For OOD prediction, GotenNet achieves top performance on 7 out of 10 tasks, and MACE achieves top performance on 2 out of 10 tasks. The strong performance of these models shows a strong indication that newer models with improved inductive biases perform well on these challenging tasks.

We note that the large OOD RMSEs Regression Transformer were found to arise from inaccuracies in the autoregressive numerical token generation, for example, predicting '00913', for a true value of '0.913'. Figure 1 (right) shows a common mode of failure for OOD predictions for most models (see parity plots for all models tested in Appendix D). We find that models performing poorly on OOD splits overwhelmingly produce an S-shaped parity plot. The models are therefore capable of clustering OOD samples together but are unable to extend the prediction region beyond the training data. Such S-shaped behavior is a known failure case that arises when models learn shortcut features that maximize ID performance, but fail to generalize to OOD data [Geirhos et al., 2020].

Furthermore, we notice a trend where ID performance is not necessarily correlated with OOD performance. MoLFormer, one of the largest models in our test suite, achieves top ID performance on $C_v$, greatly outperforming similar Transformer models like ChemBERTa and RT. But ChemBERTa and MoLFormer achieve similar results on OOD for both tasks. Considering the size of our datasets, we believe large models may be able to overfit to the ID space, while achieving subpar generalization. This suggests the common strategy of pre-training on large datasets and fine-tuning on niche domains may have pitfalls for OOD samples.

Fig. 3. shows results by task. The larger the difference between the ID and OOD bars, the higher the discrepancy between ID and OOD performance. As expected, ID performance is better than OOD performance for all model-task pairs. We highlight that some models achieve strong OOD performance on certain tasks, such as HoF, Density, ZPVE, and $C_v$, that is comparable to ID-level performance. However, Fig. 3 also highlights particular tasks (HOMO, LUMO, Gap, and $\mu$) where no models achieve good OOD performance. Since all these properties are related to the electronic structure of molecules, we hypothesize that the inability of any model to generalize well in these tasks is due to the lack of explicit electronic structure in their molecular representations. It is also important to note that for properties such as $C_v$, although most models achieve similar ID $R^2$ values, there is a large variance in OOD binned $R^2$ values— further highlighting the importance of performing OOD performance evaluations.

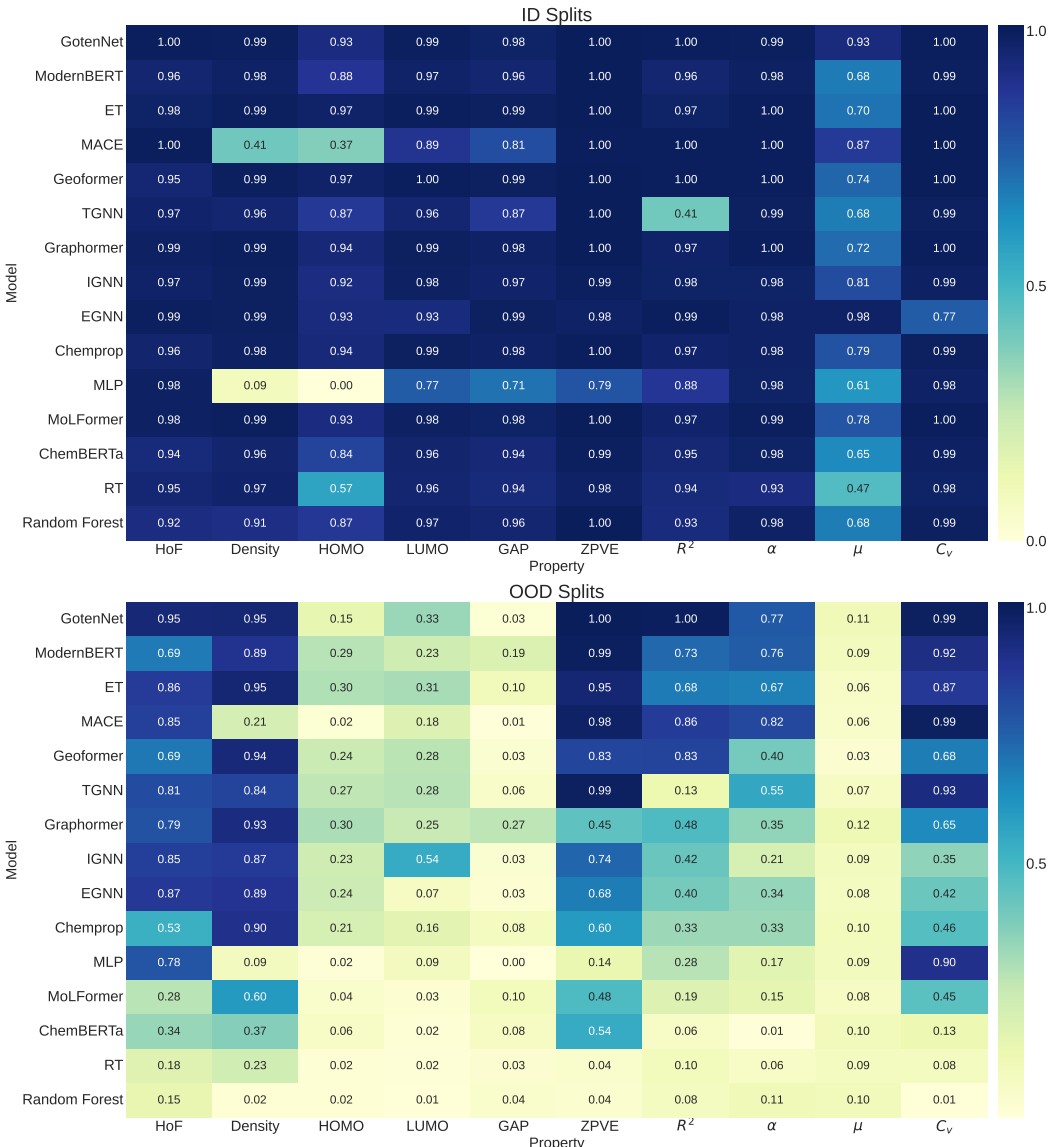

Figure 2: We provide the leaderboard of $R^2$ and binned $R^2$ for ID and OOD models, respectively. State-of-the-art models such as GotenNet do remarkably well on ID tasks, as well as some of the OOD tasks. The graph-based and hybrid models provide the best scores across nearly all tasks for OOD and ID splits. Numerical encoding issues greatly hamper RTs performance and result in large errors. We additionally provide results on using a Llama large language model for OOD property prediction in the Appendix E. All results are averaged across 3 training runs.

## 4.2 Impact of Pretraining

Chemical foundation models are commonly pretrained on datasets of billions of molecules to enable generalization across various molecule design tasks. We benchmark and ablate the pretraining of ChemBERTa and MoLFormer (both masked language modeling (MLM) pretraining) and Regression Transfomer (permutation language modeling (PLM) pretraining) to understand how pretraining impact OOD performance. Notably, the original reports of all three of these foundation models showed that this large-scale language pretraining strategy can achieve SOTA performance on in-distribution molecular property prediction tasks [Ross et al., 2022, Chithrananda et al., 2020], but did not evaluate the OOD performance.

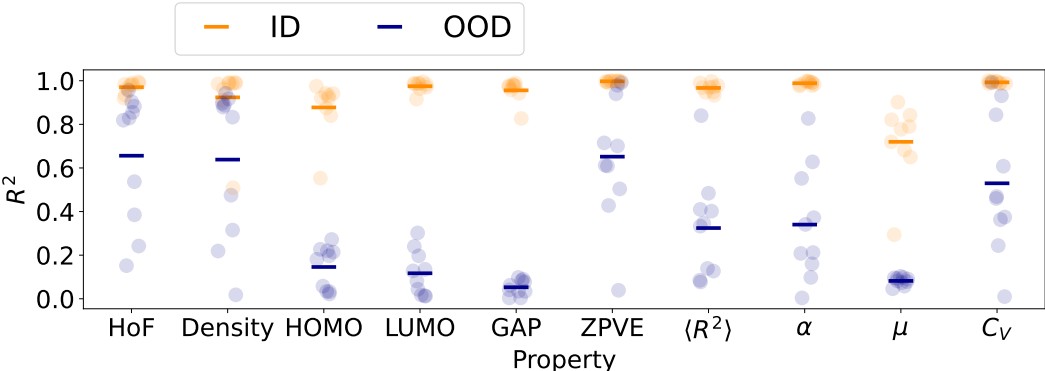

Figure 3: Binned $R^2$ scores for OOD and standard $R^2$ scores for ID on each task for all models. The orange and blue bars indicate the performance averaged across all models for ID and OOD, respectively. Nearly all models have significant discrepancies between ID and OOD performance, but some models can reach ID-level accuracy. We observe that OOD performance is highly task-dependent.

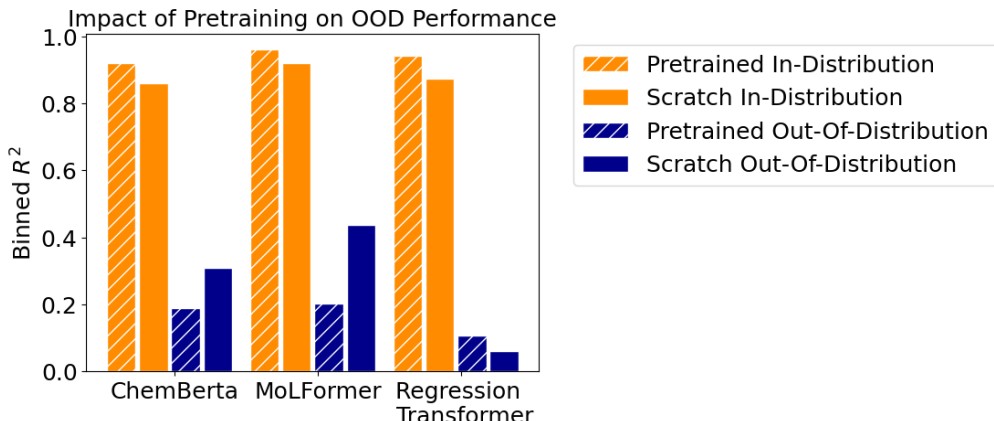

Figure 4: OOD Performance of chemical foundation models (ChemBERTA, MoLFormer and Regression Transformer) with and without pretraining, averaged across all tasks. We find that current pretraining strategies improve ID performance, but not OOD. The task-specific performances are provided in the Appendix (Figure 10).

All three foundation models improve ID performance across the majority of tasks (Figure 10). Averaged across all 10 tasks, the pretrained models show a sizable improvement in ID RMSE due to pretraining (31% for ChemBERTa, 35% for MoLFormer and 12% for Regression Transformer). These results are consistent with the findings in their original reports. For example, the MoLFormer paper found a 29% reduction in mean absolute error ID performance on the QM9 dataset due to MLM pretraining, whereas Regression Transformer reported up to a 52% reduction in RMSE when predicting ID drug-likeness (QED) from optimizing the pretraining objective. A similar ablation study was not performed in the original ChemBERTa paper.

Surprisingly, we find that all three foundation models do not show any significant improvement in OOD performance due to language modeling pretraining (Fig. 4). All three models show a negligible change in average OOD RMSE due to pretraining, and the Binned OOD $R^2$ decreases significantly for both MoLFormer (53%) and ChemBERTa (39%). Although pretraining does provide chemical foundation models with a richer understanding of chemistry, as signified by stronger ID performance, the existing pretraining procedures do not seem to allow for the models to extrapolate well to new chemistries. This result may suggest that the current pretraining tasks used by the foundation models (PLM and MLM) do not convey the relevant chemical information to allow the foundation model to extrapolate well to the downstream OOD property prediction tasks.

To explore if strong OOD generalization can be achieved through alternative pretraining tasks, we also explore pretraining on a supervised property prediction task. First, we perform supervised pretraining of a Chemprop model on the entirety of one of the eight QM9 property datasets. This pretrained model is then finetuned on only the training set of one of the other seven QM9 property dataset (see Fig. 5 with training details in Appendix C.3). This isolates only the property to be OOD, as the model has seen all the molecules in another context. Notably, across all eight QM9 datasets, we see a significant degradation in the OOD performance when the pretraining task dataset is sufficiently uncorrelated to the downstream finetuning task dataset, i.e., when their Pearson correlation coefficient is less than 0.35. Conversely, OOD performance is improved in all cases where the pretraining and finetuning tasks datasets are strongly correlated. This result may explain why the masked language modeling pretraining used in current chemical foundation models resulted in worse OOD performance (Fig. 4).

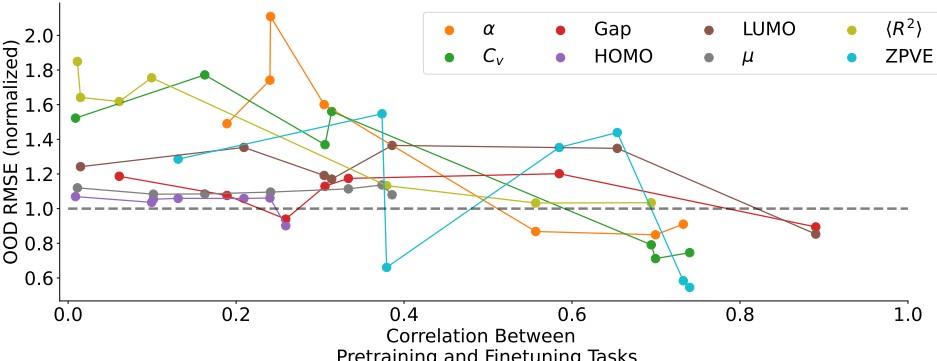

Figure 5: OOD Performance of the Chemprop MPNN model on when pre-trained on different QM9 property datasets. Each line corresponds to the OOD performance on one of the eight QM9 OOD test sets when pre-trained on one of the other seven QM9 properties. The OOD RMSE is plotted against the Pearson correlation coefficient between the pretraining property and the finetuning property in the QM9 dataset. The OOD RMSE is normalized against the Chemprop performance without any pretraining.

## 4.3 Hyperparameter Optimization

The significant gap between the ID and OOD performance in Table 7 may indicate that the models are overfit to the ID molecules, thereby hurting OOD generalization performance. Furthermore, due to the lack of prior OOD benchmarks for molecule property prediction, the default hyperparameters used by these models are also fit to maximize ID performance, which may also negatively impact OOD generalization. In this section, we explore to what extent the OOD performance of models can be improved simply by tuning the model hyperparameters to maximize OOD performance.

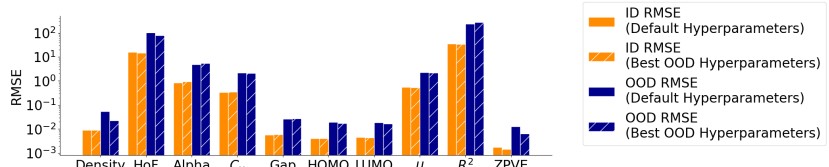

Figure 6: OOD Performance of the Chemprop MPNN model when using default hyperparameters and the best performing OOD hyperparameters. The best OOD hyperparameters are determined according to the minimum OOD test RMSE for each property. Further details are provided in Appendix C.4.

As shown in Fig. 6, we compare the OOD performance of Chemprop when using the default hyperparameters and hyperparameters that have been optimized to maximize OOD performance. Overall, we do not find that hyperparameter optimization can provide meaningful improvements to OOD performance. While we see a noticeable reduction in the OOD RMSE in relatively simple properties such as density(-60%), heat of formation(-23%) and ZPVE(-50%) following hyperparameter tuning, the models are still unable to generalize significantly beyond the training regime.

It may not solve the problem, but for certain properties, hyperparameter optimization with respect to OOD improved over the default model by 60%, without any significant decrease to the ID performance. The results here highlight that OOD performance should be considered as an important evaluation criterion for future model optimization to ensure that models strike a balance between ID performance and OOD generalization.

## 4.4 Representation

| Representation | Split | HoF | Density | HOMO | LUMO | Gap | ZPVE | $\langle R^2 \rangle$ | $\alpha$ | $\mu$ | $C_V$ |
|---|---|---|---|---|---|---|---|---|---|---|---|
| 3D | ID | **11.09** | .0121 | **.0026** | **.0030** | **.0042** | **.0005** | 21.7465 | .3234 | **.3690** | **.1296** |
| | OOD | 21.76 | .0247 | .0152 | .0137 | .0238 | .0031 | 112.7228 | .30890 | 2.2832 | .9457 |
| Graph | ID | 15.68 | **.0092** | .0041 | .0048 | .0058 | 0.0014 | 35.68 | .8305 | .55 | .3341 |
| | OOD | 100.6 | .0551 | .0192 | .0187 | .0267 | .0129 | 234.73 | .4772 | 2.3 | 2.149 |
| SMILES | ID | 22.86 | .0163 | .0068 | .0088 | .0103 | .0046 | 50.297 | 1.444 | .7134 | .4923 |
| | OOD | 99.7253 | .1173 | .0245 | .0267 | .0315 | .0214 | 306.14 | 6.303 | 2.766 | 3.0175 |

Table 1: Averaged RMSE of models on OOD and ID tasks as grouped by input representation. The best performing **ID** and **OOD** models are highlighted in **Black** and **Blue** respectively. The worst performing **ID** and **OOD** models are highlighted in **Orange** and **Red** respectively. The models included in each representation category are explicitly enumerated in Table 3.

In our study, 3D models with equivariant and invariant symmetries significantly outperform the SMILES-based models in nearly all tasks. Furthermore, the 3D GNN models like EGNN and IGNN are significantly more parameter-efficient. As we can see in Table 1, the SMILES-based models, namely the transformer models, perform significantly worse than the 3D and graph models in nearly all tasks. SMILES and graphs are interchangeable representations in that SMILES can be converted into a molecular graph and vice versa. SMILES-based representations present the same atom and topology information present in a graph-like representation, but in a sequence format. This suggests the inductive bias present in the graph-based models improves the model performance over attention-based models, especially for OOD splits. Interestingly, the graph-like models also perform comparably to the transformer-based models if we discount RT. MoLFormer, a SMILES-based based, model has strong ID performance compared to other models as well.

## 4.5 Data Ablation Study

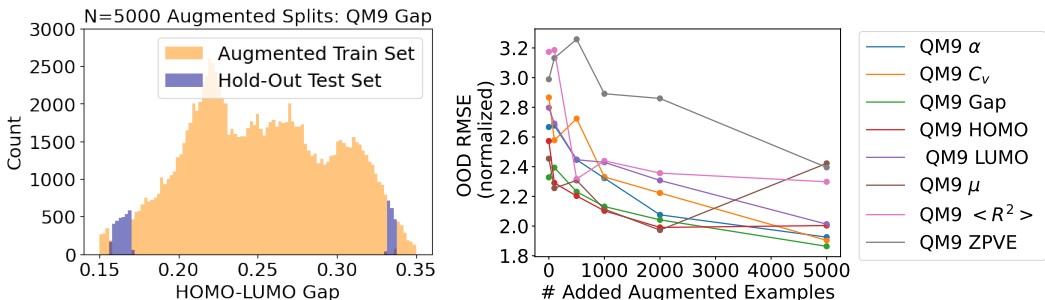

Figure 7: Performance of the Chemprop MPNN model when various amounts of OOD samples are included in the training. For each property, we separate the 10,000 OOD examples into a hold-out test set (N=5000) and various amounts of the remaining 5000 augmented examples are included during training. The OOD test RMSE is normalized against the validation RMSE of the model trained without any additional OOD examples included during training.

Beyond exploring different model architectures and molecular representations, data generation is a common strategy for improving the generalization capabilities of chemical deep learning models. [Merchant et al., 2023, Antoniuk et al., 2025] In this experiment, we seek to explore to what extent adding a relatively small number of molecules in the OOD region can improve model generalization. We emphasize that the feasibility of using molecular generative models to efficiently generate useful OOD molecules is still a significant and unsolved challenge. The throughput at which property data can be acquired, whether through experimental measurements or simulations, is also strongly

property-dependent. The goal of this experiment is not to prescribe a path towards generating OOD molecules, but to better understand the sensitivity of property prediction models to the addition of OOD molecules. We provide a complete discussion of this approach and related prior work in the Appendix C.8. Figure 7 investigates improving OOD property prediction by augmenting the QM9 training set (described in Section 2) with extreme-valued molecules from the QM9 OOD test set. The augmented molecules are selected by sampling N =[0, 100, 500, 1000, 2000, 5000] molecules from the QM9 OOD test sets with properties below and above the 25th and 75th quantiles, respectively.

Across 7 of 8 QM9 tasks, Chemprop's generalization improves with augmented data (Figure 7). The lack of improvement for QM9 dipole moments ($\mu$) likely stems from Chemprop's graph representation lacking 3D electronic structure. Data augmentation consistently yields sizable generalization improvements, even with a small fraction ( 4%) of augmented data. On the other hand, data generation may not be a viable solution in many scenarios. Further improvements may be achievable with more extensive data generation.

## 4.6 ModernBERT for Chemistry

Finally, we highlight a significant improvement in OOD performance with ModernBERT among the NLP-style models tested. While all the NLP model architectures tested don't have any chemistry specific design choices, the improvements proposed in ModernBERT translate to the chemistry domain as well (see Appendix B.3.1). We highlight the task-specific behavior in Figure 8 for OOD performance. ModernBERT performs similarly to other transformer models for the difficult tasks (HOMO, LUMO, Gap, and $\mu$), but improves significantly for the remaining properties. ModernBERT decreases OOD HoF and $C_v$ RMSE by more than 58% and 78%, respectively, over other best-performing transformer models. While not in the scope of our current work, understanding the design choices that result in these improvements can inform design choices for future chemistry foundation model design.

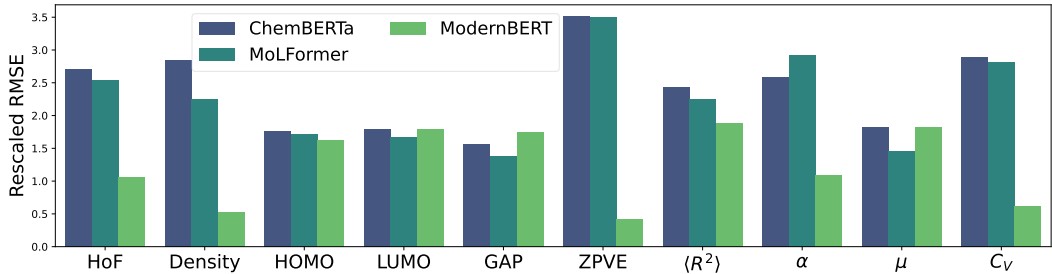

Figure 8: ModernBERT outperforms transformer models for OOD tasks. ModernBERT bridges the gap between transformer and GNN-based models on OOD splits, especially for HoF, Density, and $C_v$. RMSE values are normalized against the mean task-specific OOD RMSE across all models in Table 7.

## 4.7 Statistical Analysis

We conduct additional statistical tests on the observations from our data. First of all, we investigate the observation that ID performance is not necessarily predictive of OOD performance. We fit a Gaussian Process model to predict OOD values using the ID values in Fig. 11. For all tasks, we notice a large variance in the predictions for high ID values, suggesting a high uncertainty in the model's prediction. Furthermore, we also perform distance correlation permutation tests on the ID and OOD values. We see a moderate correlation over all tasks (d=.4678, p=.0010) from the distance correlation permutation test. Our model shows an interesting correlation; models with low ID scores also have low OOD scores, as expected. On the other hand, high ID scores do not guarantee strong OOD performance. Running the test for individual properties, certain properties, such as dipole moment and HOMO-LUMO gap, are of particular interest where we fail to reject the null hypothesis.

Furthermore, we measure the effects of different representations on OOD performance. While we note that the samples are not truly random and may present selection bias, we believe the following tests capture our observation. We categorize models as described in Table 3 into 3 categories:

3D, graph, and transformer models use their OOD values. We perform a Kruskal-Wallis test over these sets and verify there is a statistically significant difference (p=0.016, N=390) within these sets. We then perform a Mann-Whitney U test for each pair with the appropriate null hypothesis. For OOD values over all tasks, we can see that 3D models' OOD values exceed their transformer counterparts(p=1.85e-9), and graph OOD values exceed transformers (p = 4.76e-5). We further analyze the differences for each property individually using the Kruskal-Wallis test (N = 39). We see statistically significant evidence of geometric models outperforming transformer models for a majority of the properties. The full table is presented in Table 8.

## 5 Limitations

BOOM aims to challenge current and future chemical models to learn beyond the training data. The relative scarcity of samples in the QM9 and 10K dataset is a concern, but we believe BOOM can still be of practical use. In practice, practitioners fine-tune models on small datasets, and we believe BOOM can adequately capture that scenario. As we aim for generality across as large a set of chemical models as possible, benchmarking all possible available models is difficult. We select our models to represent those used in practice and hope that researchers benchmark proposed models using BOOM.

## 6 Discussion

Overall, across all 15 tested model architectures, we do not find any model that achieves strong performance on all OOD tasks. As a result, we expect that current property prediction models will struggle to consistently discover molecules with properties that extrapolate beyond known molecules. Nevertheless, given the saturation of the most commonly used chemistry benchmarks, we hope that the results presented here inspire the chemistry community to pursue OOD generalization as the next frontier challenge for further developing molecular property prediction models.

Surprisingly, we found that commonly employed molecular pretraining strategies, such as masked language modeling, often result in a decrease in OOD performance. Our experiments show that developing new pretraining tasks whereby the pretraining task and the downstream property prediction tasks are more closely related results in improved OOD generalization. Fig. 5 consistently shows that OOD performance is only improved by pretraining when the chemical information contained in the pretraining task is related to the downstream property prediction task. Randomly sampling model hyperparameters of the Chemprop GNN was found to improve OOD performance for a few properties, with very little change in ID performance. This result highlights the need to consider OOD generalization when optimizing the model hyperparameters of chemical prediction models.

While high-inductive bias (e.g. graph neural networks) and 3D models perform well on our current tests, scalability remains a significant issue. Small models can provide strong predictive power, but they do not allow for techniques such as in-context learning and test-time compute that may be available to large-scale models. Similarly, 3D models are attractive, but high-quality DFT data is not always available. While 3D molecular data is becoming increasingly more available, it dwarfs in comparison to the billions of molecules used in unsupervised molecular pretraining strategies. In general, as our results with ModernBERT show, transformer-based models can potentially catch up to the small models while enabling greater scalability. Numerical encoding is a concern for LLM-like models and was a significant drawback for RT. Improved post-hoc solutions [Golkar et al., 2023] or modern tokenization techniques [Achiam et al., 2023, Grattafiori et al., 2024a] will be key in the development of LLM-based predictive models.

## 7 Conclusion

We propose BOOM, a methodology to study the OOD performance of AI/ML chemical models and benchmark a plethora of models and techniques. Notably, we do not find any strategy that universally improves OOD performance across all property prediction tasks. Current SOTA property prediction models exhibit poor generalization with a large difference between ID and OOD performance on electronic structure properties such as HOMO and $\mu$. We anticipate that achieving strong OOD generalization on these properties will require larger datasets, in combination with molecular representations that explicitly capture the molecules' electronic structure. We hope future chemistry models can utilize the OOD benchmarks and improve upon current results.

# 8 Acknowledgments

This work was performed under the auspices of the U.S. Department of Energy by Lawrence Livermore National Laboratory under Contract DE-AC52-07NA27344 and LDRD 24-SI-008. We gratefully acknowledge use of the research computing resources [Bloom et al., 2025] of the Empire AI Consortium, Inc, with support from Empire State Development of the State of New York, the Simons Foundation, and the Secunda Family Foundation. This work used Delta at UIUC NCSAthrough allocation from the Advanced Cyberinfrastructure Coordination Ecosystem: Services & Support (ACCESS) [Boerner et al., 2023] program, which is supported by U.S. National Science Foundation grants #2138259, #2138286, #2138307, #2137603, and #2138296.

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

# A Datasets

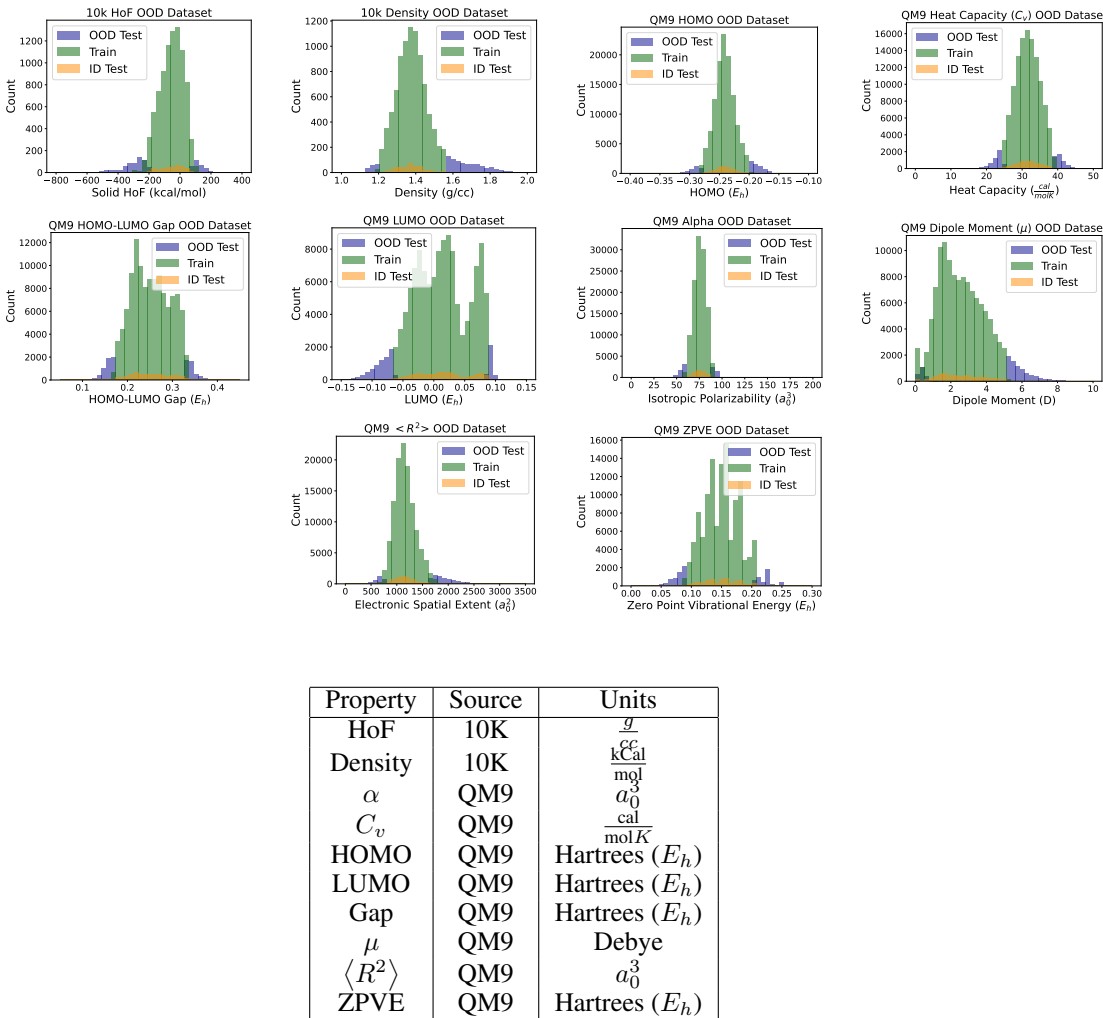

Table 2: Dataset sources and units for all BOOM property datasets.

| Property | Source | Units |
|---|---|---|
| HoF | 10K | $\frac{g}{cc}$ |
| Density | 10K | $\frac{kCal}{mol}$ |
| $\alpha$ | QM9 | $a_0^3$ |
| $C_v$ | QM9 | $\frac{cal}{mol K}$ |
| HOMO | QM9 | Hartrees $(E_h)$ |
| LUMO | QM9 | Hartrees $(E_h)$ |
| Gap | QM9 | Hartrees $(E_h)$ |
| $\mu$ | QM9 | Debye |
| $\langle R^2 \rangle$ | QM9 | $a_0^3$ |
| ZPVE | QM9 | Hartrees $(E_h)$ |

## A.1 Data Split Details

In general, one can define OOD with respect to either the model inputs (holding out a region of chemical space as the OOD test split) or with respect to the model outputs (holding out a range of chemical property values). In this work, we adopt the latter approach of benchmarking the performance of the models to extrapolate to property values not seen in training. Following the OOD definitions outlined by Farquhar et al., we here define OOD as a complement distribution with respect to the targets Farquhar and Gal [2022], Scheirer et al. [2012]. Specifically, given a molecule property dataset of chemical structures and their numerical property values, we create our OOD test set to consist of numerical values on the tail ends of the numerical property distribution (see Figure 1). In this way, our OOD benchmarking is directly aligned with the molecule discovery task in that it allows us to evaluate the consistency of ML models to discover molecules with state-of-the-art properties that extrapolate beyond the training data.

We generate our training, ID, and OOD splits based on the property distribution. For each of the 10 molecular properties, we generate OOD splits by first fitting a kernel density estimator (with Gaussian kernel) to the property values and obtain the probability of a molecule given its property. We select the molecules with the lowest probabilities for the OOD split for that property. This results in selecting the molecules at the tail end of the distribution for typical molecular property distributions

since these molecules will have low densities in property space. Unlike partitioning by cut-off values, this method of splitting allows us to capture low-probability samples for general distributions that aren't necessarily unimodal. For QM9 we take the lowest 10% of the probability scores as predicted by the kernel density estimator for the OOD set. We take the lowest 1000 molecules for the 10K dataset. We then randomly sample molecules from the remaining molecules to generate the ID test set. We sample 10% of the molecules in the case of QM9 and 5% of the molecules for ID test split for 10K data. The remaining molecules are used for training and fine-tuning.

We provide a simple library to gather, process, and use the datasets described above for model training and evaluation. The **boom** package can be installed via pip. The data in SMILES and 3D formats can be obtained through the **boom** package.

Listing 1: Getting Datasets

```
from boom.datasets.SMILESDataset import TrainDensityDataset
from boom.datasets.SMILESDataset import IIDDensityDataset
from boom.datasets.SMILESDataset import OODDensityDataset

training_data = TrainDensityDataset()
id_test_data = IIDDensityDataset()
ood_test_data = OODDensityDataset()
```

The '10K CSD' datasets available are:

- '<split>DensityDataset'
- '<split>HoFDataset'

The 'QM9' datasets available are:

- '<split>QM9_alphaDataset'
- '<split>QM9_cvDataset'
- '<split>QM9_homoDataset'
- '<split>QM9_lumoDataset'
- '<split>QM9_gapDataset'
- '<split>QM9_muDataset'
- '<split>QM9_u298Dataset'
- '<split>QM9_zpveDataset'

Where, '<split>' is one of **train**, **id**, or **ood**.

## B   Model Details

### B.1   Model Summary

### B.2   RDKit Featurizer

The RDKit Featurizer, as implemented in the Deepchem package,[Ramsundar et al., 2019] consists of 125 chemically-informed features (such as molecular weight and number of valence electrons), as well as 86 features describing the fraction of atoms that belong to notable functional groups such as alcohols or amines.

### B.3   Transformers

Transformers, including large language models (LLMs), have revolutionized language modeling and vision tasks and have gained popularity in scientific regimes. We choose four representative models

| Model Name | Architecture | Molecule Representation | Symmetry | # of parameters |
|---|---|---|---|---|
| Random Forest | Random Forest | RDKit Molecular Descriptors | N/A | N/A |
| Multilayer Perceptron | Multilayer Perceptron | RDKit Molecular Descriptors | N/A | 153k |
| ChemBERTa | Transformer | SMILES | N/A | 83M |
| MolFormer | Transformer | SMILES | N/A | 48M |
| RT | Transformer | SMILES | N/A | 27M |
| ModernBERT | Transformer | SMILES | N/A | 111M |
| Chemprop | GNN | {Atom, Bond} | permutation | 200k |
| TGNN | GNN | {Atom, Bond} | permutation | 200k |
| IGNN | GNN | {Atom, Bond, Pair-wise Distances } | E(3)-invariant + permutation | 217K |
| EGNN | GNN | {Atom, Bond, Atom Positions} | E(3)-equivariant + permutation | 217K |
| MACE | GNN | {Atom, Bond, Pair-wise Distances} | E(3)-equivariant + permutation | 3.9M |
| GotenNet | GNN | {Atom, Bond, Pair-wise Distances} | E(3)-equivariant + permutation | 6M |
| Graphormer-3D | Hybrid | {Atom, Bond, Pair-wise Distances } | E(3)-invariant + permutation | 47.1M |
| GeoFormer | Hybrid | {Atom, Bond, Pair-wise Distances } | E(3)-invariant + permutation | 47.1M |
| ET | Hybrid | {Atom, Bond, Atom Positions} | E(3)-equivariant + permutation | 6.8M |

Table 3: Summary of the model architectures included in the BOOM benchmark, along with their model architecture, molecular representation, model symmetry, and total number of model parameters.

| Model | Device | Runtimes (10k) [seconds/epoch] | Runtimes (QM9) [seconds/epoch] |
|---|---|---|---|
| Random Forest | 2.4 Ghz 8-core Intel Core i9 | 0.5 | 6 |
| RT | V100 | 632 | 10275 |
| ChemBERTa | L40 | 13 | 165 |
| MoLFormer | H100 | 14 | 72 |
| Chemprop | H100 | 10 | 23 |
| EGNN | L40 | 10 | 115 |
| IGNN | L40 | 10 | 115 |
| TGNN | L40 | 10 | 115 |
| MACE | H100 | 70 | 165 |
| Graphormer | AMD MI300A | 6 | 19 |
| ET | L40 | 10 | 75 |
| Geoformer | H100 | 60 | 230 |
| GotenNet | A100 | 20 | 120 |
| ModernBERT | L40 | 15 | 165 |
| MLP | H100 | 0.009 | 0.5 |

Table 4: Runtimes for all models used in BOOM on the 10k and QM9 datasets.

to cover the major archetypes of transformer models: MoLFormer [Ross et al., 2022], ChemBERTa [Chithrananda et al., 2020], Regression Transformer [Born and Manica, 2023], and ModernBERT [Warner et al., 2024].

We choose three representative models to cover the major archetypes of transformer models. MoL-Former [Ross et al., 2022] is an encoder-decoder model with a T5 [Raffel et al., 2020] backbone originally trained on PubChem. ChemBERTa Chithrananda et al. [2020] is an encoder-only model with a BERT [Devlin et al., 2019] backbone trained on PubChem. Finally, we also use Regression Transformer [Born and Manica, 2023], an XLNet-based [Yang et al., 2019a] model that is capable of both masked language modeling as well as autoregressive generation.

### B.3.1 ModernBERT

We also evaluate ModernBERT, a state-of-the-art (SOTA) encoder-only model with architectural improvements such as rotary positional embeddings [Su et al., 2024], pre-normalization, and GeGLU activation layers [Shazeer, 2020]. Along with different architectures, we also investigate the effects of different pre-training and tokenization schemes in our experiments. The training details are presented in Appendix C.2.

### B.4 GNNs

GNNs are neural networks designed for learning on graph-structured data. Molecules and materials are represented as graphs of atoms and bonds, with 3D Euclidean space providing a natural molecular representation. As a result, message-passing neural networks (MPNNs) serve as the de facto backbone for deep learning-based molecular property prediction [Schütt et al., 2018, Qiao et al., 2020]. Extensive work compares various GNN algorithms for this task. Instead of focusing on specific GNN variants, we examine the significance of architectural differences in our OOD task, emphasizing the relational inductive bias of molecular graphs and symmetries.

3D information and symmetries are fundamental to physical laws governing molecular behavior. Chemprop [Heid et al., 2023] serves as the baseline for a standard topological (2D) GNN. Additionally, we use three GNNs with topological, E(3) invariant, and E(3) equivariant learned models based on EGNN [Satorras et al., 2021]. MACE is a popular E(3) equivariant GNN, which uses pair-wise distances for message passing and construction [Batatia et al., 2023, Kovács et al., 2023]. Unlike EGNN, MACE also takes into account higher-order interactions, potentially allowing for greater expressivity. To explore the effects of these symmetries, we test these five GNNs for our OOD tasks.

Symmetries are inherent to the physical laws that dictate molecular properties. Algebraically, they are represented as groups, where each element corresponds to a transformation. For non-chiral molecules, the E(3) group, encompassing rotations, translations, and reflections, is key. Chiral molecules require the SE(3) subgroup, which excludes reflection. Since molecular properties remain invariant under these transformations, learned structure-to-property functions should obey the same symmetries.

GNNs naturally encode these symmetries. MPNNs enforce permutation-invariant message aggregation, making models permutation-invariant. Geometric deep learning models can extend this by enabling molecular representations in 3D space, ensuring networks are invariant or equivariant to geometric transformations. Invariance implies properties remain unchanged after transformation, while equivariance means vector properties transform consistently with applied transformations. Here, we provide rigorous definitions.

For completeness, we reproduce the GNN formulation from [Satorras et al., 2021]. For a given GNN with node features $h_i^{(l)}$ are the features of the $i$-th node for $l$-th layer. $b_{ij}$ are the edge-features between two connected nodes $i$ and $j$ such that $j \in \mathcal{N}_i$. The neighborhood $\mathcal{N}_i$ is the set of nodes connected to node $i$. $W^{(l)}$ is a learnable projection matrix of layer $l$.

*Topological GNN:*

$$h_i^{(l+1)} = h_i^{(l)} W^{(l)} + \sum_{j \in \mathcal{N}_i} \theta(b_{ij}, h_i^{(l)}, h_j^{(l)}) \tag{3}$$

Where $\theta(\cdot)$ is a learnable function of the bond and node features, shared between all node pairs.

*Invariant GNN:*

$$h_i^{(l+1)} = h_i^{(l)} W^{(l)} + \sum_{j \in \mathcal{N}_i} \theta(b_{ij}, h_i^{(l)}, h_j^{(l)}) + \sum_{j \neq i} \phi(r_{ij}, h_i^{(l)}, h_j^{(l)}) \tag{4}$$

Where, $r_{ij} = ||x_i - x_j||^2$ is the inter-atomic distance between atoms $i$ and $j$. $\phi(\cdot)$ is a learnable function of the interatomic distances and node features, shared between all node pairs.

*Equivariant GNN:*

$$h_i^{(l+1)} = h_i^{(l)} W^{(l)} + \sum_{j \in \mathcal{N}_i} \theta(b_{ij}, h_i^{(l)}, h_j^{(l)}) + \sum_{j \neq i} \phi(r_{ij}^{(l)}, h_i^{(l)}, h_j^{(l)}) \tag{5}$$

$$x^{(l+1)} = x^{(l)} + \sum_{j \neq i} \left( \frac{x_i^{(l)} - x_i^{(l)}}{r_{ij}^{(l)} + \xi} \right) \psi(r_{ij}^{(l)}, h_i^{(l)}, h_j^{(l)}) \tag{6}$$

Where, $r_{ij}^{(l)} = ||x_i^{(l)} - x_j^{(l)}||^2$ is the inter-atomic distance between atoms $i$ and $j$ at the $l$-th layer. $\xi$ is a small constant for numerical stability. $\psi(\cdot)$ is a learnable function of the inter-atomic distances and node features, shared between all node pairs.

As we can see Eq. 5 is equivalent to Eq. 4 but with a per-layer coordinate update. Furthermore, Eq. 4 is equivalent to Eq. 3 but with an additional term dependent on the pairwise distances $r_{ij}$.

### B.4.1 Readout Function

The readout function, $\mathcal{R}$ of a GNN aggregates the node-level information on the graph and combines them to get a graph-level output. The readout function can be any permutation invariant function such that, $\mathcal{R} : \mathbb{R}^{|\mathcal{V} \times F|} \to \mathbb{R}^K$, where $F$ is the per-vertex feature dimension, and $K$ is the output dimension ($K = 1$ in the case of regression). The flexibility in the readout function can be used

to provide target-specific inductive bias such as using a summing over the vertices for extensive properties while taking the mean output for the vertices for intensive properties.

MACE and ET use modified readout functions for some properties, such as $\mu$, while we are using the unmodified readout function. We have not had success modifying the readout function as described in their publication, but we are working with the authors to replicate their results. We plan on investigating this further.

### B.5  Hybrid Architectures

Recently, we have seen an emergence of hybrid architectures that combine the inductive properties of GNNs and with the flexibility of the attention mechanism in Transformers. Graphormer [Ying et al., 2021] is a GNN-Transformer model that incorporates a graph-specific encoding mechanism to the input perform attention over structured data rather than sequences. Furthermore, Graphormer adds a bias term to the Query-Key product matrix to bias the attention to include bond information. We evaluate Graphormer-3D, a variant of Graphormer that incorporates inter-atomic distances to introduce 3D information to the attention mechanism. Finally, we also evaluate Equivariant Transformer (ET) [Thölke and De Fabritiis, 2022], a 3D encoder-only transformer model that incorporates E(3) equivariance. Rather than inter-atomic distances, ET operates directly on 3D atomic coordinates.

## C  Training Details

### C.1  General Training

Across all models, we generally hold out $10\%$ of the training data for hyperparameter selection. As we had multiple different types of models, we started with publicly available settings for the starting hyperparameters (as noted below), but also performed hyperparameter sweeps (with grid search) on the non-architectural components, such as learning rate and training steps. We do not update architectural details to match the use case of practitioners using off-the-shelf models. The GNN ablation uses the architecture detailed here,Satorras et al. [2021] and training instructions listed in the Appendix of that work. The baseline models (Random Forest and MLP), use model hyperparameters previously reported.Yang et al. [2019b], Wu et al. [2018]

### C.2  Transformer Fine-tuning Details

ChemBERTa and MoLFormer models are pre-trained with a masked language modeling (MLM) task and the Regression Transformer is pretrained with a permutation language modeling (PLM) task. During MLM pretraining, a predetermined fraction of the SMILES string of the molecule is masked and then predicted by the model. The Regression Transformer foundation model uses a PLM pretraining task, which seeks to autoregressively predict masked tokens from a permuted sequence of both SMILES and property tokens.

For Regression Transformer and ChemBERTa, the models without pretraining are initialized with random weights, whereas the MoLFormer model without pretraining is loaded directly from the provided checkpoint saved at the beginning (0th iteration) of pretraining. For all three models, the pretrained models are initialized from the provided model checkpoints, before finetuning on each of the 10 downstream OOD tasks. Both the pretrained and scratch models are fine-tuned according to the same learning schedule hyperparameters.

### C.3  Chemprop Pretraining Details

For the experiments highlighted in Figure 5, we first train all model weights of the Chemprop model (v1.4.0) for 30 epochs on the entirety of one of the eight QM9 property datasets (133,885 training examples). Then, this model is finetuned for an additional 30 epochs on only the train split (see 2) of one of the other seven QM9 property datasets. During this finetuning step, the model parameters of the message-passing neural network portion of Chemprop are frozen. All other model hyperparameters are the Chemprop defaults and are provided in the Github repo.

## C.4 Chemprop Hyperparameter Optimization

To understand to what extent hyperparameter optimization can affect OOD performance, we first train the Chemprop model (v1.4.0) with all default hyperparameters on all 10 BOOM datasets. Then, we train Chemprop on each of the 10 BOOM datasets with 50 independent, random choices of model hyperparameters (i.e. with 50 random seeds). The tuned model hyperparameters are the message passing depth, sampled from between 2-6 layers, the fraction of dropout in the neural network sampled between 0-0.40 with an increment of 0.05, the number of feed-forward layers sampled from between 1-3 layers, and the size of the hidden layers, sampled between 300-2400 with an increment of 100.

## C.5    Additional Plots

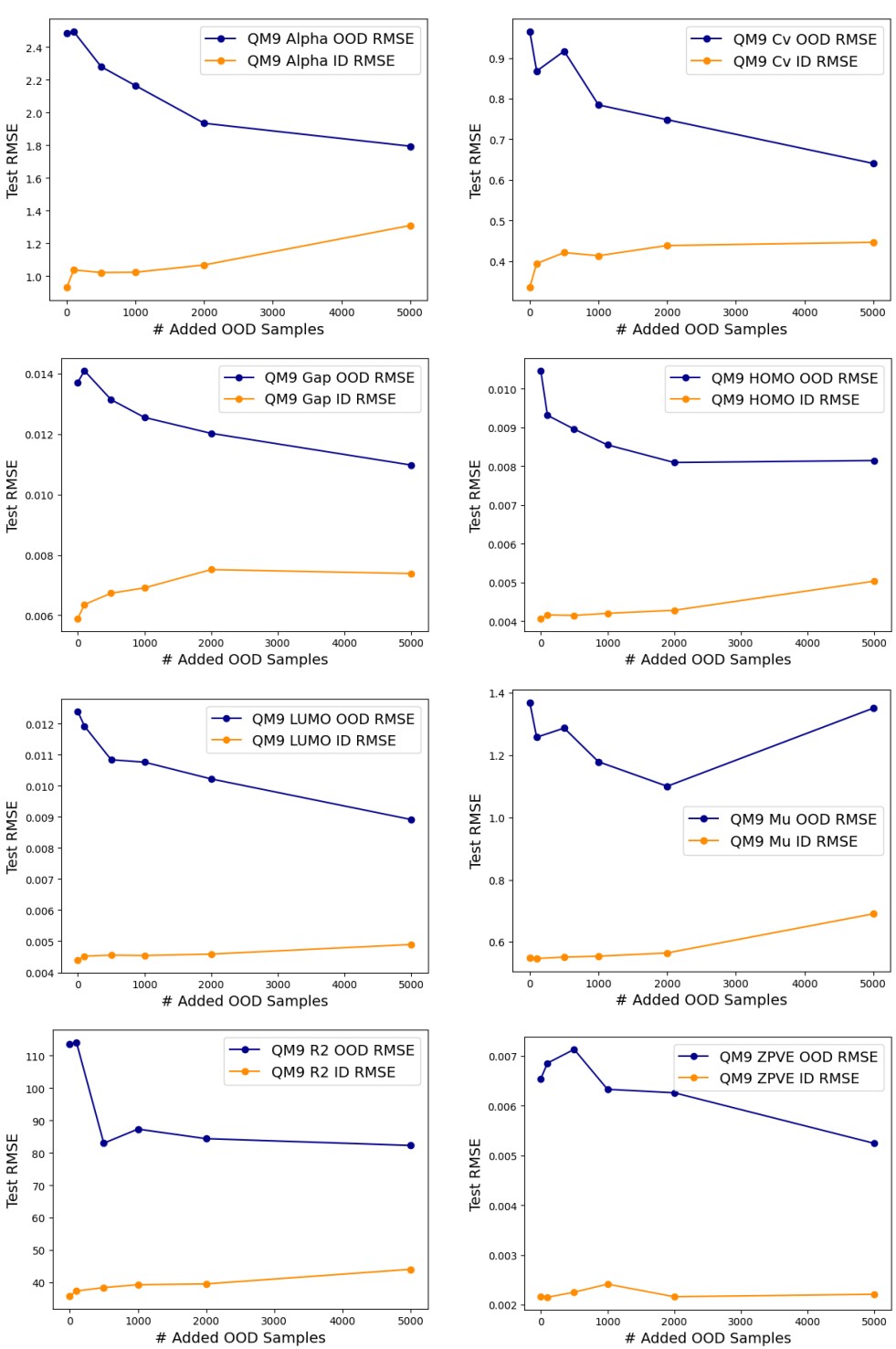

Figure 9: Chemprop MPNN Performance with Data Augmentation and QM9 OOD tasks

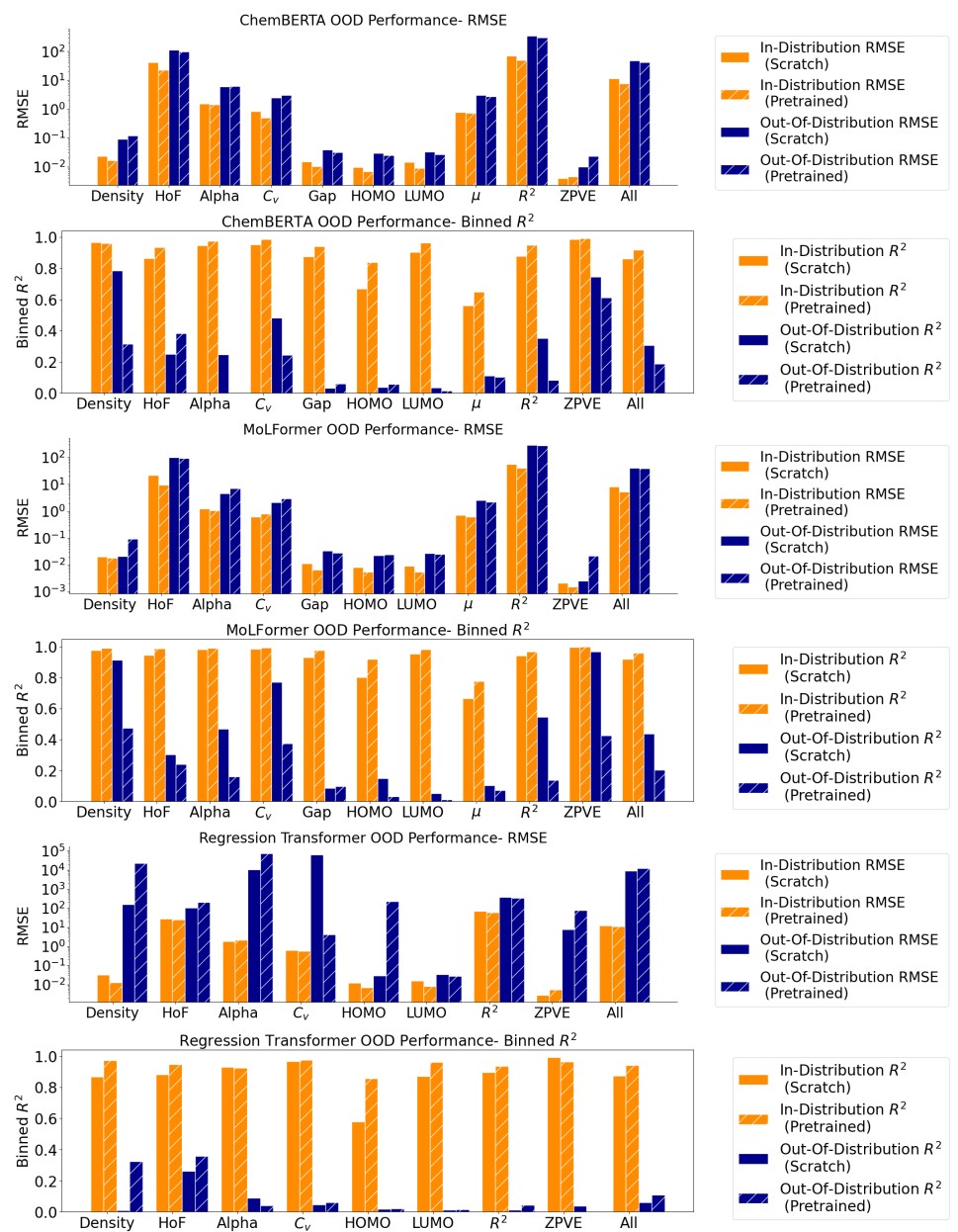

Figure 10: OOD Performance of chemical foundation models (ChemBERTA, MoLFormer and Regression Transformer) with and without pretraining.The performance of Regression Transformer on the QM9 dipole moment and HOMO-LUMO Gap properties are omitted due to the inability of the scratch Regression Transformer model to converge on these properties.

| Model | Type | Split | HoF | Density | HOMO | LUMO | GAP | ZPVE | $<R^2>$ | $\alpha$ | $\mu$ | $C_v$ |
|---|---|---|---|---|---|---|---|---|---|---|---|---|
| ChemBERTa | Transformer | ID | 0.937 | 0.963 | 0.838 | 0.963 | 0.944 | 0.992 | 0.948 | 0.977 | 0.649 | 0.985 |
| | | OOD | 0.339 | 0.372 | 0.059 | 0.021 | 0.076 | 0.544 | 0.063 | 0.009 | 0.104 | 0.126 |
| Chemprop | Explicit-Bonds | ID | 0.963 | 0.985 | 0.940 | 0.987 | 0.980 | 0.996 | 0.967 | 0.981 | 0.792 | 0.990 |
| | | OOD | 0.528 | 0.901 | 0.206 | 0.161 | 0.078 | 0.597 | 0.331 | 0.332 | 0.096 | 0.463 |
| EGNN | 3D | ID | 0.992 | 0.988 | 0.929 | 0.933 | 0.988 | 0.981 | 0.994 | 0.980 | **0.979** | 0.765 |
| | | OOD | 0.869 | 0.886 | 0.238 | 0.074 | 0.028 | 0.681 | 0.400 | 0.344 | 0.078 | 0.424 |
| ET | 3D | ID | 0.977 | **0.994** | **0.974** | 0.994 | **0.989** | 1.000 | 0.975 | 0.997 | 0.702 | **0.999** |
| | | OOD | 0.861 | 0.953 | 0.298 | 0.313 | 0.103 | 0.684 | 0.684 | 0.673 | 0.059 | 0.870 |
| Geoformer | 3D | ID | 0.954 | 0.991 | 0.973 | **0.995** | **0.989** | 1.000 | 0.997 | 0.997 | 0.740 | **0.999** |
| | | OOD | 0.695 | 0.941 | 0.243 | 0.276 | 0.035 | 0.831 | 0.835 | 0.401 | 0.034 | 0.683 |
| GotenNet | 3D | ID | **0.996** | 0.992 | 0.926 | 0.993 | 0.978 | **1.000** | **0.997** | 0.994 | 0.929 | 0.998 |
| | | OOD | 0.946 | 0.947 | 0.151 | 0.335 | 0.026 | 0.999 | 0.998 | 0.772 | 0.111 | 0.990 |
| Graphormer | 3D | ID | 0.985 | 0.991 | 0.944 | 0.989 | 0.982 | **1.000** | 0.971 | 0.995 | 0.723 | 0.997 |
| | | OOD | 0.790 | 0.926 | 0.304 | 0.249 | 0.272 | 0.445 | 0.482 | 0.351 | 0.117 | 0.651 |
| IGNN | 3D | ID | 0.973 | 0.987 | 0.922 | 0.985 | 0.970 | 0.994 | 0.977 | 0.976 | 0.812 | 0.986 |
| | | OOD | 0.852 | 0.866 | 0.229 | 0.544 | 0.035 | 0.737 | 0.424 | 0.207 | 0.090 | 0.353 |
| MACE | 3D | ID | 0.995 | 0.412 | 0.373 | 0.889 | 0.807 | 0.996 | **0.997** | **0.998** | 0.867 | 0.998 |
| | | OOD | 0.846 | 0.212 | 0.025 | 0.184 | 0.008 | 0.979 | 0.858 | 0.823 | 0.058 | 0.990 |
| MLP | No Bias | ID | 0.983 | 0.091 | 0.000 | 0.765 | 0.708 | 0.785 | 0.880 | 0.976 | 0.611 | 0.982 |
| | | OOD | 0.784 | 0.089 | 0.017 | 0.086 | 0.003 | 0.142 | 0.283 | 0.166 | 0.093 | 0.903 |
| MoLFormer | Transformer | ID | 0.984 | 0.992 | 0.927 | 0.985 | 0.978 | 0.999 | 0.970 | 0.992 | 0.782 | 0.995 |
| | | OOD | 0.281 | 0.604 | 0.044 | 0.028 | 0.097 | 0.480 | 0.187 | 0.153 | 0.076 | 0.453 |
| ModernBERT | Transformer | ID | 0.955 | 0.980 | 0.880 | 0.972 | 0.957 | 0.999 | 0.957 | 0.984 | 0.685 | 0.991 |
| | | OOD | 0.691 | 0.893 | 0.285 | 0.230 | 0.190 | 0.992 | 0.731 | 0.764 | 0.092 | 0.918 |
| RT | Transformer | ID | 0.954 | 0.969 | 0.568 | 0.958 | 0.938 | 0.981 | 0.938 | 0.926 | 0.472 | 0.977 |
| | | OOD | 0.180 | 0.226 | 0.020 | 0.017 | 0.032 | 0.044 | 0.096 | 0.059 | 0.085 | 0.084 |
| Random Forest | No Bias | ID | 0.921 | 0.912 | 0.874 | 0.969 | 0.957 | 0.999 | 0.933 | 0.979 | 0.679 | 0.988 |
| | | OOD | 0.147 | 0.021 | 0.023 | 0.013 | 0.043 | 0.041 | 0.084 | 0.108 | 0.098 | 0.011 |
| TGNN | Explicit-Bonds | ID | 0.970 | 0.961 | 0.872 | 0.963 | 0.869 | 0.998 | 0.407 | 0.986 | 0.681 | 0.989 |
| | | OOD | 0.814 | 0.841 | 0.268 | 0.284 | 0.057 | 0.990 | 0.131 | 0.554 | 0.074 | 0.926 |

Table 5: Mean Batched $R^2$ scores of all models on OOD and ID tasks. Best performing **ID** and **OOD** models are highlighted in **Black** and **Blue** respectively. The worst performing **ID** and **OOD** models are highlighted in **Orange** and **Red** respectively. The graph-based and hybrid models provide the best scores across nearly all tasks for OOD and ID splits.

| Model | Type | Split | HoF | Density | HOMO | LUMO | GAP | ZPVE | $<R^2>$ | $\alpha$ | $\mu$ | $C_v$ |
|---|---|---|---|---|---|---|---|---|---|---|---|---|
| ChemBERTa | Transformer | ID | 0.002 | 0.005 | 0.002 | 0.001 | 0.002 | 0.003 | 0.001 | 0.001 | 0.002 | 0.002 |
| | | OOD | 0.040 | 0.050 | 0.013 | 0.004 | 0.014 | 0.062 | 0.023 | 0.007 | 0.003 | 0.103 |
| Chemprop | Explicit-Bonds | ID | 0.001 | 0.001 | 0.001 | 0.001 | 0.001 | 0.001 | 0.001 | 0.002 | 0.002 | 0.001 |
| | | OOD | 0.011 | 0.005 | 0.008 | 0.033 | 0.016 | 0.059 | 0.007 | 0.016 | 0.000 | 0.027 |
| EGNN | 3D | ID | 0.003 | 0.003 | 0.005 | 0.009 | 0.003 | 0.006 | 0.002 | 0.012 | 0.004 | 0.025 |
| | | OOD | 0.037 | 0.005 | 0.014 | 0.009 | 0.005 | 0.033 | 0.007 | 0.128 | 0.001 | 0.054 |
| ET | 3D | ID | 0.007 | 0.002 | 0.001 | 0.001 | 0.000 | 0.000 | 0.025 | 0.002 | 0.354 | 0.000 |
| | | OOD | 0.037 | 0.012 | 0.022 | 0.011 | 0.011 | 0.010 | 0.173 | 0.039 | 0.011 | 0.023 |
| Geoformer | 3D | ID | 0.005 | 0.001 | 0.001 | 0.000 | 0.000 | 0.000 | 0.001 | 0.000 | 0.010 | 0.000 |
| | | OOD | 0.101 | 0.003 | 0.021 | 0.025 | 0.040 | 0.012 | 0.074 | 0.059 | 0.003 | 0.018 |
| GotenNet | 3D | ID | 0.001 | 0.000 | 0.003 | 0.001 | 0.001 | 0.000 | 0.000 | 0.003 | 0.005 | 0.000 |
| | | OOD | 0.011 | 0.005 | 0.019 | 0.031 | 0.014 | 0.000 | 0.000 | 0.011 | 0.008 | 0.001 |
| Graphormer | 3D | ID | 0.004 | 0.003 | 0.002 | 0.001 | 0.001 | 0.001 | 0.001 | 0.001 | 0.003 | 0.000 |
| | | OOD | 0.053 | 0.009 | 0.107 | 0.180 | 0.164 | 0.155 | 0.062 | 0.020 | 0.024 | 0.053 |
| IGNN | 3D | ID | 0.006 | 0.001 | 0.004 | 0.001 | 0.002 | 0.007 | 0.003 | 0.012 | 0.016 | 0.006 |
| | | OOD | 0.005 | 0.019 | 0.015 | 0.354 | 0.001 | 0.024 | 0.182 | 0.007 | 0.003 | 0.027 |
| MACE | 3D | ID | 0.000 | 0.084 | 0.156 | 0.024 | 0.017 | 0.000 | 0.000 | 0.000 | 0.049 | 0.001 |
| | | OOD | 0.145 | 0.035 | 0.007 | 0.012 | 0.004 | 0.000 | 0.083 | 0.004 | 0.035 | 0.002 |
| MLP | No Bias | ID | 0.002 | 0.019 | 0.000 | 0.101 | 0.035 | 0.098 | 0.003 | 0.001 | 0.011 | 0.004 |
| | | OOD | 0.009 | 0.064 | 0.009 | 0.007 | 0.002 | 0.104 | 0.027 | 0.015 | 0.005 | 0.007 |
| MoLFormer | Transformer | ID | 0.003 | 0.001 | 0.005 | 0.002 | 0.001 | 0.000 | 0.001 | 0.001 | 0.005 | 0.001 |
| | | OOD | 0.036 | 0.112 | 0.017 | 0.013 | 0.001 | 0.049 | 0.056 | 0.016 | 0.002 | 0.100 |
| ModernBERT | Transformer | ID | 0.012 | 0.003 | 0.029 | 0.006 | 0.012 | 0.000 | 0.009 | 0.006 | 0.030 | 0.002 |
| | | OOD | 0.061 | 0.005 | 0.058 | 0.076 | 0.121 | 0.002 | 0.000 | 0.190 | 0.015 | 0.011 |
| RT | Transformer | ID | 0.012 | 0.002 | 0.491 | 0.002 | 0.013 | 0.016 | 0.005 | 0.018 | 0.009 | 0.002 |
| | | OOD | 0.157 | 0.090 | 0.003 | 0.015 | 0.011 | 0.036 | 0.049 | 0.047 | 0.015 | 0.070 |
| Random Forest | No Bias | ID | 0.001 | 0.002 | 0.001 | 0.001 | 0.001 | 0.001 | 0.000 | 0.001 | 0.003 | 0.001 |
| | | OOD | 0.006 | 0.003 | 0.001 | 0.001 | 0.003 | 0.003 | 0.006 | 0.009 | 0.001 | 0.002 |
| TGNN | Explicit-Bonds | ID | 0.001 | 0.003 | 0.020 | 0.014 | 0.043 | 0.001 | 0.453 | 0.001 | 0.029 | 0.001 |
| | | OOD | 0.007 | 0.007 | 0.047 | 0.041 | 0.049 | 0.002 | 0.006 | 0.009 | 0.007 | 0.005 |

Table 6: Standard deviation of Batched $R^2$ scores of all models on OOD and ID tasks.

## C.6 Tabulated results

| Model | Representation | Split | HoF | Density | HOMO | LUMO | GAP | ZPVE | $\langle R^2 \rangle$ | $\alpha$ | $\mu$ | $C_v$ |
|---|---|---|---|---|---|---|---|---|---|---|---|---|
| Random Forest | SMILES | ID | 24.43 | 0.0248 | 0.0061 | 0.0071 | 0.0085 | 0.00113 | 52.3 | 0.940 | 0.674 | 0.375 |
| | | OOD | 139.89 | 0.1815 | 0.0304 | 0.0372 | 0.0371 | 0.02303 | 363.0 | 8.470 | **2.899** | 3.362 |
| MLP | SMILES | ID | 13.66 | 0.0532 | 0.0094 | 0.0091 | 0.0130 | **0.00560** | 49.9 | 0.817 | 0.696 | 0.384 |
| | | OOD | 38.43 | 0.0941 | 0.0247 | 0.0201 | **0.0468** | 0.01370 | 470.9 | 6.859 | 2.389 | 0.593 |
| RT | SMILES | ID | 22.23 | 0.0163 | 0.0090 | 0.0102 | 0.0133 | 0.00289 | 68.2 | **2.264** | **1.104** | 0.654 |
| | | OOD | **2428764** | **7558.9** | **539.74** | **584.88** | 0.0339 | **27.6906** | **25458** | **69968** | 2.719 | **11435** |
| ChemBERTa | SMILES | ID | 22.72 | 0.0154 | 0.0070 | 0.0093 | 0.0104 | 0.00390 | 51.7 | 1.254 | 0.713 | 0.489 |
| | | OOD | 100.86 | 0.1195 | 0.0244 | 0.0256 | 0.0309 | 0.02253 | 302.6 | 6.328 | 2.723 | 2.765 |
| MoLFormer | SMILES | ID | 10.94 | 0.0273 | 0.0050 | 0.0052 | 0.0064 | 0.00106 | 40.1 | 1.047 | 0.602 | **0.785** |
| | | OOD | 93.6 | 0.0770 | 0.0236 | 0.0256 | 0.0275 | 0.01990 | 314.1 | 7.356 | 2.232 | 3.667 |
| Chemprop | Graph | ID | 15.43 | 0.0092 | 0.0041 | 0.0046 | 0.0058 | 0.00188 | 35.8 | 0.866 | 0.545 | 0.340 |
| | | OOD | 99.72 | 0.0347 | 0.0189 | 0.0179 | 0.0269 | 0.01277 | 233.7 | 4.850 | 2.304 | 2.118 |
| EGNN | 3D | ID | 10.07 | 0.0077 | 0.0048 | 0.0052 | 0.0069 | 0.00103 | 19.6 | 0.566 | 0.481 | 0.267 |
| | | OOD | 19.19 | **0.0279** | 0.0212 | 0.0236 | 0.0312 | 0.00583 | 181.3 | 5.659 | 2.446 | 2.079 |
| IGNN | 3D | ID | 14.68 | 0.0084 | 0.0050 | 0.0053 | 0.0070 | 0.00173 | 77.5 | 0.903 | 0.519 | 0.405 |
| | | OOD | 23.35 | 0.0281 | 0.0818 | 0.0194 | 0.0297 | 0.00677 | 128.6 | 5.611 | 2.501 | 2.212 |
| TGNN | Graph | ID | 14.46 | 0.0258 | 0.0057 | 0.0072 | 0.0178 | 0.00167 | **211.6** | 0.751 | 0.673 | 0.377 |
| | | OOD | 29.20 | 0.0331 | 0.0184 | 0.0190 | 0.0424 | 0.00260 | 625.5 | 2.787 | 2.524 | 0.627 |
| MACE | 3D | ID | 5.56 | **0.0617** | 0.0150 | **0.0135** | **0.0181** | 0.00180 | 9.8 | **0.322** | 0.430 | 0.134 |
| | | OOD | 38.86 | 0.0670 | 0.0339 | 0.0247 | 0.0409 | 0.00210 | 68.3 | **1.543** | 2.228 | **0.229** |
| GotenNet | 3D | ID | **5.44** | 0.0070 | 0.0052 | 0.0039 | 0.0088 | 0.00043 | 11.8 | 0.553 | **0.319** | 0.197 |
| | | OOD | **15.54** | 0.0360 | **0.0126** | **0.0118** | **0.0229** | **0.00053** | **16.7** | 1.825 | **2.173** | 0.302 |
| Graphormer | 3D | ID | 9.64 | **0.0068** | 0.0040 | 0.0042 | 0.0055 | **0.00024** | 33.4 | 0.431 | 0.626 | 0.180 |
| | | OOD | 31.62 | 0.0770 | 0.0236 | 0.0256 | 0.0275 | 0.01990 | 314.1 | 7.356 | 2.232 | 3.667 |
| ET | 3D | ID | **29.97** | 0.0081 | **0.0027** | 0.0031 | **0.0043** | 0.00057 | 28.2 | 0.490 | 0.368 | 0.160 |
| | | OOD | 52.50 | 0.0479 | 0.0220 | 0.0236 | 0.0271 | 0.01710 | 298.0 | 6.568 | 2.257 | 3.405 |
| Geoformer | 3D | ID | 17.77 | 0.0071 | **0.0027** | 0.0028 | 0.0046 | 0.00030 | 10.9 | 0.326 | 0.847 | **0.124** |
| | | OOD | 43.32 | 0.0366 | 0.0157 | 0.0186 | 0.0240 | 0.00557 | 63.8 | 4.201 | 2.544 | 1.354 |
| ModernBERT | SMILES | ID | 14.68 | 0.0117 | 0.0064 | 0.0076 | 0.0095 | 0.00073 | 40.1 | 0.870 | 0.698 | 0.407 |
| | | OOD | 44.49 | 0.0287 | 0.0216 | 0.0232 | 0.0324 | 0.00170 | 228.7 | 2.489 | 2.657 | 0.611 |

Table 7: RMSE scores of all models on OOD and ID tasks. Best performing **ID** and **OOD** models are highlighted in **Black** and **Blue** respectively. The worst performing **ID** and **OOD** models are highlighted in **Orange** and **Red** respectively. The graph-based and hybrid models provide the best scores across nearly all tasks for OOD and ID splits. Numerical encoding issues greatly hamper RTs performance and result in large errors. We additionally provide results on using a Llama large language model for OOD property prediction in Appendix E. All results are averaged across 3 training runs.

## C.7 Broader Impacts

BOOM provides a set of benchmarks that are designed to accelerate the development of generalizable chemical foundation models. In turn, we aim for these chemical foundation models to be used to tackle important societal issues such as developing revolutionary pharmaceuticals or energy storage materials. Nevertheless, we note that it is important to develop appropriate safeguards to ensure that such chemistry machine learning models are not used for the development of dangerous chemicals. To this end, we advocate for the continued development of chemical safety benchmarks to assess the potential for chemistry machine learning models to design harmful materials.

## C.8 Discussion of Data Augmentation

The data augmentation experiment in the main text serves to demonstrate that even a few thousand OOD samples can effectively convert an OOD region into an in-distribution (ID) one. This has important implications for real-world applicability. Our experiment shows that property prediction performance can be improved with minimal OOD data-highlighting that even a relatively small number of molecules in the OOD region can significantly enhance model generalization. Our hope with this experiment is that it inspires future work exploring how targeted generation can improve OOD generalization, rather than definitively prescribing a solution for solving OOD generalization.

Nevertheless, the feasibility of identifying useful OOD points in the first place is a significant, unsolved challenge and likely requires approaches based on Bayesian Optimization or Active Learning. Nevertheless, recent work that demonstrates that generative models combined with active learning may be able to extrapolate in property space. Prior work has shown that generative models without active learning were not able to extrapolate beyond the training data.Antoniuk et al. [2025] However, once active learning on DFT simulations was incorporated into the generative loop, the model showed strong potential to generate molecules with properties that extend far beyond the training data, demonstrating an ability to extrapolate in property space. In another more extreme example, the authors used their STGG+ autoregressive generative model with active learning to discover molecules with an oscillator strength of 27.7, compared to a maximum of 9.3 in their training data, and a value of 13.01 without active learning.Jolicoeur-Martineau et al. [2025] These two examples serve to empirically demonstrate that iteratively generating molecules, labeling them with ground-truth simulations, and then retraining the property prediction models may lead generative models to recognize extreme-property domains in chemical space.

Conceptually, we hypothesize that this approach may be possible because this iterative active learning will continually trend towards molecules with improved properties as long as the property prediction models are able to determine the relative ordering of molecules with respect to the property of interest, rather than needing quantitatively correct predictions. Then, it is the ground truth simulations (DFT) that will provide the true property labels of the molecules. Our scatter plots shown in Figure 13 as an example, show that the property prediction models do seem to have this capability, as the most extreme-property molecules are consistently identified, and thus, would be preferentially generated.

Encouragingly, the proposed overall approach of improving OOD performance through iterative active learning has already seen some reported success in the literature. In recent work, the authors note that after three iterations of active learning, the prediction RMSE reduces by $83\%$ when evaluated on hold-out test molecules from across the entire active learning run.Antoniuk et al. [2025] Similarly in another work, the authors found that multiple iterations of active learning to generate novel inorganic structures reduced the error on structure-based OOD energy predictions from >200meV/atom down to 25meV/atom.Merchant et al. [2023] Although there is still much to explore, we feel that there is some growing evidence to believe that OOD generalization can be improved in this manner.

## C.9 Lipophilicity Dataset

To provide a further assessment of OOD performance beyond the computational datasets discussed in the main text, we also evaluate a subset of the models on the Lipophilicity Dataset from MoleculeNet.Wu et al. [2018] This dataset consists of 4200 experimental measurements of the octanol/water distribution coefficient, which is of relevance for drug compounds. The inclusion of the Lipophilicity dataset serves as an exemplary dataset for performing OOD evaluations on experimentally measured properties, rather than only computed physicochemical properties.

## C.10 Statistical Analysis

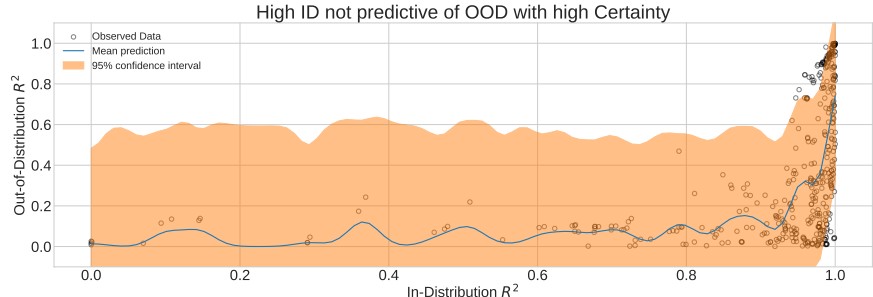

Figure 11: Gaussian process fitting of OOD $R^2$ values given ID $R^2$ scores. Interestingly, lower ID $R^2$ scores are correlated with low OOD $R^2$ scores, as expected. Higher ID scores are not necessarily predictive of OOD values, signalling a need to test models with ID-OOD splits as well as random splits.

| Property | N | P-value (Kruskal-Wallis) | P-value (Mann-Whitney U) | | |
|---|---|---|---|---|---|
| | | | Geometric > Transformer | Geometric > Graph | Graph > Transformer |
| HoF | 39 | 0.00000 | 0.00000 | 0.00398 | 0.00485 |
| Density | 39 | 0.00424 | 0.00175 | 0.07845 | 0.00673 |
| HOMO | 39 | 0.05119 | 0.01569 | 0.48837 | 0.02197 |
| LUMO | 39 | 0.00046 | 0.00010 | 0.20389 | 0.00485 |
| GAP | 39 | 0.10577 | 0.98113 | 0.78453 | 0.85461 |
| ZPVE | 39 | 0.05163 | 0.01067 | 0.51164 | 0.05052 |
| $R^2$ | 39 | 0.00006 | 0.00018 | 0.00002 | 0.30314 |
| $\alpha$ | 39 | 0.01024 | 0.00165 | 0.23808 | 0.05123 |
| $\mu$ | 39 | 0.49232 | 0.83475 | 0.76714 | 0.87963 |
| $C_v$ | 39 | 0.02554 | 0.00641 | 0.45356 | 0.02074 |

Table 8: We perform statistical analysis on the OOD performance of model groups while controlling for the property. We first use the Kruskal-Wallis test to detect whether there is a statistically significant difference between Geometric, Transformer, and Graph models, given a property. Then we perform the Mann-Whitney U hypothesis tests to identify orderings within the groups. Interestingly, we only fail to reject the null hypothesis for $\mu$ and GAP, as many of the models performed poorly on the two tasks.

| Model | MLP | Random Forest | Regression Transformer | MoLFormer | Chemprop |
|---|---|---|---|---|---|
| ID | $0.866 \pm 0.09$ | $0.548 \pm 0.001$ | $1.139 \pm 0.02$ | $0.473 \pm 0.006$ | $0.463 \pm 0.01$ |
| OOD | $2.041 \pm 0.2$ | $1.576 \pm 0.006$ | $1.164 \pm 0.003$ | $0.956 \pm 0.004$ | $1.051 \pm 0.02$ |

Table 9: RMSE values of various models on the Lipophilicity Dataset from MoleculeNet. We report the RMSE values, averaged across 3 training runs, along with their standard deviations.

## D Parity Plots

As there are more than 150 plots, we provide the parity plots for all of our experiments in a compressed layout in the following section, intended for observing the prediction trends for the model. We also upload the higher resolution images, as well as the actual predictions, including the training/fine-tuning code for all models, to our repository.

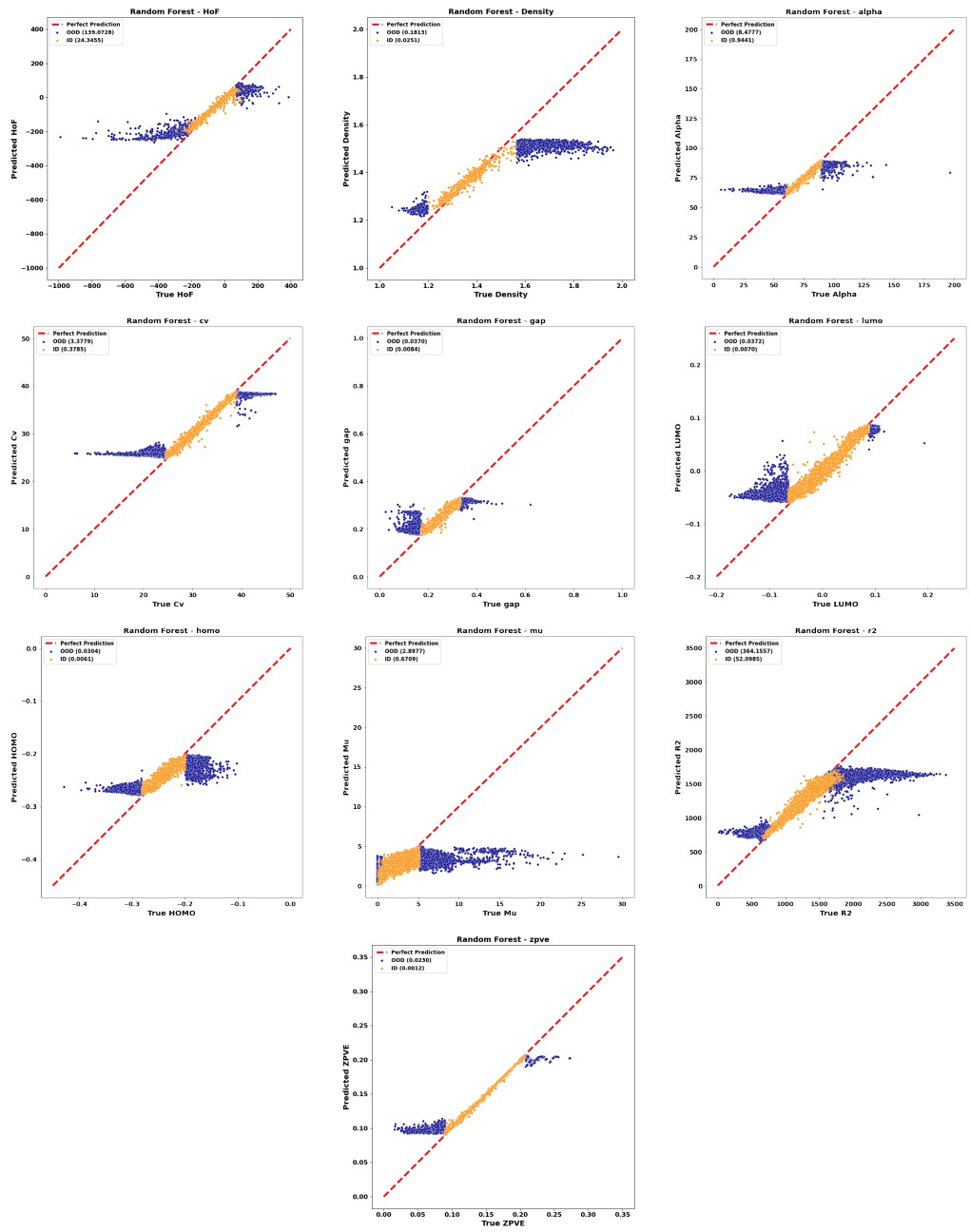

Figure 12: Parity Plots for Random Forest on 10K and QM9 OOD tasks.

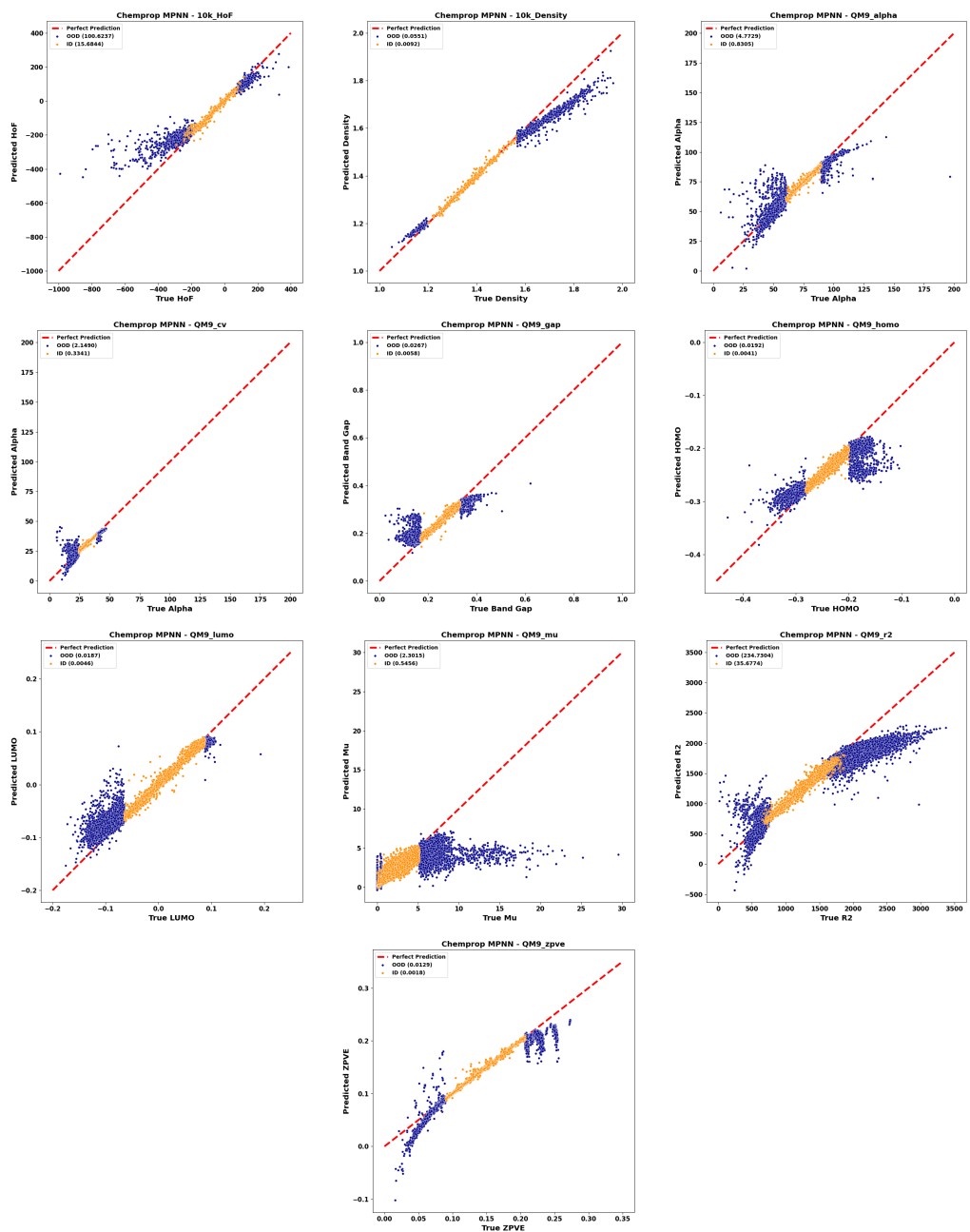

Figure 13: Parity Plots for Chemprop on 10K and QM9 OOD tasks.

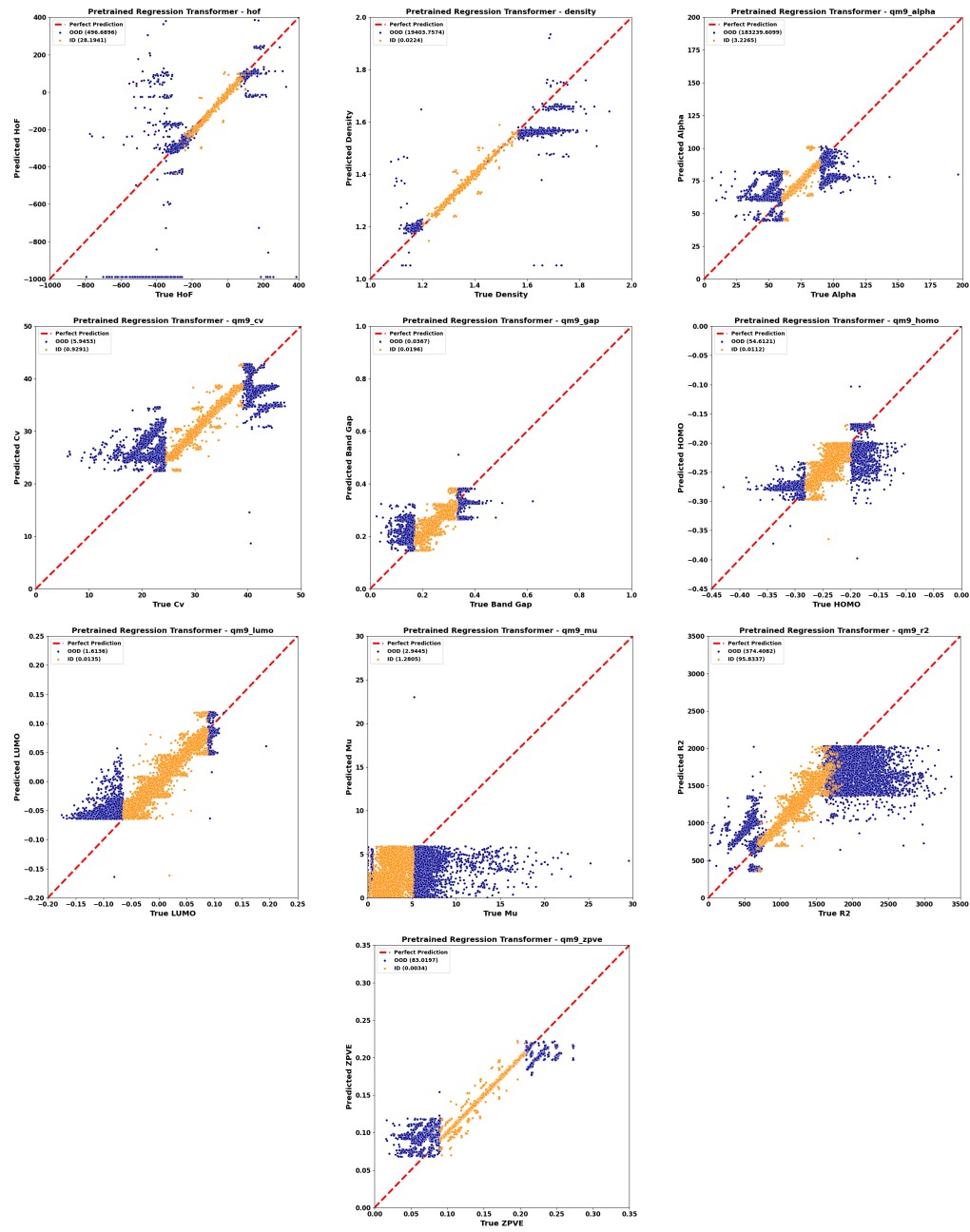

Figure 14: Parity Plots for Regression Transformer (with Pretraining) on 10K and QM9 OOD tasks.

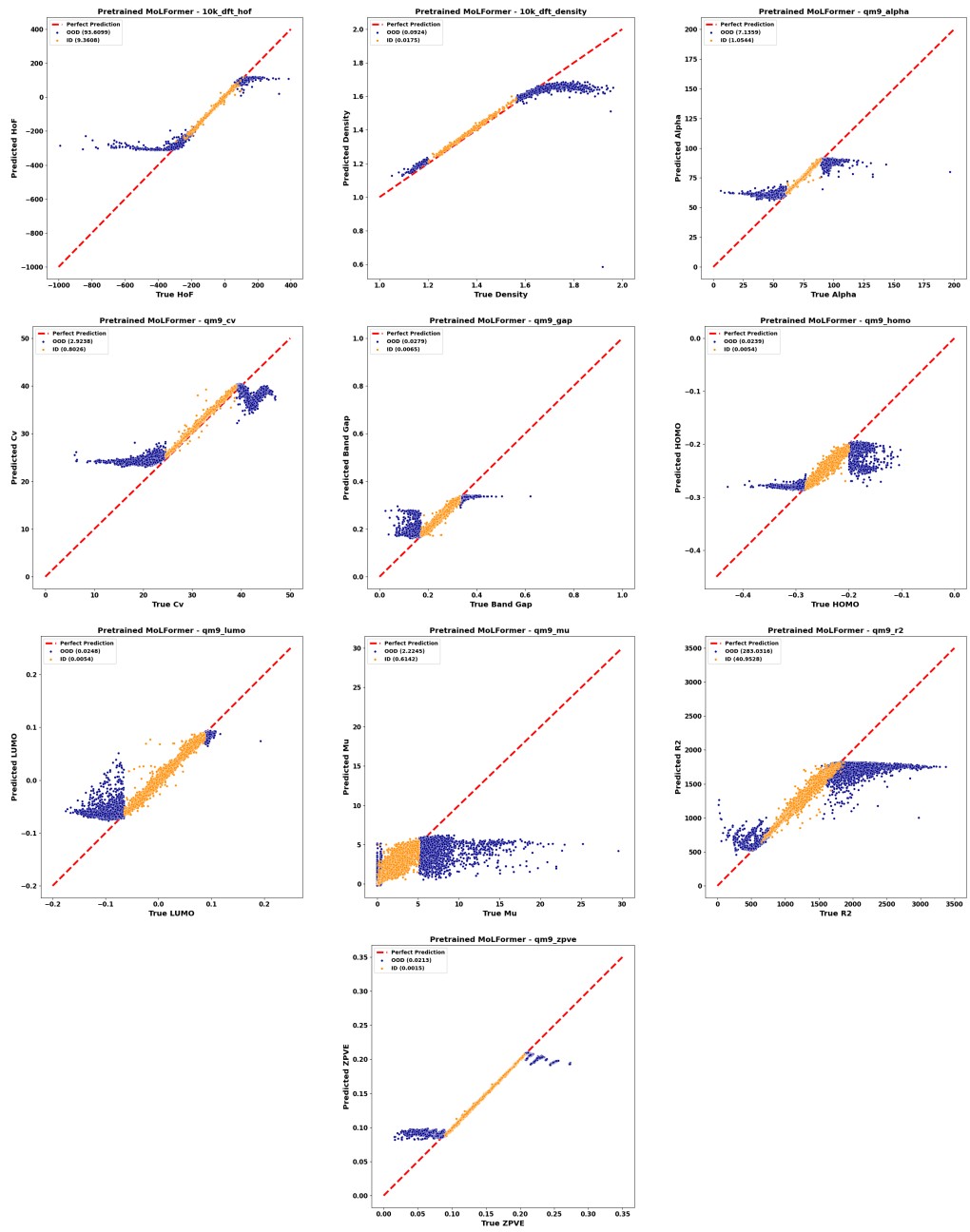

Figure 15: Parity Plots for MoLFormer (with Pretraining) on 10K and QM9 OOD tasks.

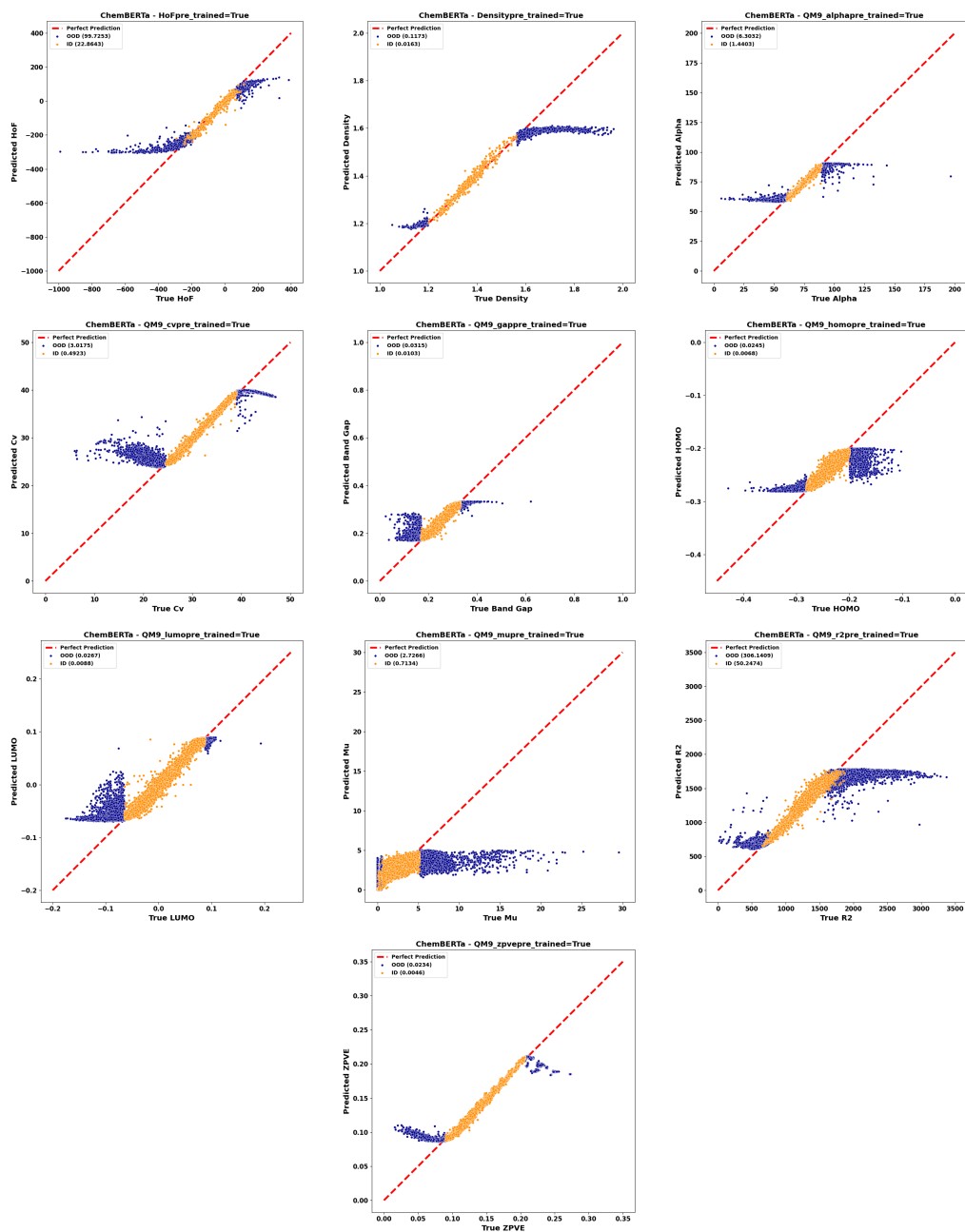

Figure 16: Parity Plots for ChemBERTa (with Pretraining) on 10K and QM9 OOD tasks.

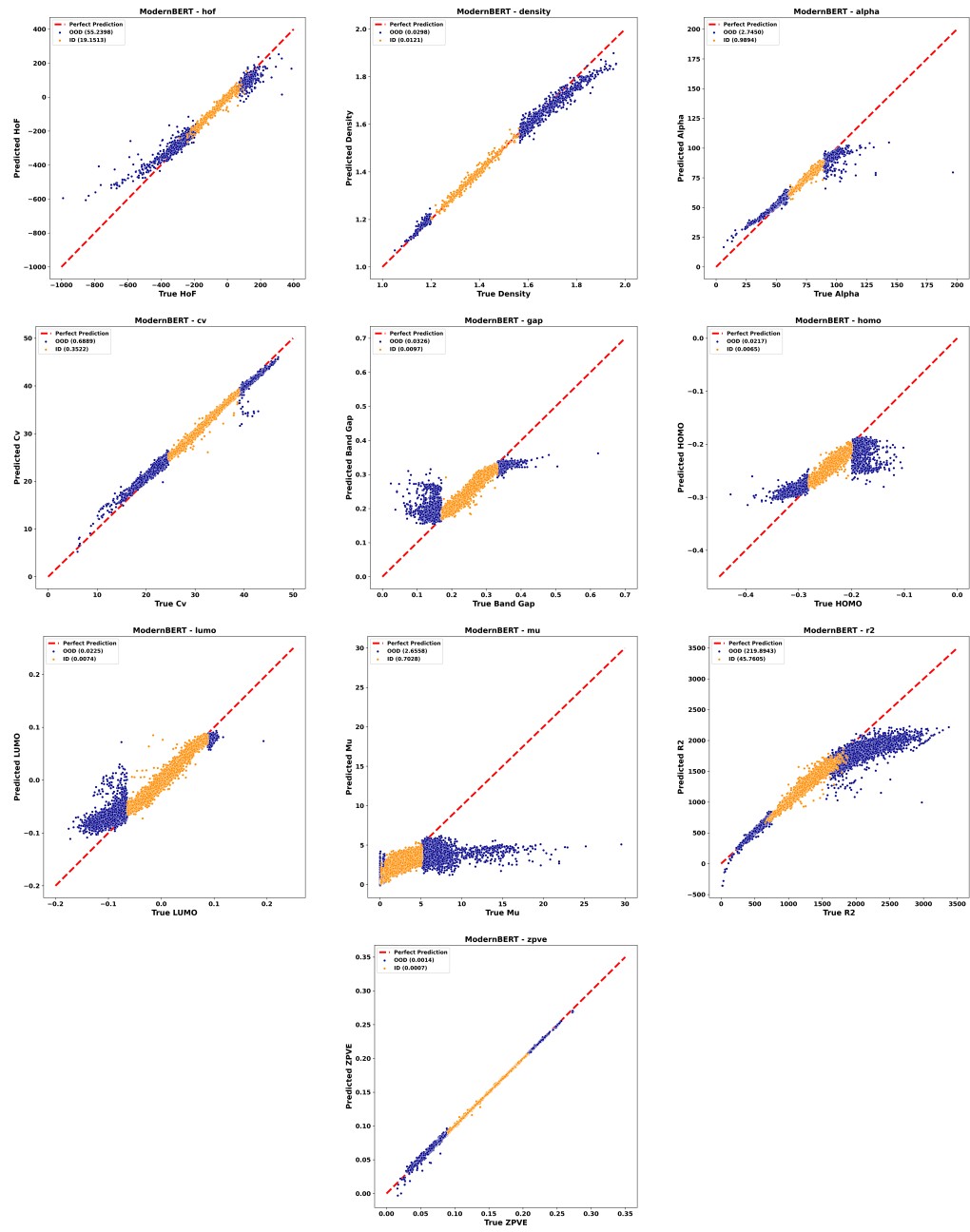

Figure 17: Parity Plots for ModernBERT on 10K and QM9 OOD tasks.

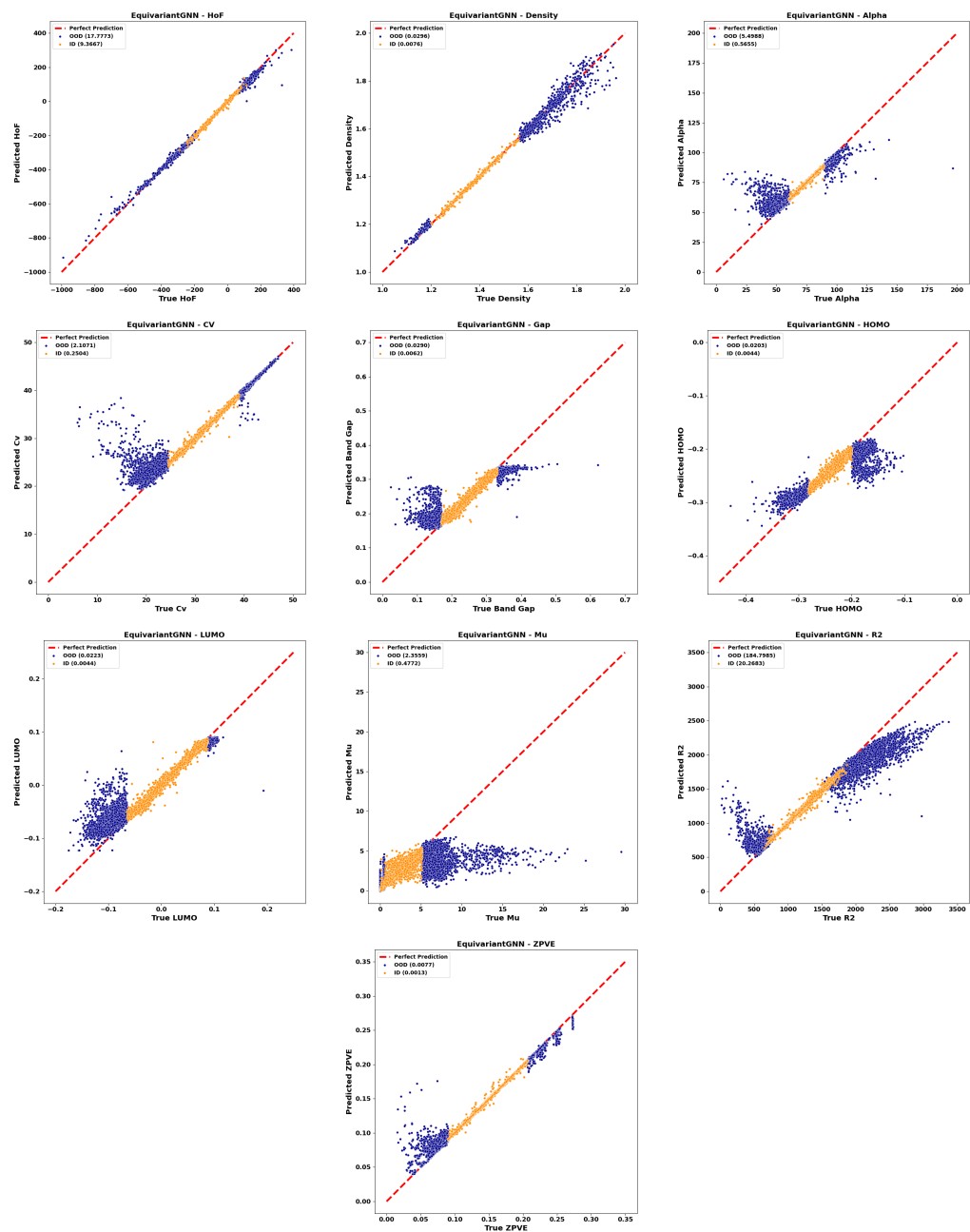

Figure 18: Parity Plots for EGNN on 10K and QM9 OOD tasks.

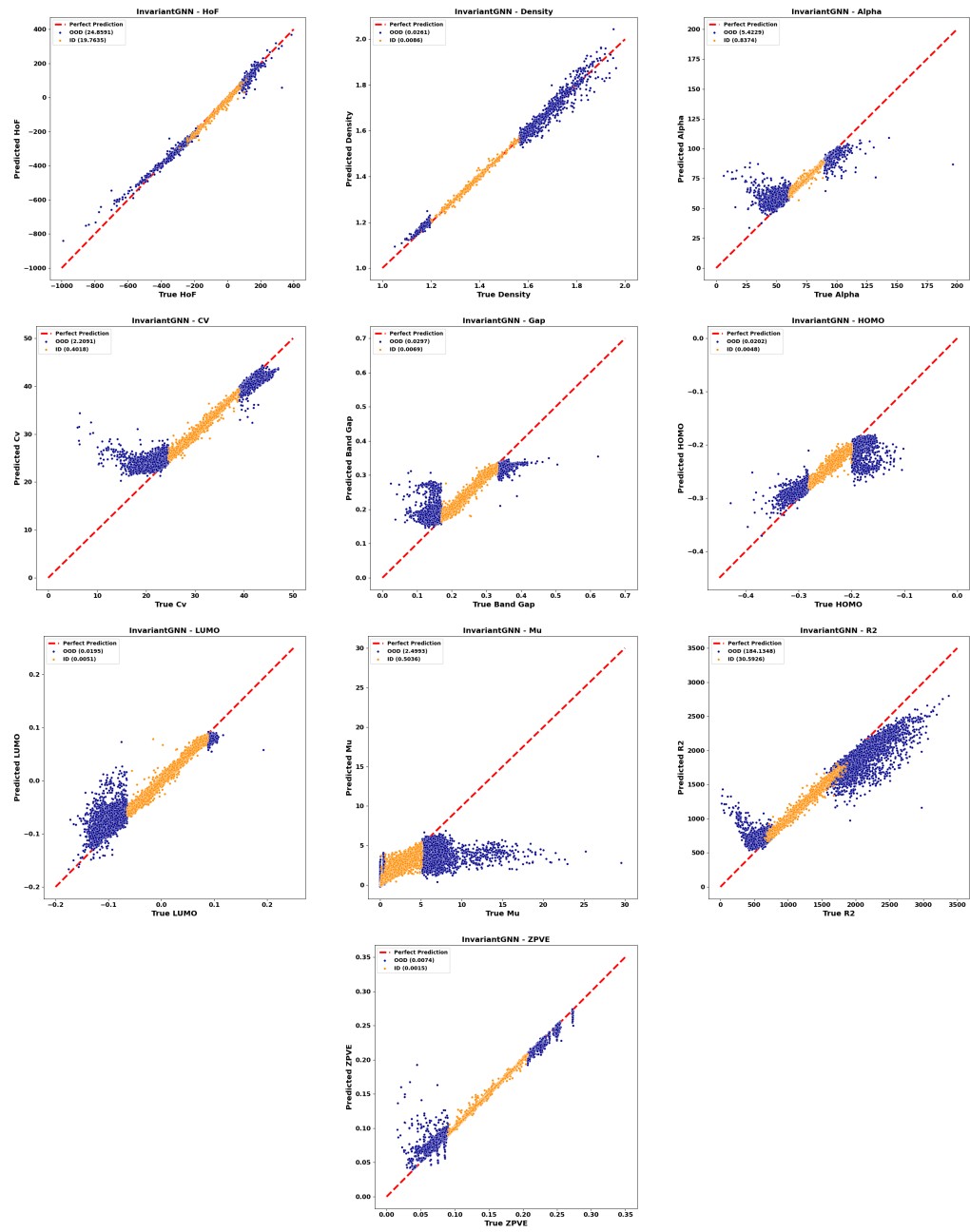

Figure 19: Parity Plots for IGNN on 10K and QM9 OOD tasks.

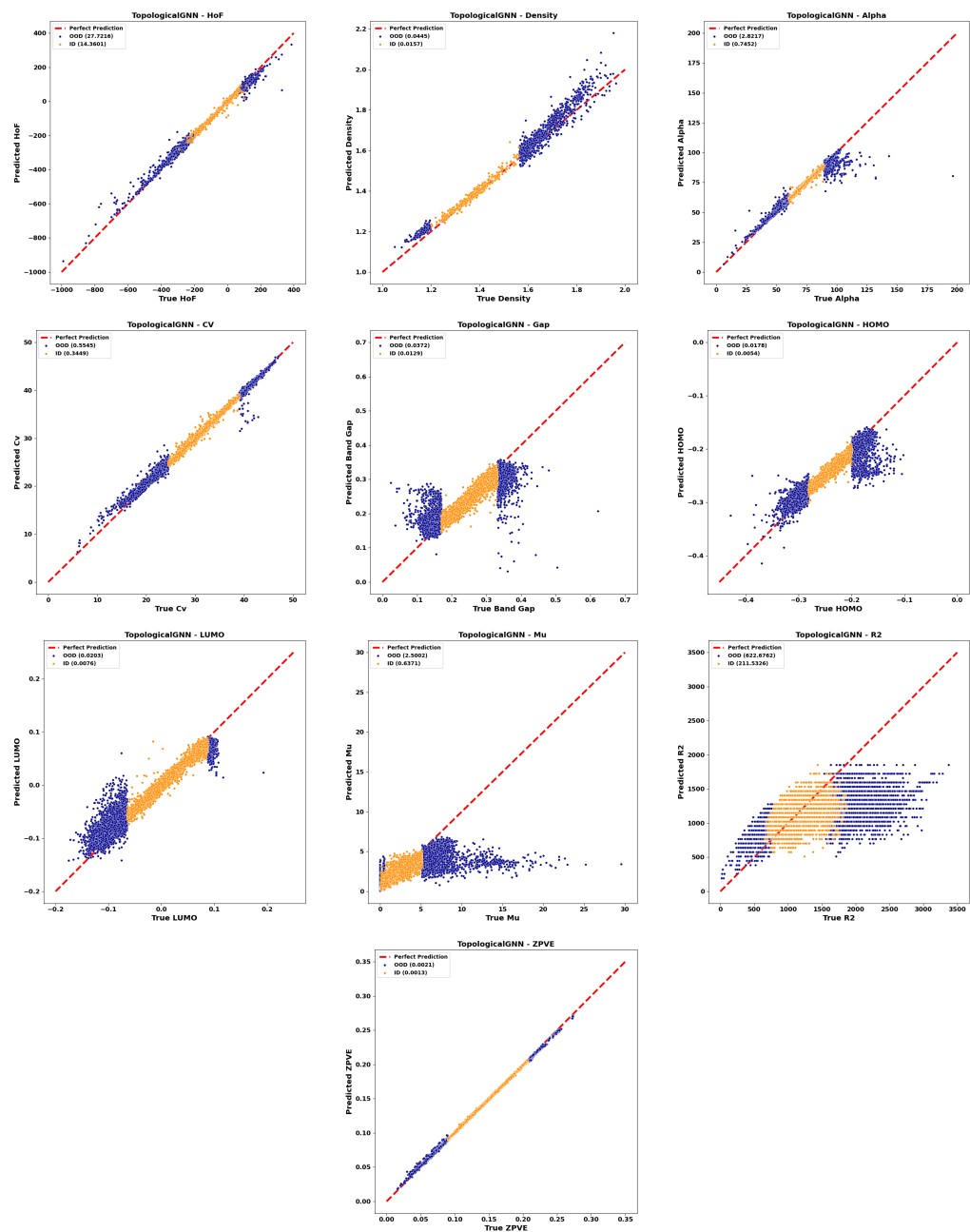

Figure 20: Parity Plots for TGNN on 10K and QM9 OOD tasks.

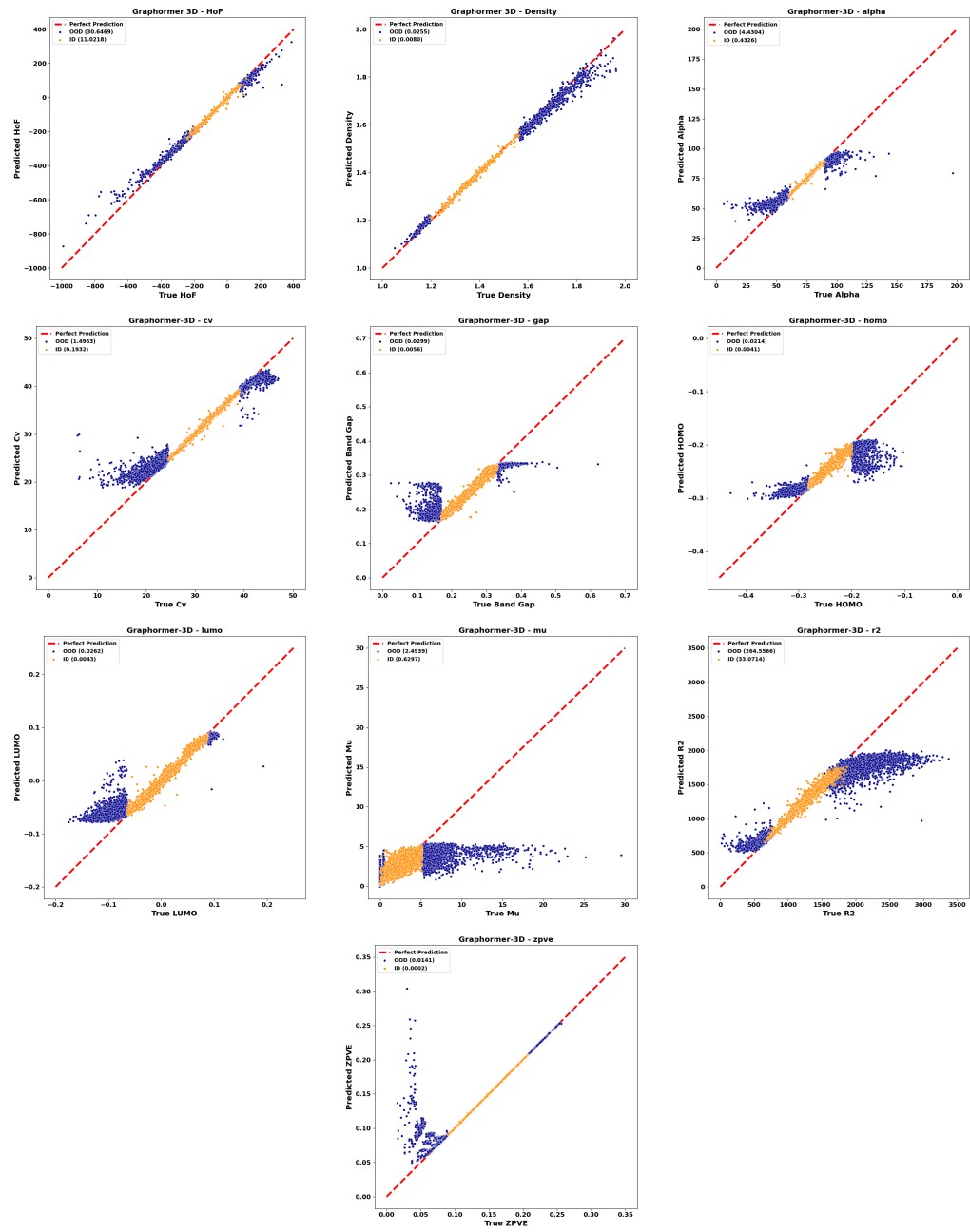

Figure 21: Parity Plots for Graphormer(3D) on 10K and QM9 OOD tasks.

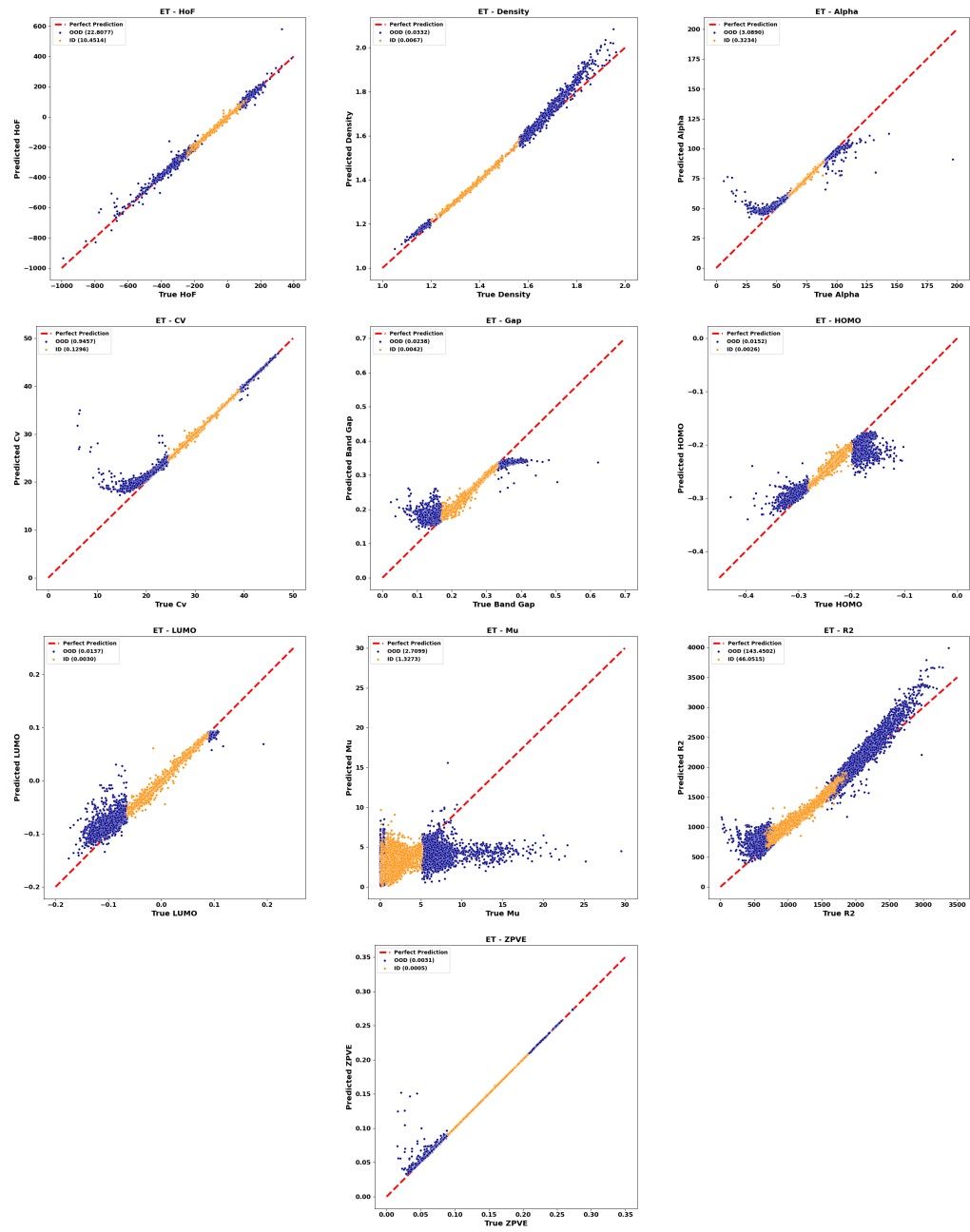

Figure 22: Parity Plots for TorchMD-ET on 10K and QM9 OOD tasks.

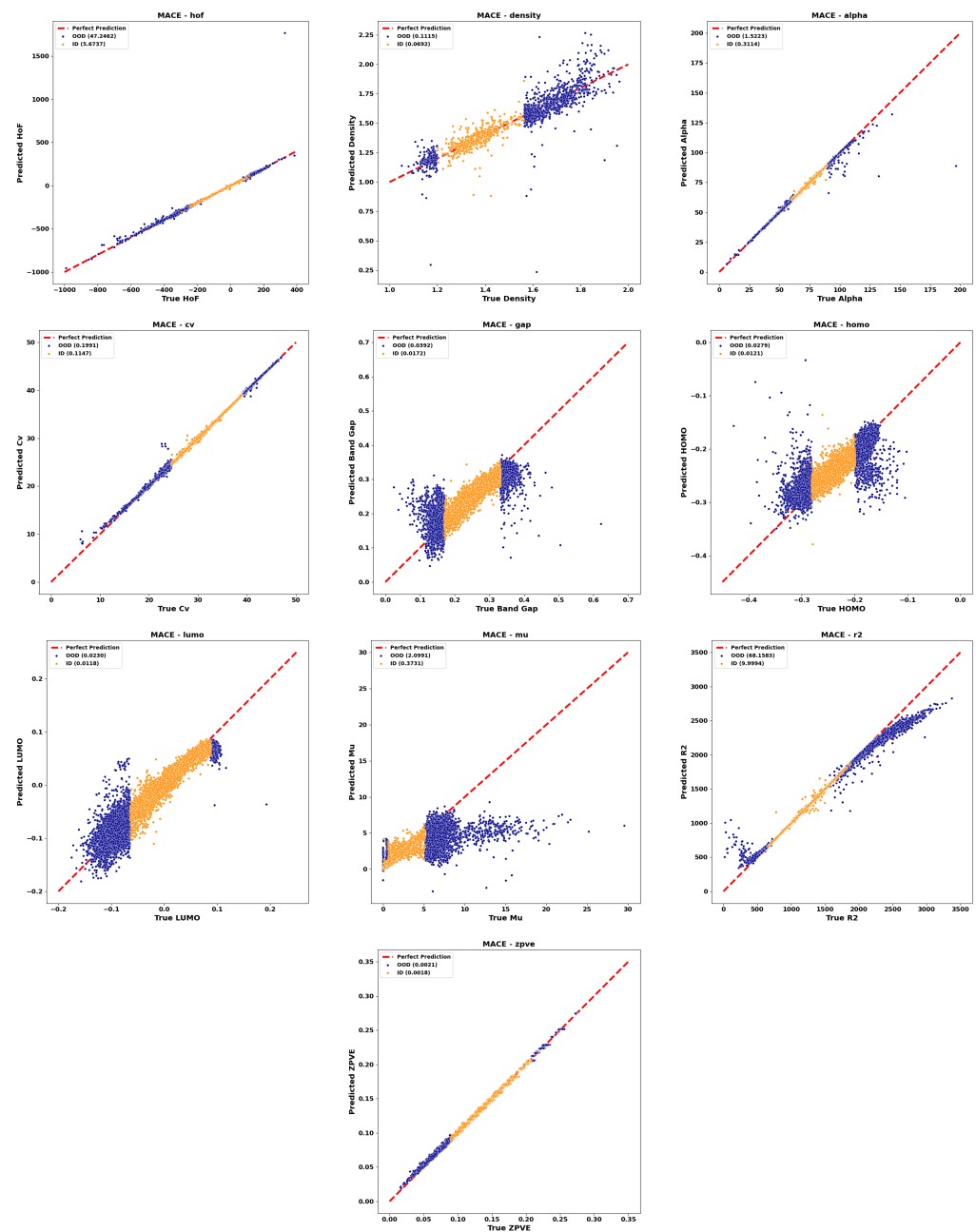

Figure 23: Parity Plots for MACE on 10K and QM9 OOD tasks.

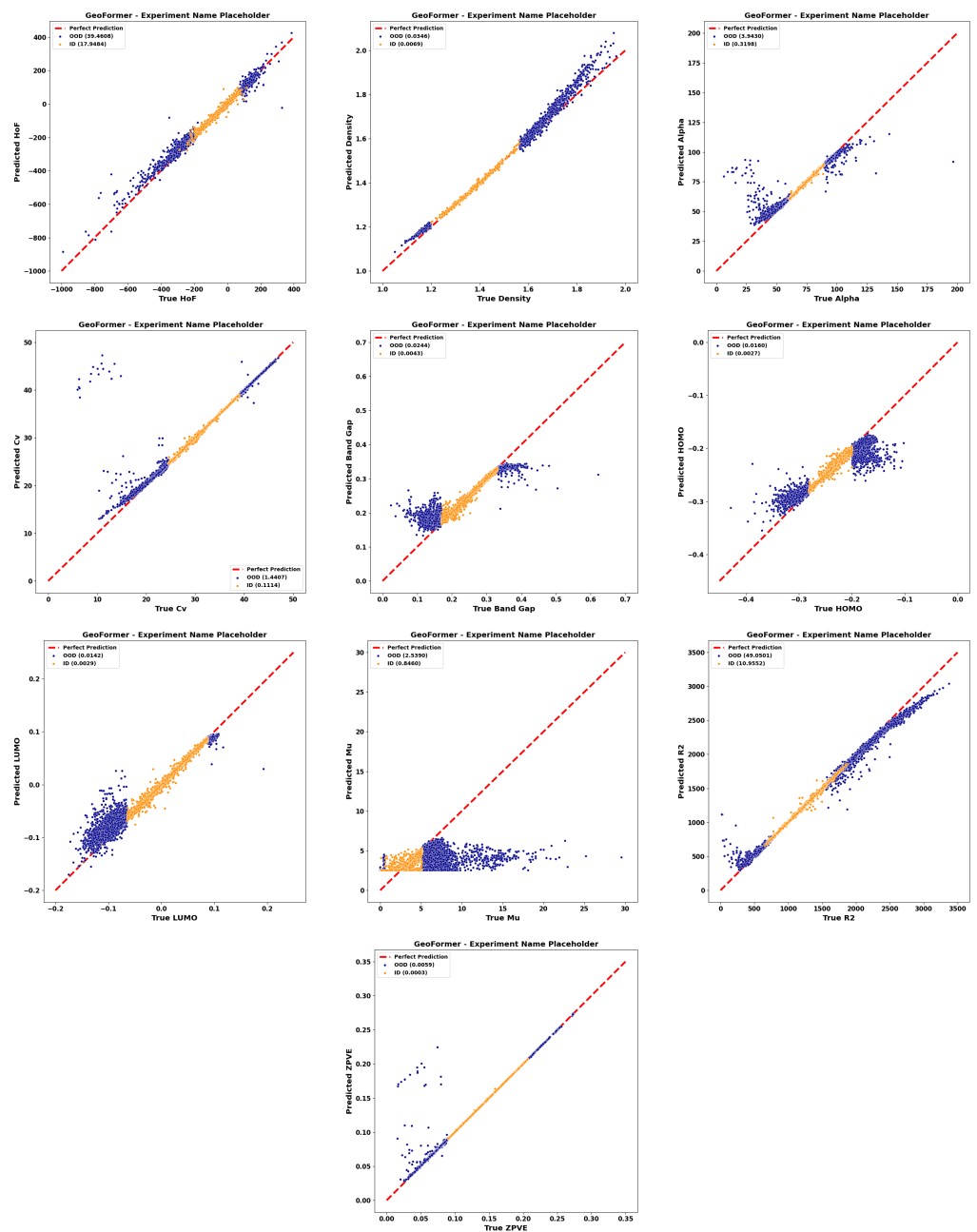

Figure 24: Parity Plots for GeoFormer on 10K and QM9 OOD tasks.

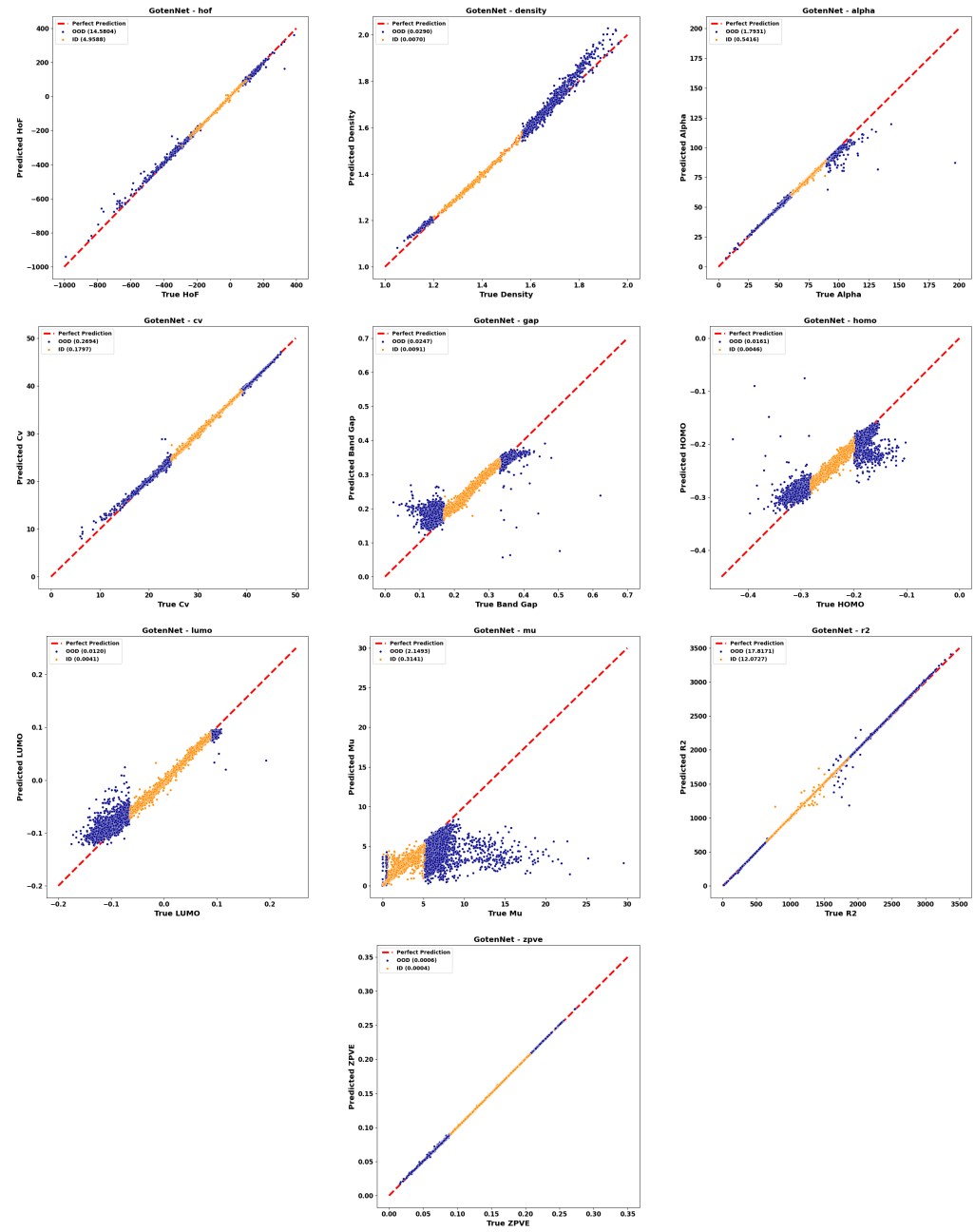

Figure 25: Parity Plots for GotenNet on 10K and QM9 OOD tasks.

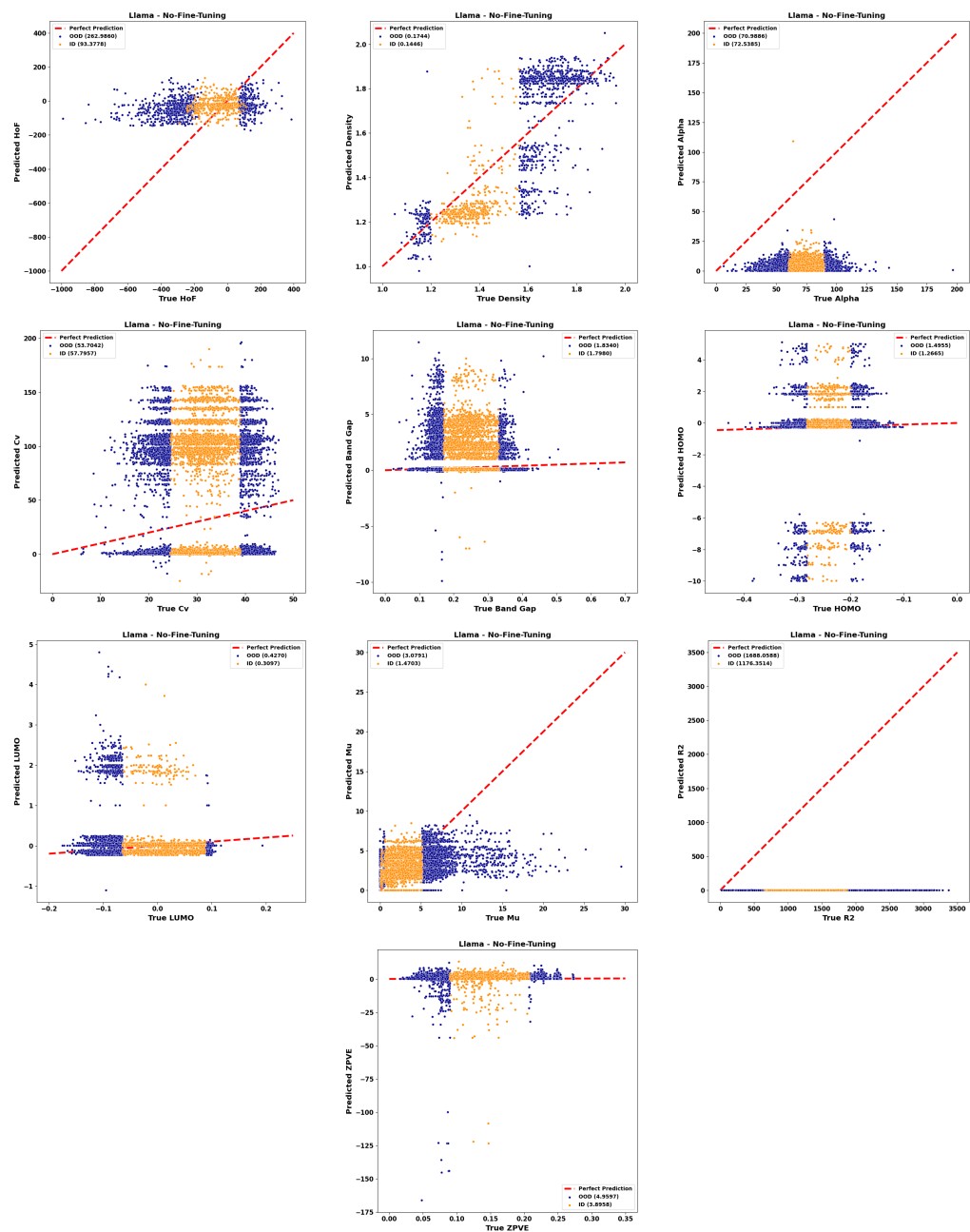

Figure 26: Parity Plots for LLAMA on 10K and QM9 OOD tasks.

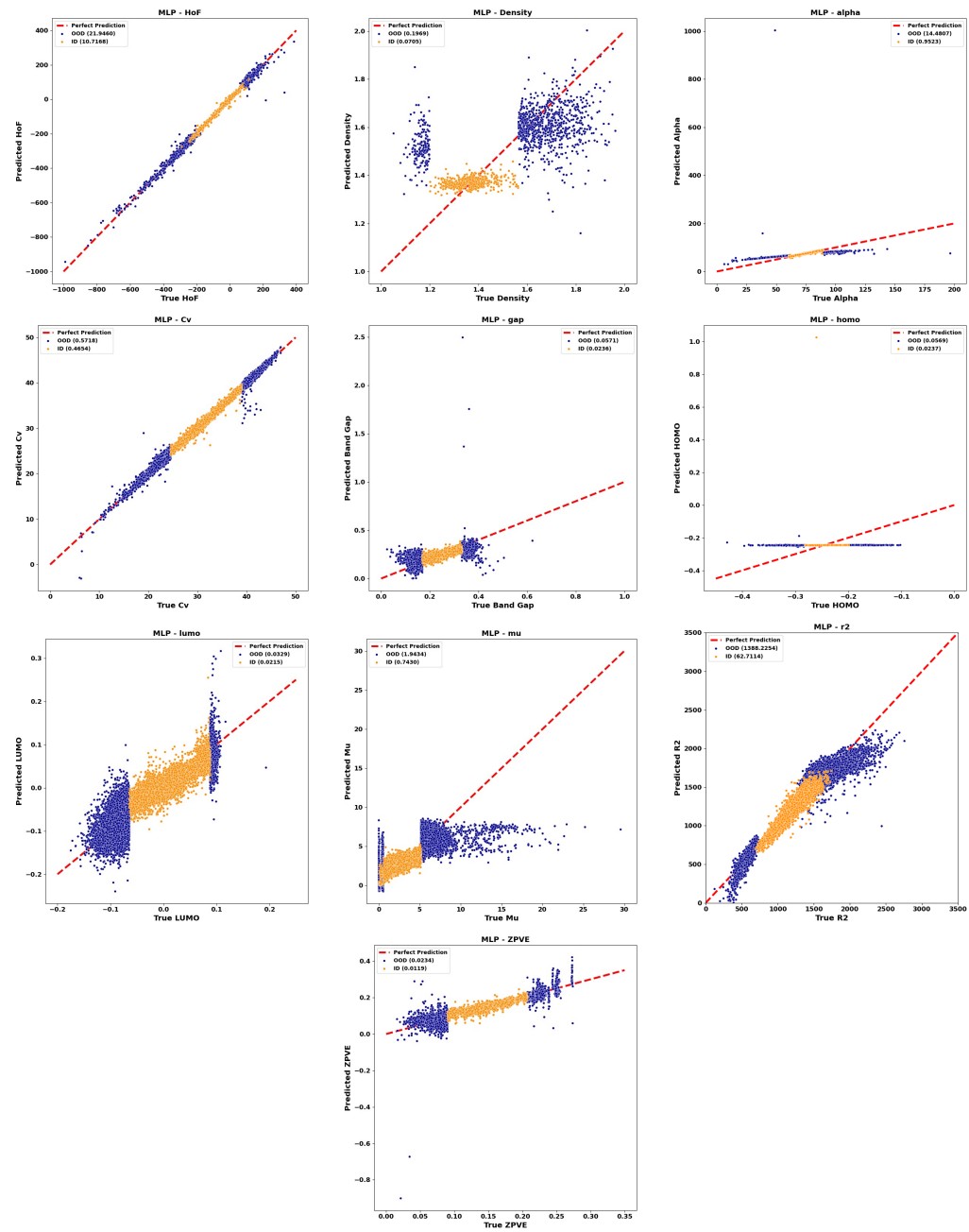

Figure 27: Parity Plots for MLP on 10K and QM9 OOD tasks.

# E    Property Prediction with LLMs

Large language models (LLMs) have seen increasing usage for a wide range of molecular design tasks including property prediction,[Jacobs et al., 2024, Jablonka et al., 2023] molecular synthesis prediction, and property-guided molecule design.[Jablonka et al., 2023, Bhattacharya et al., 2024] In this section, we benchmark the performance of LLMs on BOOM's OOD property prediction tasks. We note that in the context of LLMs, it is more challenging to robustly define OOD molecules without knowing the exact training corpus of the LLM. For example, we anticipate that it is possible that the density of some of the molecules in our 10k Density OOD test set appear as natural language in the training corpus of the LLM. Although we believe that including benchmarking of the

OOD performance of LLMs is important due to their widespread usage, we caution against direct comparison of the models in the main text due to these possible data leakage concerns.

## E.1  Experiment Details

We use the LLAMA-3.1-8B model provided by Meta.[Grattafiori et al., 2024b] We use the following prompt to generate the properties:

> Do not include any other text. \n Only return a floating point number with 4 digits after the decimal point. For SMILES: <smiles> predict the <property> (<property_description>) in <units> value: "

Where, **<smiles>** is the SMILES representation of the molecules. **<property>** is one of the ten properties mentioned above. **<property_description>** is the description of the property, and **<units>** is the units of the properties.

| Property | Description Text | Unit Text |
|---|---|---|
| HoF | solid heat of formation (using a group additivity approach) | g/cc |
| Density | crystalline density | kCal/mol |
| $\alpha$ | isotropic polarizability | a_0^3 |
| $C_v$ | heat capacity at 298.15 K | cal/molK |
| HOMO | energy of the highest occupied molecular orbital | Hartree energy |
| LUMO | energy of the lowest unoccupied molecular orbital | Hartree energy |
| Gap | energy difference between the highest occupied and lowest unoccupied molecular orbital | Hartree energy |
| $\mu$ | dipole moment | Debye |
| $\langle R^2 \rangle$ | electronic spatial extent | a_0^3 |
| ZPVE | zero point vibrational energy | Hartree energy |

Table 10: Property descriptions and units used in the LLAMA prompt

We prompt the model to output only floating-point values and use a parser to extract the first decimal numerical values from the generated output.

## E.2  Llama Results

| Model | Split | HoF | Density | HOMO | LUMO | GAP | ZPVE | $\langle R^2 \rangle$ | $\alpha$ | $\mu$ | $C_v$ |
|---|---|---|---|---|---|---|---|---|---|---|---|
| LLaMA 3.1 8B | ID | 93.3778 | 0.1446 | 93.3778 | .3097 | 1.0234 | 3.8958 | 1176 | 75.44 | 1.4703 | 57.796 |
| (no finetuning) | OOD | 262.9860 | 0.1744 | 262.9860 | 1.1436 | 0.9545 | 4.9597 | 1688 | 73.34 | 5.6357 | 32.026 |

Table 11: RMSE of LLaMA models on all OOD and ID tasks.

| Model | Split | HoF | Density | HOMO | LUMO | GAP | ZPVE | $\langle R^2 \rangle$ | $\alpha$ | $\mu$ | $C_v$ |
|---|---|---|---|---|---|---|---|---|---|---|---|
| LLaMA 3.1 8B | ID | 0.008 | 0.2050 | 0.0009 | 0.0025 | 0.0127 | 0.0008 | 0.0000 | 0.0034 | 0.0845 | 0.0000 |
| (no finetuning) | OOD | 0.0254 | 0.1048 | 0.0007 | 0.0000 | 0.0018 | 0.0022 | 0.0005 | 0.0016 | 0.0631 | 0.0007 |

Table 12: Batched $R^2$ scores of LLaMA models on all OOD and ID tasks.

