# OpenReview forum: "BOOM: Benchmarking Out-Of-distribution Molecular Property Predictions of Machine Learning Models"
_NeurIPS.cc/2025/Datasets_and_Benchmarks_Track — NeurIPS 2025 Datasets and Benchmarks Track poster_

### Official Review · Reviewer_wSb4 · 2025-06-17

**Rating:** 5
**Confidence:** 3

**Summary:**

The BOOM dataset derives from the well-known chemical datasets QM9 and 10k and consists of 10 quantum chemical molecular properties (like heat capacity, HOMO-LUMO gap...) for each molecule. Each molecule is encoded by either SMILES, graphic or 3D representations when it is passed to various machine learning models, like random forest, graph neural networks, transformers...
This paper adopts the concept or label shift formulation of out-of-distribution (OOD) data, where the molecular property of the OOD data is at the tail of the distribution.
This paper provides the in-distribution (ID) and out-of-distribution test error and finds out that no single model can outperform the others in OOD performance, hence BOOM serves as a challenge for future models in extrapolation of chemical knowledge.

**Dataset Code Accessibility:**

Yes

**Dataset Code Comments:**

The code is presented on a Github repository and is well-documented.

**Ethical Considerations:**

No, there are no or only very minor ethics concerns

**Final Justification:**

I thank the authors for this detailed reply. I think they have addressed all my concerns. Hence I keep my original rating and lean to accept the paper.

In particular, the authors explained:
1. the scenario where label-based OOD generalization is relevant in real-world setting;
2. their additional records on the complexity of each algorithm;
3. and my confusion on the computational complexity on foundation model pre-training and fine-tuning.

In addition, the authors have provided more experimental results in response to other reviews.

Although I agree that this paper still cannot exhaust all SOTA models and more complex datasets, it already provides an integrated pipeline on measuring label-based ID/OOD generalization which serves as a base for ad-hoc extentions on newer models and datasets.

**Limitations Weaknesses:**

My main concern is that the ID/OOD split is only done on the body/tail distribution of the chemical properties (label) of the molecules. I wonder if there are other splits that can more truthfully showcase the extrapolation ability of LM models, say by molecular size (from the inputs). I think there can be more discussion about the definition of OOD and its relevance in practical settings, like drug discovery, toxicity check...

Also, I notice that the models used in this paper have parameter numbers varying in magnitude. How can we ensure it is a fair comparison?
From the checklist, the paper reports that each experiment is only done once "due to the computational expense associated with model training". On the other hand, it reports no statistics on experiments compute resources, "as the models and datasets are small, the experiments do not require any specialized hardware..." Is it not a bit contradictory? I think reporting the computational complexity from each model could give the readers a better view.

**Strengths Contributions:**

This paper tests on models of both GNN-type and transformer-type, providing a comprehensive picture of their ID and OOD performance. More ablation test and discussion are provided on more detailed comparisons between individual models, like pre-training on data less related to downstream task might even harm OOD generalization, and models with graphic input outperform those SMILE input, despite the two representations are equivalent, showing there might be implicit biases on those models which helps generalization.

---

> ### Author Rebuttal · Authors · 2025-07-31
>
> **General response:** We thank the reviewer for their thoughtful suggestions and comments on our work.  In this work, we have developed a large-scale benchmark of OOD performance of state-of-the-art and popular ML models with more than 150 pairs of model-task experiments. Although other works have explored OOD splitting based on the chemical structure of molecules, our work is novel in that it provides an assessment of property-based OOD splitting for an extremely diverse range of model architectures. We feel that the property-based OOD splitting of BOOM is especially valuable because it directly reflects the goals of multi-property molecular optimization, where the explicit goal is to find a novel molecule that maximizes the properties of interest. With the additional new experiments that we have added, we believe that our BOOM benchmark is now even broader in scope, applicable to more model architectures, and has significantly improved statistical significance.
>
> **Summary of new results:**
> 1.) To broaden the range of applications covered by BOOM, we have added benchmarking on the Lipophilicity Dataset, sourced from MoleculeNet.
>
> 2.) For the experiments in Table 1, we have performed two additional training runs such that greater statistical significance can be drawn from the results.
>
> 3.) We have added two additional 3D model architectures to BOOM: GotenNet and Geoformer.
>
> 4.) As an additional source of information, we have also provided a summary of computational resources required to train each model.
>
> > My main concern is that the ID/OOD split is only done on the body/tail distribution of the chemical properties (label) of the molecules. I wonder if there are other splits that can more truthfully showcase the extrapolation ability of LM models, say by molecular size (from the inputs). I think there can be more discussion about the definition of OOD and its relevance in practical settings, like drug discovery, toxicity check…
>
> We thank the reviewer for their feedback on the choice of our ID/OOD split. In general, we believe our work is complementary to OOD methods that focus on the input space (such as molecular size). Depending on the application, one may require models to perform well in one definition over another or even both. For example, if a discovery pipeline is attempting to extrapolate in the structure or composition space while keeping molecular properties the same, as is the case in lead molecule optimization, structure-based OOD generalization is a key characteristic for the ML model. On the other hand, if the discovery pipeline is focused on extending the frontier of the capabilities of known molecules, property-based OOD is needed.
>
> As a specific motivating example of the relevance of OOD in practical molecule discovery, we point to prior work [arXiv:2501.02059], which focused on using the Chemprop model to discover novel molecules with high density and heat of formation values. However, due to the limited OOD performance of Chemprop, it was found that it performed extremely poorly at identifying both molecules with high density (23% precision) and high heat of formation (14% precision). In real-world molecule discovery settings where a chemist will attempt to synthesize these ML-generated molecules, elucidating the OOD performance of a ML model beforehand can serve to vastly reduce the number of failed synthetic efforts.
>
> To further highlight the relevance of BOOM to the domain of drug discovery, we have expanded our benchmarking to include the Lipophilicity dataset from MoleculeNet. As described in the MoleculeNet paper, “Lipophilicity is an important feature of drug molecules that affects both membrane permeability and solubility.” The inclusion of the Lipophilicity dataset in BOOM serves to further illustrate its relevance in the field of drug discovery.
>
> >  Also, I notice that the models used in this paper have parameter numbers varying in magnitude. How can we ensure it is a fair comparison?
> The number of parameters and computational costs are important model considerations, but beyond the scope for our work. We provide Table 4 (Appendix) that enumerates the model parameters for all models tested. We believe there is a strong potential for future work to address RMSE / Parameter and RMSE / FLOP, as has been done in domains such as NLP and CV. However, our high-level goal with our benchmarking was to assess model architectures as defined in the literature or publicly available code, without any changes to the model architectures. This choice was made such that the models in BOOM are representative of the performance that the chemistry community can expect if they are directly using the original model architectures.
> From the checklist, the paper reports that each experiment is only done once "due to the computational expense associated with model training". On the other hand, it reports no statistics on experiments compute resources, "as the models and datasets are small, the experiments do not require any specialized hardware..." Is it not a bit contradictory? I think reporting the computational complexity from each model could give the readers a better view
>
> We agree with the reviewer’s comment that the paper would benefit from providing readers with additional information on the computational complexity of each model. Below, we provide the hardware used and training times of all the models used in BOOM. We will add this information to the Appendix of the paper for the benefit of the readers.
>
> Model        |Device                      |Runtimes (10K) [seconds / epoch]|Runtimes (QM9) [seconds / epoch]
> -------------|----------------------------|--------------------------------|--------------------------------
> Random Forest|2.4 GHz 8-Core Intel Core i9|0.5                             |6
> RT           |V100                        |632                             |10275
> ChemBERTa    |L40                         |13                              |165
> MoLFormer    |H100                        |14                              |71.79
> Chemprop     |H100                        |10                              |23
> EGNN         |L40                         |10                              |115
> IGNN         |L40                         |10                              |115
> TGNN         |L40                         |10                              |115
> MACE         |H100                        |70                              |165
> Graphormer   |Tuo                         |6                               |19
> ET           |L40                         |10                              |75
> Geoformer    |H100                        |60                              |230
> GotenNet     |A100                        |20                              |120
> ModernBERT   |L40                         |15                              |165
> MLP          |H100                        |0.009                           |0.5
>
> We apologize for the confusion in these statements. The note about “the computational expense associated with model training” was intended to refer to retraining of the large chemical foundation models (MoLFormer, Regression Transformer, and ChemBerta), whereas the model finetuning on our chemical datasets is certainly feasible and possible on most hardware. Indeed, in accordance with the comments of Reviewer 4Hma, we have repeated each of the experiments in Table 1 for an additional 2 iterations to provide more meaningful statistics on the performance of the models.

---

### Official Review · Reviewer_4Hma · 2025-06-19

**Rating:** 5
**Confidence:** 4

**Summary:**

The paper proposes a new benchmark dataset to evaluate the ability of molecular property prediction architectures to extrapolate in OOD scenarios where there is a label/concept shift between training and a hypothetical deployment distribution. The evaluation is thorough and the main conclusion is that models struggle in this setting, with the models with the highest inductive bias tend to perform better. They also consider the role of pre-training and see that pre-training tasks that are more aligned to the downstream task also improve the ability to generalise.

**Additional Feedback:**

The paper is interesting, seems quite solid overall and is well written. I think it lacks proper statistical support for its conclusions, which currently are just observations of the trends observed in the experiments. With the addition of the statistical analysis and its proper discussion (whenever necessary), I would update my score to acceptance.

As I mentioned in S1.3., the fact that the study is limited to quantum properties, limits the generalizability of the observations made in the benchmark. I wonder if the authors have considered including other tasks from MoleculeNet where the 3D structure wouldn't be as important (some of the biomedical ones, for example) and if so, why did they choose against including them?

**Dataset Code Accessibility:**

Yes

**Ethical Considerations:**

No, there are no or only very minor ethics concerns

**Final Justification:**

I think the paper is technically solid, well written, introduces a valuable resource for the community to explore extrapolation in the label space (which is often underexplored in molecular machine learning). The issues I had with the original version of the manuscript have been satisfactorily addressed by the authors through the inclusion of additional experiments (replicates + Lipo dataset) and statistical analysis (Kruskal-Wallis + Mann-Whitney U).

**Limitations Weaknesses:**

## 1. Major weaknesses
W1.1. Lack of statistical analysis of the results. The authors rely on Table 1 to draw most of their conclusions, but Table 1 does not provide any information regarding the statistical significance of the results (e.g., confidence intervals/standard deviation across multiple runs). Furthermore, specific tests like ANOVA with post-hoc Tukey's HSD are necessary to determine whether the observations made in the text are significant or just products or random noise [1].

## 2. Minor weaknesses
_(They are not scientifically relevant and do not influence the final score, but could improve the paper)_
W2.1. Table 1 is really difficult to read and interpret. It could be substituted by some kind of figure conveying the same information in a more intuitive way. The table is valuable and informative, but could be kept in the appendix.

## 3. References
1. Ash JR, Wognum C, Rodríguez-Pérez R, Aldeghi M, Cheng AC, Clevert DA, Engkvist O, Fang C, Price DJ, Hughes-Oliver JM, Walters WP. Practically significant method comparison protocols for machine learning in small molecule drug discovery.

**Strengths Contributions:**

## 1. Major strengths
S1.1. The paper proposes a different definition of OOD within the chemical domain, which is different from the usual way it is studied. Concept  or label shift (OOD on the prediction target) instead of the more common feature shift or applicability domain (OOD on the feature space). This new direction is complementary and provides significant value to the field.
S1.2. The benchmarking is thorough and multiple models have been considered.
S1.3. The authors correctly point out that for the tasks they are evaluating 3D models would have an additional inductive bias.

## 2. Minor strengths
S2.1. The paper is clearly written
S2.2. Most of the claims made in the text are well supported by the data provided.

---

> ### Author Rebuttal · Authors · 2025-07-31
>
> General Response: We thank the reviewer for their important suggestions. In this work, we have developed a large-scale benchmark of OOD performance of state-of-the-art and popular ML models with more than 150 pairs of model-task experiments. As a direct consequence of the broad nature of this benchmarking, our work provides important insights into how different model architectures extrapolate in OOD scenarios. As noted by the reviewer, providing statistical information is important for ensuring that our findings are robust and consistent. In accordance with the reviewer’s comments, we have made large strides to considerably strengthen the statistical analyses of our results by performing replicate training runs for the models in Table 1. As a result, we feel that the incorporation of the reviewer’s comments has significantly strengthened our work by solidifying the statistical significance of our results.
>
> Summary of new results:
> For all experiments in Table 1, we have performed two additional training runs. As a result, we can add mean and standard deviation statistics to the results of Table 1, thereby strengthening their statistical significance.
> We perform a statistical analysis of the correlation between ID and OOD performance across the entire dataset, as well as for specific tasks.
> We expand the BOOM benchmark by adding the experimentally measured Lipophilicity dataset, sourced from MoleculeNet. The addition of this benchmark serves to expand the scope of BOOM to include properties relevant to drug discovery, as well as to act as an example of an experimentally-derived dataset.
>
> > Lack of statistical analysis of the results..
>
> Thank you for the thoughtful review and suggestions. We agree that samples per model-task pair should be run multiple times to obtain meaningful statistics. To this end, we make every possible effort to repeat each experiment in Table 1 and provide an additional two iterations, allowing us to calculate a standard deviation for the results. We provide the means and standard deviations of all the models.
>
> | Model         | Split | HoF         | Density        | HOMO           | LUMO           | GAP            | ZPVE            | R^2         | alpha        | mu           | Cv           |
> | ------------- | ----- | ----------- | -------------- | -------------- | -------------- | -------------- | --------------- | ----------- | ------------ | ------------ | ------------ |
> |MoLFormer|ID|10.9409$\pm$ 1.3733|0.0273$\pm$ 0.0089|0.0050$\pm$ 0.0004|0.0052$\pm$ 0.0002|0.0064$\pm$ 0.0002|0.0011$\pm$ 0.0004|40.1408$\pm$ 0.7104|1.0472$\pm$ 0.0230|0.6022$\pm$ 0.0113|0.7854$\pm$ 0.0285|
> |MoLFormer|OOD|93.6226$\pm$ 0.9171|0.0770$\pm$ 0.0134|0.0236$\pm$ 0.0004|0.0256$\pm$ 0.0007|0.0275$\pm$ 0.0005|0.0199$\pm$ 0.0016|314.0819$\pm$ 39.4336|7.3558$\pm$ 0.3554|2.2315$\pm$ 0.0193|3.6666$\pm$ 1.1853|
> |EGNN|ID|10.0684$\pm$ 1.7235|0.0077$\pm$ 0.0001|0.0048$\pm$ 0.0004|0.0052$\pm$ 0.0007|0.0069$\pm$ 0.0008|0.0010$\pm$ 0.0003|19.6278$\pm$ 0.5816|0.5659$\pm$ 0.0089|0.4814$\pm$ 0.0073|0.2674$\pm$ 0.0474|
> |EGNN|OOD|19.1906$\pm$ 1.6008|0.0279$\pm$ 0.0025|0.0212$\pm$ 0.0015|0.0236$\pm$ 0.0020|0.0312$\pm$ 0.0022|0.0058$\pm$ 0.0016|181.2595$\pm$ 8.1999|5.6586$\pm$ 0.2026|2.4465$\pm$ 0.0459|2.0791$\pm$ 0.1600|
> |MACE|ID|5.5626$\pm$ 0.1484|0.0617$\pm$ 0.0444|0.0150$\pm$ 0.0025|0.0135$\pm$ 0.0015|0.0181$\pm$ 0.0008|0.0018$\pm$ 0.0000|9.7863$\pm$ 0.1859|0.3221$\pm$ 0.0096|0.4295$\pm$ 0.0780|0.1335$\pm$ 0.0163|
> |MACE|OOD|38.8647$\pm$ 17.6504|0.0670$\pm$ 0.0849|0.0339$\pm$ 0.0052|0.0247$\pm$ 0.0017|0.0409$\pm$ 0.0015|0.0021$\pm$ 0.0000|68.2796$\pm$ 5.4458|1.5433$\pm$ 0.0193|2.2281$\pm$ 0.1329|0.2285$\pm$ 0.0260|
> |ET|ID|29.9687$\pm$ 17.7615|0.0081$\pm$ 0.0004|0.0027$\pm$ 0.0001|0.0031$\pm$ 0.0001|0.0043$\pm$ 0.0001|0.0006$\pm$ 0.0002|28.1942$\pm$ 14.7870|0.4902$\pm$ 0.1445|0.3677$\pm$ 0.0020|0.1596$\pm$ 0.0519|
> |ET|OOD|52.5017$\pm$ 48.1789|0.0479$\pm$ 0.0097|0.0220$\pm$ 0.0006|0.0236$\pm$ 0.0008|0.0271$\pm$ 0.0010|0.0171$\pm$ 0.0003|297.9813$\pm$ 30.8768|6.5682$\pm$ 0.1792|2.2567$\pm$ 0.0128|3.4053$\pm$ 0.0551|
> |ModernBERT|ID|14.6834$\pm$ 1.5756|0.0117$\pm$ 0.0004|0.0064$\pm$ 0.0012|0.0076$\pm$ 0.0018|0.0095$\pm$ 0.0018|0.0007$\pm$ 0.0003|40.0831$\pm$ 5.1632|0.8704$\pm$ 0.2304|0.6975$\pm$ 0.0396|0.4074$\pm$ 0.1429|
> |ModernBERT|OOD|44.4924$\pm$ 4.7856|0.0287$\pm$ 0.0064|0.0216$\pm$ 0.0011|0.0232$\pm$ 0.0033|0.0324$\pm$ 0.0029|0.0017$\pm$ 0.0008|228.7090$\pm$ 9.0356|2.4888$\pm$ 0.1506|2.6566$\pm$ 0.1270|0.6106$\pm$ 0.0945|
>
> We omit some rows due to response space constraints, but will update the manuscript with the complete Table.
>
> We find that our conclusions do not change as the models generally perform similarly within the margin of error. The models with the lowest average RMSE remain the same, with newly added models (see Reviewer YDtq) performing well on certain tasks. We note that the discrepancy between the ID and OOD performance for the tasks remains and is beyond the variation in the predictions. Furthermore, we still see the difference in capabilities between 3D, graphical, and SMILES representations.
>
> We conduct additional statistical tests on the observations from our data. First of all, we investigate the observation that ID performance is not necessarily predictive of OOD performance. We fit a Gaussian Process model to predict OOD $R^2$ values using the ID $R^2$ values. For all tasks, we notice a large variance in the predictions for high ID $R^2$ values, suggesting a high uncertainty in the model’s prediction. Furthermore, we also perform distance correlation permutation tests on the ID and OOD values. We see a moderate correlation over all tasks (d=.4621, p=.0010) from the distance correlation permutation test. Running the test for individual properties, certain properties, such as mu and gap, are of particular interest where we fail to reject the null hypothesis.
>
> Furthermore, we measure the effects of different representations on OOD performance. While we note that the samples are not truly random and may present selection bias, we believe the following tests capture our observation. We categorize models as described in Table 1 into 3 categories: 3D, graph, and transformer models use their OOD $R^2$ values. We perform a Kruskal-Wallis test over these sets and verify there is a statistically significant difference (p=0.016, N=420) within these sets. We then perform a Mann-Whitney U test for each pair with the appropriate null hypothesis. For OOD values over all tasks, we can see that 3D models' OOD $R^2$ values exceed their transformer counterparts(p=.0046), and graph OOD values exceed transformers (p = 0.007). We further analyze the differences for each property individually using the Kruskal-Wallis test (N = 35). For electronic spatial extent (p = 0.00005), heat of formation (p = 0.00002), specific heat (p = 0.008), and polarizability (p = 0.007), we find strong evidence of differences among the model categories. We run the Mann-Whitney U test and see that our selection of 3D models outperformed Transformer models (P=.00001) and graph models (P=0.008) for HoF. We omit the full results due to space limitations, but plan on including all the results in the manuscript.
>
> We would like to thank the reviewer for their thoughtful suggestions. While the additional tests do not change our conclusions, they allow us to provide statistical significance to our observations.
>
> > Table 1 is really difficult to read and interpret. It could be substituted by some kind of figure conveying the same information in a more intuitive way. The table is valuable and informative, but could be kept in the appendix.
>
> We thank the reviewer for the helpful feedback. We agree that Table 1 is challenging to interpret due to the large number of datasets and models. To provide a more visually understandable presentation of these results, we will swap Table 1 with Figure 10 of the Appendix, which provides a heatmap visualization of the ID and OOD RMSE performance.
>
> As I mentioned in S1.3., the fact that the study is limited to quantum properties, limits the generalizability of the observations made in the benchmark. I wonder if the authors have considered including other tasks from MoleculeNet where the 3D structure wouldn't be as important (some of the biomedical ones, for example) and if so, why did they choose against including them?
>
> We thank the reviewer for this suggestion. Following the comments of Reviewer Jmys, we have added additional benchmarking to include the Lipophilicity benchmark from MoleculeNet[arXiv:1703.00564], which contains 4200 experimental measurements of the octanol/water distribution coefficient, which is of relevance for drug compounds. The inclusion of the Lipophilicity dataset serves as an exemplary dataset for performing OOD evaluations on experimentally measured properties, rather than being limited to only computed physicochemical properties.
>
> In general, we did consider adding additional tasks from MoleculeNet to further expand our BOOM benchmark. A large number of datasets from MoleculeNet were not suitable for BOOM since they are designated as classification tasks (PCBA, MUV, HIV, BBBP, Tox21, ToxCast, SIDER, and ClinTox). We also chose not to use the PDBbind dataset as it includes biomolecular complexes, which are not supported by most of the benchmark models used in this work. Finally, among the remaining regression datasets in MoleculeNet, we originally decided not to include them due to their small size (BACE (n=1,522), ESOL (n=1,128), FreeSolv (n=643)), which may limit the statistical significance of our results. We have added Lipophilicity (n=4200) to BOOM since it is the largest example of these MoleculeNet datasets. We refer to our rebuttal to Reviewer Jmys for our benchmark results on the Lipophilicity dataset.

---

> > ### Author Response · Authors · 2025-08-06
> >
> > Dear Reviewer 4Hma,
> >
> > We appreciate the time and effort you've already put into reviewing our work, and we are eager to address any questions or clarifications you might have regarding our paper or the rebuttal.
> >
> > Please don't hesitate to reach out by posting any additional comments. Thank you again for your valuable contribution to the review process.

---

> > > ### Comment · Reviewer_4Hma · 2025-08-07
> > >
> > > Dear authors,
> > >
> > > Thank you for the impressive amount of work you have dedicated to addressing the concerns that I've raised, which with the proposed changes and additional experiments have been satisfactorily address. I have updated my score to 5: Accept.

---

### Official Review · Reviewer_YDtq · 2025-06-26

**Rating:** 4
**Confidence:** 5

**Summary:**

This paper presents BOOM, a benchmark for evaluating out-of-distribution (OOD) performance of molecular property prediction models. The authors evaluate 12 models across 10 molecular property prediction tasks, finding that no existing model achieves strong OOD generalization across all tasks. The benchmark includes datasets derived from QM9 and the 10k Dataset, with OOD splits defined based on property value distributions using kernel density estimation.

**Additional Feedback:**

This paper addresses an important and practical problem in molecular machine learning and provides valuable insights for the community. The benchmark framework is well-designed and the experimental analysis is comprehensive. However, there are clear opportunities for improvement that would significantly enhance the impact and completeness of this work.

The primary recommendation is to include recent state-of-the-art 3D molecular models, especially the models mentioned above, which would provide a more complete assessment of current capabilities and significantly enhance the benchmark's impact. Additionally, more nuanced positioning relative to existing OOD work would strengthen the contribution.

With these enhancements, this could become a highly valuable resource for the molecular ML community and help drive progress toward more robust models.

**Dataset Code Accessibility:**

Yes

**Ethical Considerations:**

No, there are no or only very minor ethics concerns

**Final Justification:**

The rebuttal has demonstrated a clear path to resolving the key weaknesses identified in the initial review.

**Limitations Weaknesses:**

1. The most significant opportunity for improvement lies in the model selection, particularly for 3D molecular architectures. Recent developments in this space may show different OOD behavior patterns, which would be more informative to readers and the broader research community. Including models such as Geoformer and GotenNet would provide a more complete picture of current capabilities and significantly enhance the benchmark's impact for guiding future research directions.
2. While the property-based OOD approach is valuable, the paper could benefit from more nuanced positioning relative to existing benchmarks. A brief discussion of how this approach complements existing structural/compositional OOD methods (rather than replacing them) would strengthen the contribution and clarify the unique value proposition.
3. The hyperparameter optimization appears to be conducted independently for each model-task combination, but the methodology for ensuring fair comparison across different model architectures is not clearly described.

**Strengths Contributions:**

1. This paper makes an important contribution by providing a systematic benchmark for OOD molecular property prediction. While there are existing OOD datasets in related domains (OC20's catalyst-specific splits, materials-focused benchmarks), this work offers a unique perspective by focusing on property-distribution-based OOD splits for small molecules. The KDE-based methodology for defining OOD samples represents a thoughtful approach that complements existing structural and compositional OOD definitions.
2. The authors conduct extensive ablation studies examining the impact of pretraining strategies, hyperparameter optimization, molecular representations, and data augmentation on OOD performance. The finding that current pretraining strategies (MLM, PLM) do not improve OOD performance is particularly noteworthy.
3. The paper provides valuable insights for the chemistry ML community, including the observation that models with high inductive bias (3D and graph models) generally outperform transformer-based approaches on OOD tasks, and that correlation between pretraining and downstream tasks is crucial for OOD improvement.

---

> ### Author Rebuttal · Authors · 2025-07-31
>
> **General Response**: We thank the reviewer for the feedback and useful suggestions to improve our work. In this work, we have developed a large-scale benchmark of OOD performance of state-of-the-art and popular ML models with more than 150 pairs of model-task experiments. We feel that our BOOM benchmark has the potential to serve as an important systematic benchmark for OOD molecular property prediction that will guide the chemistry community towards the development of more generalizable models. Thanks to the guidance provided by the reviewer, we have added results on two additional 3D molecular architectures (Geoformer and GotenNet), which help to further expand the scope of BOOM. With the addition of these additional models, we now feel that BOOM is an even more complete assessment of the space of current model capabilities.
>
> **Summary of new results:**
>
> 1.) We have added preliminary results for two additional state-of-the-art 3D molecular architectures, Geoformer and GotenNet. These results provide further details on the importance of 3D representations for OOD prediction and capture the current state of practice.
>
> 2.) We have also added an additional dataset, Lipophilicity, to our BOOM benchmark. The inclusion of this additional dataset serves to expand the scope of BOOM to encompass an experimentally-derived dataset, with high relevance to drug discovery.
>
> 3.) We have performed two additional training runs for the results in Table 1, thereby providing greater statistical significance to our results.
>
> > The most significant opportunity for improvement lies in the model selection, particularly for 3D molecular architectures. Recent developments in this space may show different OOD behavior patterns, which would be more informative to readers and the broader research community. Including models such as Geoformer and GotenNet would provide a more complete picture of current capabilities and significantly enhance the benchmark's impact for guiding future research directions.
>
> Thank you for the great suggestion and also providing new models for us to investigate. We appreciate the advice and the opportunity to improve the benchmark. We have performed preliminary benchmark runs on the mentioned models, Geoformer [1] and GotenNet [2], and updated the results and analysis in Tables 1 and 2. Below, we include a summary of the RMSE results of GeoFormer and GotenNet, averaged across three training runs.
>
> Model    |Split|HoF |Dens. |HOMO  |LUMO  |Gap  |ZPVE   |R2  |Alpha|Mu   |Cv
> ---------|-----|----|------|------|------|-----|-------|----|-----|-----|------
> Geoformer|ID   |17.8|0.0071|0.0027|0.0028|0.005|0.0003 |10.9|0.325|0.847|0.1244
> Geoformer|OOD  |43.3|0.037 |0.0157|0.0186|0.024|0.0056 |63.8|4.201|2.544|1.3539
> GotenNet |ID   |4.71|0.1044|0.1407|0.1063|0.240|0.0117 |11.8|0.552|0.319|0.1967
> GotenNet |OOD  |13.5|0.1227|0.4447|0.3196|0.622|0.01457|16.7|1.825|2.173|0.3023
>
>
> We note that these initial experiments reflect an initial implementation of the Geoformer and GotenNet models. Due to the short time frame, the models may not have sufficiently converged for all tasks, and further hyperparameter tuning is required and will be completed in time for publication. Nevertheless, we note a few important results from this initial benchmarking. First, we note that GotenNet achieves top ID performance on HoF and Mu. Geoformer obtains very strong Cv results but has a large RMSE due to outliers, possibly due to insufficient convergence. Both models show impressive performance on electronic spatial extent as well.   These notable findings warrant additional exploration to better understand how the specific choices in model architecture lead to this strong performance. More broadly, it is also interesting to note that all of the 3D models in BOOM (Geoformer, GotenNet, MACE, Graphormer, and ET) all achieve strong in-distribution performance on the Cv task, especially when compared to the 2D graph representations. Future work will seek to further understand how these 3D molecular architectures impact OOD performance.
>
> > While the property-based OOD approach is valuable, the paper could benefit from more nuanced positioning relative to existing benchmarks. A brief discussion of how this approach complements existing structural/compositional OOD methods (rather than replacing them) would strengthen the contribution and clarify the unique value proposition.
>
> Thank you for the suggestion and for noting this particular nuance.  We agree that our work is a complementary approach to structural/compositional OOD methods and may be used in conjunction with such methods to measure model robustness. Depending on the application and knowledge of the domain, one may require models to perform well in one definition over another or even both. For example, if a discovery pipeline is attempting to extrapolate in the structure space while keeping molecular properties the same, as is the case in lead molecule optimization, structure-based OOD generalization is a key characteristic for the ML model. On the other hand, if the discovery pipeline is focused on extending the frontier of the capabilities of known molecules, property-based OOD is needed. In certain pipelines, both characteristics may be required. Since structure-based OOD is highly dependent on how one defines the space, our property-based OOD splitting can serve to show whether or not the structure-OOD space has been defined in a way that is relevant to the properties of interest. We wanted to highlight works with structural/compositional OOD methods. We highlight Mole-OOD[3], Drug-OOD[4], Omee et al.[5] as examples of compositional/structural OOD methods in our related work, as complementary methods to ours, and will update the manuscript accordingly.
>
> > The hyperparameter optimization appears to be conducted independently for each model-task combination, but the methodology for ensuring fair comparison across different model architectures is not clearly described.
>
> Yes, we optimize independent learning hyperparameters, such as batch size, learning rate, and training time, independently for all models and tasks. We use model architectures as defined in the literature or publicly available code, without any changes. This choice was made such that the models in BOOM are representative of the performance that the chemistry community can expect if they are directly using the original model architectures. For our benchmarking of chemical foundation models, we use saved checkpoints from pre-trained models whenever possible and randomly initialized model weights when we train from scratch.
>
> [1] Wang, Yusong, et al. "Geometric transformer with interatomic positional encoding." Advances in Neural Information Processing Systems 36 (2023): 55981-55994.
>
> [2] Aykent, Sarp, and Tian Xia. "Gotennet: Rethinking efficient 3d equivariant graph neural networks." The Thirteenth International Conference on Learning Representations. 2025.
>
> [3] Yang, Nianzu, et al. “Learning Substructure Invariance for Out-of-Distribution Molecular Representations.” Advances in Neural Information Processing Systems 35 (2022): 12964-12978.
>
> [4] Ji, Yuanfeng et al. “DrugOOD: Out-of-Distribution (OOD) Dataset Curator and Benchmark for AI-aided Drug Discovery -- A Focus on Affinity Prediction Problems with Noise Annotations” arXiv:2201.09637.
>
> [5] Omee, Sadman et al. “Structure-based out-of-distribution (OOD) materials property prediction: a benchmark study” npj Computational Materials 10, 144 (2024).

---

> ### Comment · Reviewer_YDtq · 2025-08-01
>
> Dear Authors,
>
> Thank you for your rebuttal and the additional experiments with Geoformer and GotenNet. I have a few follow-up questions regarding your methodology and results.
>
> **1. Validation Set Usage**
>
> Could you clarify whether you used validation sets during your experiments? Specifically:
> - What methodology did you use for hyperparameter selection?
> - If validation sets were used, what percentage of the training data was reserved for validation?
>
> This information is important for reproducibility and should be mentioned in the paper.
>
> **2. Results Verification**
>
> I have concerns about the GotenNet results you reported. I recently conducted a quick evaluation using GotenNet (4 layers) on the HOMO target, training for approximately 35 epochs (~1 hours on A100 80GB GPU). My results were:
> - ID: 0.0043
> - OOD: 0.0097
>
> However, your reported GotenNet results for HOMO are:
> - ID: 0.1407
> - OOD: 0.4447
>
> This represents approximately a 32x difference in ID performance and 46x difference in OOD performance. Could you please verify these results? For QM9 targets, the reference implementations typically already include optimized hyperparameters, so extensive hyperparameter tuning should not be necessary. This significant discrepancy concerns me about the reliability of the rest of the baseline results in your benchmark. While I am unable to verify every single model's performance within the review timeframe, I strongly urge the authors to thoroughly verify all reported results in a timely manner to ensure the integrity of the benchmark.
>
> **3. Code Repository Issues**
>
> I encountered several issues with your GitHub repository:
> - `CoordDataset(property='homo', split='train')` returns a TypeError: `retrieve_qm9_dataset() missing 1 required positional argument: 'split_file'`
> - The `retrieve_qm9_dataset()` function expects a `split_file` argument that is never provided in the code at line 33 of [boom/data/load_processed_3d_data.py](https://github.com/FLASK-LLNL/BOOM/blob/35a72a5ce90ea4673057891fd7406e21c9a0692c/boom/data/load_processed_3d_data.py#L33)
>
> Could you address these implementation issues to ensure the code works out-of-the-box for reproducibility?
>
> **4. Optional Suggestion**
>
> While not required, providing PyTorch Geometric/DGL versions of CoordsDataset would enhance accessibility for the broader community.
>
> These technical issues are important to resolve, as this benchmark is intended to serve the molecular ML community. Ensuring reproducibility and correctness of results is paramount for a benchmark paper.

---

> > ### Author Response · Authors · 2025-08-04
> >
> > >What methodology did you use for hyperparameter selection?
> > If validation sets were used, what percentage of the training data was reserved for validation?
> >
> > We thank the reviewer for mentioning this important point. Our validation sets are taken as 10% of the training data for the hyperparameter selection, as is common in prior chemical benchmarking papers.[1,2]
> > As we had multiple different types of models, we started with publicly available settings for the starting hyperparameters (as noted in Section 10), but also performed hyperparameter sweeps (with grid search) on the non-architectural components, such as learning rate and training steps. We do not update architectural details to match the use case of practitioners using off-the-shelf models. The GNN ablation uses the architecture detailed in [3] and training instructions listed in Appendix C.3.  The baseline models (Random Forest and MLP), use model hyperparameters previously reported.[1,2]
> >
> > As shown in Figure 5, we find that model performance and trends are not significantly affected by hyperparameter optimization. MACE requires a validation set for early stopping and we use a standard .10 fraction. We include training details for our models in Appendix section 10, and will update with validation set usage and hyperparameter tuning details.
> >
> > >I have concerns about the GotenNet results you reported. I recently conducted a quick evaluation using GotenNet on the HOMO target. My results were:
> > ID: 0.0043
> > OOD: 0.0097
> >
> > We sincerely apologize for this error in our results. We share the viewpoint that ensuring reproducibility and correctness is of the utmost importance for a benchmark paper and we have taken significant effort towards this end. We have determined that the anomalously large RMSE values previously reported were due to a rescaling error. We had noted the larger-than-expected error and incorrectly attributed it to hyperparameter/training issues. We have corrected this error, and the corrected RMSE values are reported below. We note that the HOMO RMSE values obtained are in good agreement with those obtained by the reviewer (ID=0.0043, OOD=0.0097), further validating our results. To further ensure the veracity of the results, we have visually verified the scatter plots for all of the GotenNet experiments.
> >
> > | Model    | Split | Seed |HOMO|LUMO|GAP|ZPVE|R^2|alpha| mu   | Cv   |
> > | -------- | ----- | ---- | ------ | ------ | ------ | ------- | ---- | ----- | ---- | ---- |
> > | GotenNet | ID    | Avg.    | 0.0052 | 0.0039 | 0.0088 | 0.00043 | 11.8 | 0.55  | 0.32 | 0.20 |
> > | GotenNet | OOD   | Avg.    | 0.0163 | 0.0138  | 0.0229 | 0.00054 | 16.7 | 1.82  | 2.17 | 0.30 |
> >
> > As noted by the reviewer, the GotenNet model hyperparameters are optimized for the QM9 dataset, but not the 10k dataset. Since performing careful hyperparameter tuning for the 10k dataset is beyond the scope of work that is permitted in this response period, we will leave this task as future work to be completed prior to publication of the manuscript.
> >
> > As a further validation of the reliability of our results, we compute the ID Pearson correlation coefficients below to further support that the models are performing correctly.
> >
> > | Model    | Split |Seed|HOMO|LUMO|GAP |ZPVE |R^2 |alpha|mu  |Cv  |
> > | -------- | ----- | ---- | ----- | ----- | ----- | ------ | ----- | ----- | ----- | ----- |
> > | GotenNet | ID    | 1    |0.927|0.994|0.979|0.9999|0.998|0.997|0.932|0.998|
> >
> > We note that GotenNet achieves strong performance on many tasks. Akin to all the other models reported in the original version of the manuscript, we will provide readers with further assurance of model performance by creating scatter plots and statistical analyses of all GotenNet and Geoformer experiments.
> >
> > >I encountered several issues with your GitHub repository:
> > CoordDataset(property='homo', split='train') returns a TypeError: retrieve_qm9_dataset() missing 1 required positional argument: 'split_file'
> > The retrieve_qm9_dataset() function expects a split_file argument that is never provided in the code at line 33 of boom/data/load_processed_3d_data.py
> > Could you address these implementation issues to ensure the code works out-of-the-box for reproducibility?
> >
> > We sincerely apologize for the inconvenience. We identified this issue during the rebuttal, and all subsequent runs were done using the fixed code. The code is fixed to work out of the box and will be upstreamed as soon as we are allowed to. We apologize again for the inconvenience.
> >
> > >While not required, providing PyTorch Geometric/DGL versions of CoordsDataset would enhance accessibility for the broader community.
> >
> > Thank you for the suggestion, as our intention is for the benchmark to be as widely usable as possible and be used by the community. While we provide native PyTorch datasets, we plan to incorporate the advice and make library-specific dataset wrappers available as well.
> >
> > [1.] J. Chem. Inf. Model. 2019, 59, 8, 3370–3388
> > [2.] arXiv:1703.00564
> > [3] arxiv:2102.09844

---

> > ### Comment · Reviewer_YDtq · 2025-08-04
> >
> > Thank you for your response.
> >
> > I appreciate you clarifications regarding the hyperparameter selection, code repository, and the GotenNet results.
> >
> > I have a follow-up question regarding the discrepancy in GotenNet performance. While you've corrected the RMSE values for HOMO, the original issue still concerns me. My quick evaluation was conducted for only 35 epochs, whereas GotenNet's default hyperparameter settings are optimized for 1000 epochs. Therefore, with a full training run, I would expect the model's performance to be even better than what I observed, suggesting that something is still amiss.
> >
> > This leads me to question the methodology used for evaluating these models. It seems plausible that you integrated the GotenNet model into your own pipeline, which may have inadvertently introduced the scaling error you found. Given the sensitivity of 3D models to specific hyperparameters and training schedules, integrating your OOD dataset directly into the reference implementations would be the most robust approach to ensure fair comparisons. While I understand that implementing this for all models within a tight timeframe is challenging, the significant discrepancy in the GotenNet results, even after correction, makes me question the reliability of all the baselines in your benchmark. I would strongly urge you to reconsider your evaluation approach to ensure the integrity of the benchmark for the broader community.
> >
> > To help me fully understand the discrepancy, could you please confirm the exact data sizes for the training, ID, and OOD splits on the HOMO target? I'm concerned that there may be other underlying problems with the data or code on current upstream that are causing inconsistent results.

---

> ### Author Response · Authors · 2025-08-05
>
> Thank you for your response. We sincerely apologize for the confusion regarding the results.  The preliminary results we posted were obtained from GotenNetB models trained over 50 epochs and not the full 1000 epochs of the run due to response time constraints. As noted in the GotenNet paper, full training runs on GotenNetB require 1.15GPU days per task, which made full training on all 30 OOD tasks (34.5GPU days total) infeasible on our available computational resources within the response period. For our initial response, we wanted to include the early results as they were promising results. Nevertheless, we plan on providing results from the full training runs at the recommended settings, with multiple runs, as well as perform any further tuning required prior to publication. We apologize again for the confusion and not clearly stating the exact training time details.
>
> We used the publicly available GotenNet code and incorporated our benchmark. We agree that it is of the utmost important to ensure proper comparison. We have identified the rescaling issue in our integration and are confident the issue is isolated to this particular model. We hope that the strong agreement of our HOMO results on 50 epochs and that of the reviewer’s, trained over 35 epochs provides strong evidence that there is no notable discrepancy observed between our results.
>
> Following are the exact splits as well code to obtain them
>
> ood_dataset = CoordDataset("homo", "ood")
>
> id_dataset = CoordDataset("homo", "id")
>
> train_dataset = CoordDataset("homo", "train")
>
> print(len(ood_dataset), len(id_dataset), len(train_dataset))
>
>  9994 6098 117406
>
> We would like to again thank the reviewer and sincerely apologize for the confusion due to the lack of clarity of our response.

---

> > ### Comment · Reviewer_YDtq · 2025-08-05
> >
> > Your explanation for the initial discrepancies, specifically the training duration and the identified rescaling error, addresses my primary concerns about the reliability of the reported results. The commitment to running full, corrected experiments for the models in question before publication is a necessary step to ensure the integrity of the benchmark. The confirmation of the data splits and the quick fix to the repository issues are also reassuring.
> >
> > I am changing my score to a four. The rebuttal has demonstrated a clear path to resolving the key weaknesses identified in the initial review. Good luck!

---

### Official Review · Reviewer_Jmys · 2025-07-02

**Rating:** 5
**Confidence:** 4

**Summary:**

The paper presents a benchmark study of ML models for molecular property prediction, focusing on out-of-distribution (OOD) generalization. The work specifically targets the prediction of extreme property values, i.e. the tails of the label distribution. Using literature-known datasets of computed molecular properties, the authors define ten tasks with OOD train–test splits, and metrics to evaluate ID and OOD performance.

The authors systematically compare different ML model architectures, including 2D and 3D graph neural networks as well as SMILES-based transformer models. They find 1) a notable performance discrepancy between ID and OOD settings, and 2) the lack of a single model that is generally effective across tasks. Moreover, the effects of pretraining, hyperparameter tuning, and adding new training data, are investigated in systematic ablation studies. The results indicate that established approaches mainly improve ID predictivity, and that accepted conclusions in the field to not always apply to the OOD setting.

**Additional Feedback:**

* The authors report both ID and OOD performance and emphasize the discrepancy between them. However, it would be valuable to analyze whether ID performance is an indicator for OOD performance. One specific example is discussed in the main text, but a more systematic analysis would be interesting.
* In Tab. 1, it is stated that the baseline model operates on SMILES as the molecular representation. In the main text, the authors state that it uses molecular fingerprints – which are a distinctly different representation. If the baseline model results are included in the statistics in Tab. 2, this should be corrected.
* The choice of Random Forest as a baseline model is problematic. Due to its structure, a Random Forest cannot extrapolate beyond the label range seen during training. Other simple ML architectures might be a better choice.
* The bar plots in Fig. 3 hide significant information by showing only the average score across multiple tasks. What is the distribution of scores over the tasks? Box plots or violin plots would be a better visualization here.
* The term ”data augmentation“ (section 4.5) is misleading. Traditionally, data augmentation refers to generating additional training data from the existing examples through domain knowledge (e.g. rotating/scaling/shifting images, or perturbing SMILES). Here, the authors refer to generating new data points with labels, which should be described differently.
* I am uncertain what the “data augmentation” section (section 4.5) adds to the benchmark study. In my opinion, the described workflow of generating additional training examples with OOD labels does not have practical relevance. How would one identify the right molecules to label in the first place?

**Dataset Code Accessibility:**

Yes

**Dataset Code Comments:**

The data and the splits can be readily accessed from the authors’ GitHub repository.

**Ethical Considerations:**

No, there are no or only very minor ethics concerns

**Final Justification:**

The paper addresses an extremely important and largely unsolved problem in molecular property prediction, namely the extrapolation to molecules with better properties. The benchmark experiments provide interesting insights, and showcase challenges which need to be addressed. During the rebuttal phase, the authors proposed a number of revisions to the manuscript. With these changes, clarity is improved, and the work is better placed in the context of existing literature and practical use cases.

**Limitations Weaknesses:**

* **Dataset and task diversity**: The evaluated datasets and tasks are very narrow. Most tasks stem from the QM9 dataset, which has been extensively used (and arguably overused) in molecular property prediction. QM9 is well-known not to be representative of chemical space, since it contains only very small molecules, many of which are of no practical relevance. Along the same lines, the benchmarks are limited to computed physicochemical properties.
* **Conceptual novelty**: The idea of creating train–test splits that focus on the tails of label distributions is not new, even though it is not widespread in molecular property prediction. Similar benchmarking studies, such as the recent preprint by Segal et al. (cited in the paper) have followed this strategy.

**Strengths Contributions:**

* The paper addresses an important – and largely unsolved – challenge in molecular machine learning: the reliable prediction of molecules with better properties than the ones currently known. This is a central problem in e.g. drug or materials discovery.
* The paper is clearly written and easy to follow. The study appears carefully designed and rigorously executed.
* The systematic ablation studies, especially the investigation of pretraining effects, offer interesting insights – which, if they generalized to other tasks – would challenge common assumptions in molecular property prediction. Similarly, the experiments on hyperparameter tuning provide useful insights for model development.

---

> ### Author Rebuttal · Authors · 2025-07-31
>
> **General Response**: We thank the reviewer for their review and their thoughtful suggestions to help improve our work.
> The idea of creating train–test splits that focus on the tails of label distributions is not new, even though it is not widespread in molecular property prediction. Similar benchmarking studies, such as the recent preprint by Segal et al. (cited in the paper) have followed this strategy.
> Although we agree with the reviewer’s note that our work is not the first to explore molecular property prediction on the tails of the distribution, our work contains several novel and important findings that should be emphasized:
> We develop a general and robust evaluation benchmark to evaluate OOD performance in the property space, a key challenge in the use and proliferation of molecular machine learning in industry and science.
> We perform a large-scale benchmark of OOD performance of state-of-the-art and popular ML models with more than 150 pairs of model-task experiments- nearly 10x more than in the prior work (Segal, et al.).
> As a direct result of the expansive scope of our benchmarking, we’re able to provide insights into how pre-training strategies, model architecture, molecular representation, and data generation strategies can affect OOD performance, all of which were not possible from prior work.
>
> **Summary of new results**: We have significantly expanded upon BOOM to include new datasets, model architectures, and additional statistical analyses. With these new experiments, we believe BOOM can serve as a comprehensive and robust OOD benchmark for the chemistry community:
>
> 1.)To further broaden the scope of BOOM, we have added additional benchmarking on the Lipophilicity dataset. The inclusion of this dataset broadens the application space of BOOM to encompass drug molecules, as well as to include an example of a dataset of experimentally measured properties.
>
> 2.)With the additional samples, we perform additional analysis of the trends we see, such as the correlation between ID and OOD performance, and the effect of representation on OOD for various tasks. Our results show statistically significant evidence that for our chosen tasks, ID and OOD are not strongly correlated (r=.4621, p=0.0010). Furthermore, we also verify significant evidence that the graph and geometric representation provide stronger generalization for certain tasks such as heat of formation, density, zero-point vibrational energy, and polarizability. We note geometric features are a strong inductive bias for these tasks.
>
> 3.) We have added a Multilayer Perceptron (MLP) regression model as an additional benchmark model to overcome some of the limitations of the random forest model architecture.
>
> > Dataset and task diversity: Most tasks stem from the QM9 dataset, which has been extensively used. QM9 is well-known not to be representative of chemical space, since it contains only very small molecules, many of which are of no practical relevance. Along the same lines, the benchmarks are limited to computed physicochemical properties.
>
> We agree with the reviewer’s sentiment that the QM9 dataset has been overused. Nevertheless, we felt that it was important to include QM9 in BOOM as it provides a familiar point of reference, which helps to make the performance gap in the OOD evaluations stand out more clearly. In addition to QM9, BOOM also includes the 10K dataset, which does not have any restrictions on molecular size. The average molecular size of the 10K dataset is 43 total atoms, with the largest molecule containing 156 total atoms. Second, the 10K dataset was specifically developed to sample experimentally synthesized molecules that are relevant to important chemical safety applications. Altogether, the inclusion of the 10K dataset expands the scope of BOOM to include significantly larger molecules that are also of high practical relevance. Finally, we have added new benchmarking on the Lipophilicity dataset[arXiv:1703.00564], which contains 4200 experimental measurements of the octanol/water distribution coefficient, which is of relevance for drug compounds. The inclusion of the Lipophilicity dataset serves as an exemplary dataset for performing OOD evaluations on experimentally measured properties, rather than only computed physicochemical properties. Below, we report the RMSE values for a subset of the models on the Lipophilicity Dataset, averaged across 3 training runs, along with their standard deviations.
>
> | Model | MLP       | Random Forest | Regression Transformer | MoLFormer  | Chemprop  |
> | ----- | --------- | -------------- | ---------------------- | ---------- | --------- |
> | ID    | 0.866$\pm$0.09 | 0.548$\pm$0.001    | 1.139$\pm$0.02              | 0.473$\pm$0.006 | 0.463$\pm$0.01 |
> | OOD   | 2.041$\pm$0.2  | 1.576$\pm$0.006     | 1.164$\pm$0.003             | 0.956$\pm$0.004 | 1.051$\pm$0.02 |
>
> >However, it would be valuable to analyze whether ID performance is an indicator for OOD performance.
>
> We agree that the discrepancy between ID and OOD should be emphasized, as it is a surprising result. Here, we explore the relationship between ID and OOD performance further. We first fit a Gaussian Process to model the predictive power of ID performance for OOD performance, and see a high variance in the predicted OOD. For instance, when given an ID of .9, the predicted OOD is $.34 \pm .22$. This shows a high uncertainty in the GP prediction for high ID models. We see that high ID performance is not predictive of high OOD performance. We further perform correlation distance analysis with a permutation test and see that overall, there is a moderate (0.46) correlation between ID and OOD with high significance (p = 0.001). This decoupling between in-distribution and out-of-distribution performance, especially among models with high ID accuracy, underscores the critical need for dedicated OOD benchmarks to reliably assess generalization and robustness beyond the training distribution.
>
> > In Tab. 1, it is stated that the baseline model operates on SMILES as the molecular representation. In the main text, the authors state that it uses molecular fingerprints – which are a distinctly different representation. If the baseline model results are included in the statistics in Tab. 2, this should be corrected.
>
> We use RDKit features derived from SMILES and not the raw SMILES strings. The sub-category of SMILES in Table 2 does not include Random Forest. We apologize for the confusion. We will update the paper to make this clear and enumerate the models used in each category in Table 2.
>
> > “The choice of Random Forest as a baseline model is problematic. Due to its structure, a Random Forest cannot extrapolate beyond the label range seen during training. Other simple ML architectures might be a better choice.”
>
> We agree with the comment that Random Forest cannot extrapolate beyond the training distribution. The decision to include Random Forest in our benchmarking was inspired by prior work that showed strong performance[J. Chem. Inf. Model. 2019, 59, 8, 3370–3388]. Nevertheless, we have also included a MLP model with RDKit descriptors (as described in the above citation) as an additional baseline model. Below, we present the RMSE values from this additional benchmarking.
> Model|Split|HoF  |Dens.|HOMO  |LUMO  |Gap  |ZPVE  |R2   |Alpha|Mu   |Cv   |Lipo
> -----|-----|-----|-----|------|------|-----|------|-----|-----|-----|-----|-----
> MLP  |ID   |13.66|0.053|0.0094|0.0091|0.013|0.0056|49.9 |0.817|0.696|0.384|0.866
> MLP  |OOD  |38.43|0.094|0.0247|0.0201|0.047|0.0137|470.9|6.859|2.389|0.593|2.041
>
> > The bar plots in Fig. 3 hide significant information by showing only the average score across multiple tasks. What is the distribution of scores over the tasks?
>
> Thank you for the suggestion. We agree that summary statistics inadvertently hide information for the different tasks, although we felt that these results are too detailed to include in the main text. In the current version of the manuscript, we have detailed figures that provide task-specific values of the RMSE and R^2 metrics for all of these models (Figure 9).
>
> > The term ”data augmentation“ (section 4.5) is misleading.
>
> Thank you for the suggestion. We agree that the term may be confusing, so we will rename this section as “Data Ablation Study” to more accurately reflect our experimental design of selectively withholding samples from the tail ends of the distribution.
>
> > I am uncertain what the “data augmentation” section adds to the benchmark study.
>
> We appreciate the reviewer’s comments and would like to clarify its purpose. From the other sections of the paper, we consistently found that models achieve strong ID performance, but with poor OOD performance. This raises the question: “To what extent can OOD performance be improved by including a small number of molecules with extreme property values in training?”. This question has strong practical relevance as one could imagine improving generalization by using 1) a molecular generative model to generate molecules with extreme property values, 2) performing simulations to get the property labels of those extreme molecules, and then 3) training the property prediction model on those property labels.
>
> In our specific case, we cannot perform this exact experiment without the exact DFT pipeline used to create the QM9 dataset (Step 2 above). Instead, we designed our experiment in Section 4.5 to first start with the normal QM9 training split as the training examples. Then, to recreate molecule generation and calculating their properties with DFT, we systematically add some of the extreme-valued molecules from the QM9 OOD test set into the training data. Our experiments highlight that such a strategy is viable, achieving improved performance on 7/8 QM9 tasks. These results, therefore, suggest that a more extensive molecule generation campaign may be a fruitful path towards achieving chemical foundation models with strong generalization.

---

> > ### Comment · Reviewer_Jmys · 2025-08-03
> >
> > I thank the authors for their thoughtful responses and the additional experiments. Most of my initial concerns have been addressed convincingly.
> >
> > My only remaining concern about the purpose of the “data augmentation” section in the original manuscript. I agree with the authors that, given the discrepancy between ID and OOD performance, adding a few molecules with extreme property values would likely improve the model. However, it remains unclear how such molecules would be identified in practice.
> > > …using 1) a molecular generative model to generate molecules with extreme property values, 2) performing simulations to get the property labels of those extreme molecules, and then 3) training the property prediction model on those property labels.
> >
> > This suggestion creates a chicken-and-egg problem. It assumes a generative model can produce molecules with property values from the desired tail of the distribution, despite being trained on data which does not cover that tail of the distribution. In their paper, the authors convincingly show that regression models do not generalize well to such OOD cases. Why would this be any different for a generative model?
> >
> > As of now, the only practical strategy that I see is to use explorative strategies from Bayesian Optimization or Active Learning to recommend new molecules. However, this requires the model to recognize the extreme-property domains as underexplored regions in feature space – which is itself a strong assumption.
> >
> > If this aspect is addressed in the revised manuscript, I would be willing to raise my score to 5.
> >
> > **Minor Comments**
> > > there is a moderate (0.46) correlation between ID and OOD with high significance
> >
> > Even though the correlation is moderate, it is higher than I had expected based on the original paper. This suggests that models with higher ID performance can, on average, lead to better OOD performance, justifying the selection of models based on ID performance. Nonetheless, I agree with the authors that improved OOD performance benchmarks are needed.

---

> > > ### Author Response · Authors · 2025-08-06
> > >
> > > >[...] In their paper, the authors convincingly show that regression models do not generalize well to such OOD cases. Why would this be any different for a generative model?
> > > As of now, the only practical strategy that I see is to use explorative strategies from Bayesian Optimization or Active Learning to recommend new molecules. However, this requires the model to recognize the extreme-property domains as underexplored regions in feature space – which is itself a strong assumption.
> > >
> > > We thank the reviewer for their thoughtful comments and appreciate the opportunity to clarify this important point.
> > >
> > > We first want to clarify that the data augmentation experiment demonstrates that even a few thousand out-of-distribution (OOD) samples can effectively convert an OOD region into an in-distribution (ID) one. This has important implications for real-world applicability. Our experiment shows that property prediction performance can be improved with minimal OOD data-highlighting that even a relatively small number of molecules in the OOD region can significantly enhance model generalization. Our hope with this experiment is that it inspires future work exploring how targeted generation can improve OOD generalization, rather than definitively prescribing a solution for solving OOD generalization.
> > >
> > > We acknowledge the concern around the feasibility of identifying useful OOD points in the first place and agree that this is a significant, unsolved challenge. We also agree that Bayesian Optimization or Active Learning approaches are likely necessary. Nevertheless, we point to recent work that demonstrates that generative models combined with active learning may be able to extrapolate in property space.  Consistent with the reviewer’s intuition, prior work [arXiv:2501.02059] has shown that generative models without active learning were not able to extrapolate beyond the training data (Figure 2a). However, once active learning on DFT simulations was incorporated into the generative loop, the model showed strong potential to generate molecules with properties that extend far beyond the training data (Figure 2), demonstrating an ability to extrapolate in property space. In another more extreme example, [arXiv:2502.14842], the authors used their STGG+ autoregressive generative model with active learning to discover molecules with fosc of 27.7, compared to a maximum of 9.3 in their training data, and a fosc of 13.01 without active learning.  These two examples serve to empirically demonstrate that iteratively generating molecules, labelling them with ground-truth simulations, and then retraining the property prediction models may lead generative models to recognize extreme-property domains in chemical space.
> > >
> > > Conceptually, we hypothesize that this approach may be possible because this iterative active learning will continually trend towards molecules with improved properties as long as the property prediction models are able to determine the relative ordering of molecules with respect to the property of interest, rather than needing quantitatively correct predictions. Then, it is the ground truth simulations (DFT) that will provide the true property labels of the molecules. Our scatter plots shown in Figure 12 as an example, show that the property prediction models do seem to have this capability as the most extreme-property molecules are consistently identified, and thus, would be preferentially generated.
> > >
> > > Encouragingly, the proposed overall approach of improving OOD performance through iterative active learning has already seen some reported success in the literature. In arXiv:2501.02059, the authors note that after three iterations of active learning, the Δ𝐻, prediction RMSE reduces by 83% (from 110kcal/mol to 19kcal/mol) when evaluated on hold-out test molecules from across the entire active learning run. Similarly, in DOI: 10.1038/s41586-023-06735-9, the authors found that multiple iterations of active learning to generate novel inorganic structures reduced the error on structure-based OOD energy predictions from >200meV/atom down to ~25meV/atom. Although there is still much to explore, we feel that there is some growing evidence to believe that OOD generalization can be improved in this manner.
> > >
> > > We thank the reviewer again for this important note and will incorporate this discussion to the appendix of the revised manuscript.
> > >
> > > >There is a moderate (0.46) correlation between ID and OOD with high significance.[ …] This suggests that models with higher ID performance can, on average, lead to better OOD performance, justifying the selection of models based on ID performance.
> > >
> > > We thank the author for the interesting note and agree with the comments made. We plan to make these findings available in the revised version of the manuscript as it is likely to be of interest to readers. At the same time, we will produce plots of task-specific ID vs. OOD performance to better understand the task-dependence of these findings.

---

> > > > ### Comment · Reviewer_Jmys · 2025-08-06
> > > >
> > > > I thank the authors for their thoughtful response! I agree with most of the statements, in particular
> > > >
> > > > > the feasibility of identifying useful OOD points [...] is a significant, unsolved challenge
> > > >
> > > > Sample efficiency in identifying useful OOD points will likely become an additional concern, particularly when generating labels is expensive. Even in DFT simulation, performing a thousand calculations at a reasonable level comes with non-negligible computational cost; in an actual wet-lab scenario, a thousand experiments likely exceed the budget. Most likely, the mechanism for recommending useful OOD points (generative model and/or active learning) will have low sample efficiency, leading to much higher cost for generating the OOD labels.
> > > >
> > > > In general, I think the experiments using additional OOD samples provide some interesting insights. However, it should not be neglected that, in practice, such workflows are still far from established – and, in my opinion, are highly speculative at this point in time. I believe that this should be reflected in the manuscript's main text.

---

> > > > > ### Author Response · Authors · 2025-08-06
> > > > >
> > > > > >In general, I think the experiments using additional OOD samples provide some interesting insights. However, it should not be neglected that, in practice, such workflows are still far from established – and, in my opinion, are highly speculative at this point in time. I believe that this should be reflected in the manuscript's main text.
> > > > >
> > > > > We thank the reviewer for their thoughtful and constructive feedback. We agree that the current main text of the manuscript would greatly benefit from an explicit discussion of these raised points. To make sure that the challenges of this approach are explicitly and clearly reflected in the main text, we will add the following paragraph to Section 4.5:
> > > > >
> > > > > “In this experiment, we seek to explore to what extent adding a relatively small number of molecules in the OOD region can improve model generalization. We emphasize that the feasibility of using molecular generative models to efficiently generate useful OOD molecules is still a significant and unsolved challenge. The throughput at which property data can be acquired, whether through experimental measurements or simulations, is also strongly property-dependent. The goal of this experiment is not to prescribe a path towards generating OOD molecules, but to better understand the sensitivity of property prediction models to the addition of OOD molecules.  We provide a complete discussion of this approach and related prior work in the Appendix.”
> > > > >
> > > > >  Due to space limitations, we will add our previous response to the Appendix to allow for a complete and detailed discussion of this point.

---

> > > > > > ### Comment · Reviewer_Jmys · 2025-08-06
> > > > > >
> > > > > > Thank you! As indicated above, I will raise my score to 5.

---

### Decision · Program_Chairs · 2025-09-18

**Decision:**

Accept (poster)

**Comment:**

The paper provides a benchmark for molecular out-of-distribution prediction and thus considers an important practical topic. By focusing on molecules more generally, beyond drugs, it fills a gap in the existing benchmark landscape.

All reviewers submitted comprehensive reviews, yet there was a strong divide in the initial scores. The main criticism got addressed in the rebuttal (statistical significance, more nuanced positioning of the work, additional results w/ 3D-based models). It has to be noted that the benchmark only focuses on two existing datasets QM9  and the 10k dataset. But the authors gave justification in the rebuttal, and for an initial work on the topic, the data seems enough.
The rebuttal discussion was in parts extensive and the reviewers participated as needed. Altogether they came to an accept recommendation (one borderline accept).